# Higher absorption enhancement of black carbon in summer shown by two year measurements at the high-altitude mountain site of Pic du Midi Observatory in the French Pyrenees

Sarah Tinorua[1], Cyrielle Denjean[1], Pierre Nabat[1], Thierry Bourrianne[1], Véronique Pont[2], François Gheusi[2], and Emmanuel Leclerc[2]

[1]CNRM, Université de Toulouse, Météo-France, CNRS, Toulouse, France
[2]Laboratoire d'Aérologie, UPS Université Toulouse 3, CNRS (UMR 5560), Toulouse, France

**Correspondence:** Sarah Tinorua (sarah.tinorua@umr-cnrm.fr), Cyrielle Denjean (cyrielle.denjean@meteo.fr)

**Abstract.** Particles containing Black Carbon (BC) strongly absorb light, causing substantial radiative heating of the atmosphere. The climate-relevant properties of BC are poorly constrained in high-altitude mountain regions, where many complex interactions between BC, radiation, clouds and snow have important climate implications. This study presents two-year measurements of BC microphysical and optical properties at the Pic du Midi (PDM) research station, a high-altitude observatory located at 2877 m above sea level in the French Pyrenees. Among the long-term monitoring sites in the world, PDM is subject to limited influence from the planetary boundary layer (PBL), making it a suitable site for characterizing the BC in the free troposphere (FT).

The classification of the dominant aerosol type using aerosol spectral optical properties indicates that BC is the predominant aerosol absorption component at PDM and controls the variation in Single Scattering Albedo (SSA) throughout the two years. Single-particle soot photometer (SP2) measurements of refractory BC (rBC) show a mean mass concentration ($M_{rBC}$) of 35 ng m$^{-3}$ and a relatively constant rBC core mass-equivalent diameter of about 180 nm, which are typical values for remote mountain sites. Combining the $M_{rBC}$ with in situ absorption measurements a rBC mass absorption coefficient ($MAC_{rBC}$) of 9.2 $\pm$ 3.7 m$^2$ g$^{-1}$ at $\lambda$=880 nm has been obtained, which corresponds to an absorption enhancement ($E_{abs}$) of $\sim$ 2.2 compared to that of bare rBC particles with equal rBC core size distribution. A significant reduction in the $\Delta M_{rBC}/\Delta CO$ ratio when precipitation occurred along the air mass transport suggests wet removal of rBC. However we found that the wet removal process did not affect the rBC size, resulting in unchanged $E_{abs}$. We observed a large seasonal contrast in rBC properties with higher $M_{rBC}$ and $E_{abs}$ in summer than in winter. In winter a high diurnal variability of $M_{rBC}$ ($E_{abs}$) with higher (lower) values in the middle of the day was linked to the injection of rBC originating from the PBL. On the contrary, in summer, $M_{rBC}$ showed no diurnal variation despite more frequent PBL conditions, implying that $M_{rBC}$ fluctuations are rather dominated by regional and long-range transport in the FT. Combining the $\Delta M_{rBC}/\Delta CO$ ratio with air mass transport analysis, we observed additional sources from biomass burning in summer leading to an increase in $M_{rBC}$ and $E_{abs}$. The diurnal pattern of $E_{abs}$ in summer was opposite to that observed in winter with maximum values of $\sim$ 2.9 observed at midday. We suggest that this daily variation may result from a photochemical process driving rBC mixing state rather than a change in BC emission sources.

Such direct two-year observations of BC properties provide quantitative constraints for both regional and global climate models and have the potential to close the gap between model predicted and observed effects of BC on regional radiation budget and climate. The results demonstrates the complex influence of BC emission sources, transport pathways, atmospheric dynamics and chemical reactivity in driving the light absorption of BC.

## 1   Introduction

Black Carbon (BC) is a light-absorbing carbonaceous aerosol produced by incomplete combustion of fossil fuels, including anthropogenic emissions from traffic, residential heating and cooking, power plants, industries, but also natural emissions such as biomass burning (Bond et al., 2013; Bond and Bergstrom, 2006). Recent scientific assessments of the 6[th] IPCC (Intergovernmental Panel on Climate Change) report (Szopa et al., 2021) estimates that BC is the most absorbing atmospheric aerosol with a best estimate of effective radiative forcing around +0.107 W m$^{-2}$ , thereby increasing the global mean surface air temperature by 0.063 °C for the period 1750–2019 (Szopa et al., 2021). The contribution of BC to climate change is estimated to be among the highest uncertainties (∼90%) in climate models, limiting their accuracy (Bellouin et al., 2020). The large uncertainty of BC direct radiative forcing due to BC-radiation interactions can be attributed, in addition to uncertainties in BC emissions and lifetime, to variations of its optical properties that are neglected by climate models (Matsui et al., 2018).

A crucial factor in estimating the BC radiative effect is the mass absorption cross-section ($MAC_{BC}$), which is defined as the light absorption-equivalent cross-section of BC per unit of mass concentration ($M_{BC}$). The $MAC_{BC}$ can be calculated either by dividing the measured absorption coefficient of BC by its mass concentration or by using Mie's Theory and the BC size distribution and coating thickness as input variables. Observations show that the BC radiative forcing is likely underestimated by around 10 to 40% in current climate models due to too low simulated $MAC_{BC}$ (Bond et al., 2013; Boucher et al., 2016; Matsui et al., 2018; Myhre and Samset, 2015). In-situ measurements of $MAC_{BC}$ have reported a wide range of values, from 3.8 m$^2$ g$^{-1}$ to 58 m$^2$ g$^{-1}$ (Wei et al., 2020). Although such high variability can be attributed, in part, to the determination method of the $MAC_{BC}$ based on $M_{BC}$ and absorption measurement techniques, differences in $MAC_{BC}$ values were found even for the same measurement technique.

Values of $MAC_{BC}$ depend on BC microphysical and chemical properties, which are related to their emission sources (Schwarz et al., 2008) and the effects of aging processes during the transport in the atmosphere (Ko et al., 2020; Laborde et al., 2013; Sedlacek et al., 2022). Freshly emitted BC is made of porous, fractal-like aggregates of nanoparticles (Beeler and Chakrabarty, 2022; China et al., 2013) that can become coated by condensation and/or coagulation with non-BC components (such as sulfate, nitrate, and organic components) during atmospheric aging (Fierce et al., 2020). Conversely this coating can be removed through evaporation and/or chemical processing via the production of more volatile substances (Sedlacek et al., 2022). Numerous studies have demonstrated that coating of BC with non-absorbing materials is accompanied by an enhancement of light absorption ($E_{abs}$) through the so-called lensing effect (Cappa et al., 2012; Denjean et al., 2020; Yuan et al., 2021; Healy et al., 2015; Liu et al., 2015; McMeeking et al., 2014; Peng et al., 2016; Van de Hulst, 1957; Xie et al., 2019; Schwarz

et al., 2006; Yus-Díez et al., 2022). However, most of these measurements were performed in the Planetary Boundary Layer (PBL) and over short periods from a few hours to as long as a season.

Both observations and model simulations pointed out an amplification of the warming rate by greenhouse gases and absorbing aerosols at high-mountain sites compared to PBL areas (Gao et al., 2018; Liu et al., 2009; Pepin et al., 2019; Rangwala, 2013). López-Moreno et al. (2014) have shown by running several regional climate models that the occurrence of winter warm events in the Spanish Pyrenees will gradually increase until 2080. This includes an increase in the number of warm days and nights and the number of snow/ice melting days at altitudes above 2000 m above sea level (asl). This so-called Elevation Dependent Warming (EDW) has been reviewed by the Mountain Research Initiative EDW Working Group, 2015, who listed the possible mechanisms behind this phenomenon (Pepin et al., 2015). Among the reasons given, BC is a potential driver of EDW by both absorbing solar radiation in the troposphere and decreasing the surface albedo when deposited on the cryosphere, thereby accelerating snow melt (Réveillet et al., 2022). In addition, BC has been found to have a higher radiative effect when it is located above clouds rather than near the surface (Samset and Myhre, 2015; Sanroma et al., 2010). All these findings highlight the importance of studying BC at high altitude mountain sites, where its effects on climate could be even more significant.

This study presents two-year continuous measurements of BC and aerosol properties conducted during the Hygroscopic properties of Black Carbon (h- BC) campaign at the high-altitude long-term monitoring station Pic du Midi (PDM). Located at 2877 m asl in the French Pyrenees, PDM has been early identified as a clean remote station (Marenco et al., 1994).

This paper aims to provide comprehensive picture of the seasonal and diurnal variability of rBC properties at PDM, and to explore the processes driving these properties. Specifically, the following questions are addressed:

1. What are the air mass transport pathways impacting PDM ?

2. What is the seasonal variability of aerosol optical properties and dominant aerosol types ? What is the specific contribution of rBC to aerosol absorption ?

3. How do the microphysical and optical properties of rBC vary on a seasonal and daily basis ?

4. What are the roles of wet deposition, source and transport pathway in driving rBC absorption ?

## 2  Methods

### 2.1  Measurement site and observation period

Measurements were performed at the Pic du Midi (PDM, 42.9° N, 0.1° E, 2877 m. asl) mountain research station in the French Pyrenees. This station is part of the Pyrenean Platform for Observation of the Atmosphere (P2OA)[1].

As shown in Fig. 1, the site is located 150 km east of the Atlantic coast. The high isolated summit lies around 20 km north of the main ridge of the Pyrenees (on the France-Spain border) and thus closely overlooks the French plain. Using a backward

---
[1]http://p2oa.aeris-data.fr

particle dispersion model, Henne et al. (2010) found the influence of local anthropogenic emissions to be very limited at PDM, and classified the station in the "mostly remote" category. Collaud Coen et al. (2018) defined an "ABL-Topoindex" as a metrics of the atmospheric boundary layer influence for a mountain site. PDM has been found to have a low ABL-Topoindex, similar to other Alpine high altitude stations. PDM is thus a suitable site to study both the background lower free troposphere (FT) over long timescales and injection of air masses from the PBL (Hulin et al., 2019; Tsamalis et al., 2014; Fu et al., 2016; Marusczak et al., 2017). Long-term monitoring of numerous meteorological, gas and aerosol parameters has been conducted for mostly two decades, notably through the Global Atmospheric Watch (GAW) program of the World Meteorological Organization (WMO), as well as the national research infrastructure ACTRIS-France. Results from the h-BC campaign performed from February 2019 to January 2021 at PDM (in addition to the routine measurements) are presented in this paper.

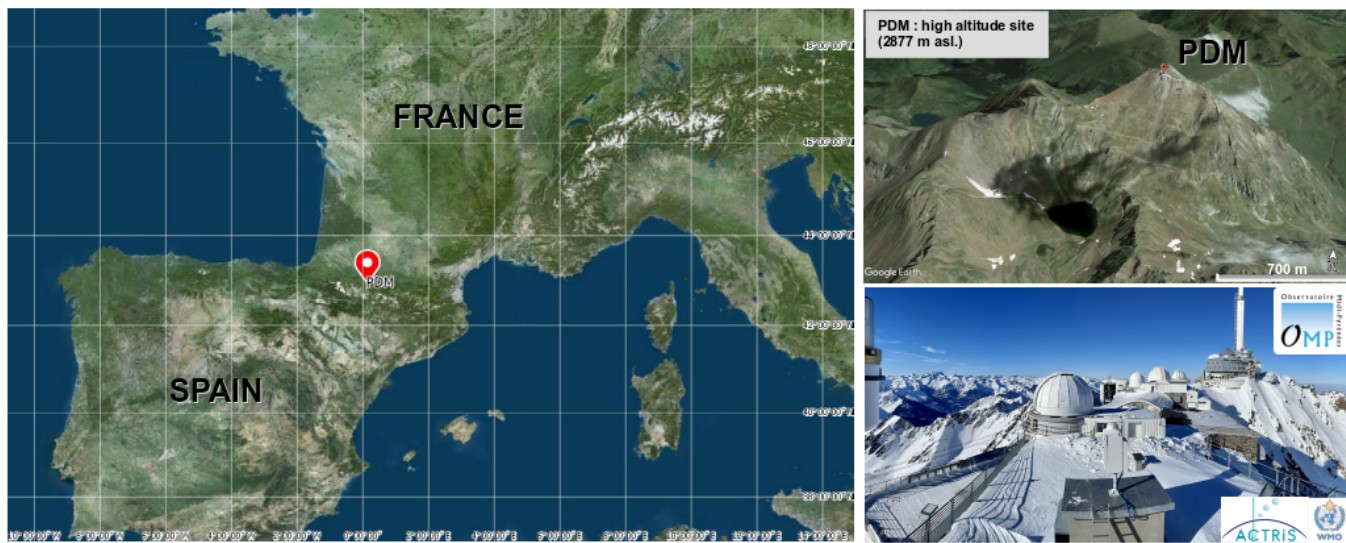

**Figure 1.** Geographical location of the Pic du Midi Observatory in the French Pyrenees (©IGN and ©Google Earth).

## 2.2 Instrumentation

### 2.2.1 Total inlet

All particle-measuring instruments sampled air taken in parallel from a whole air inlet, located 2 m above the building roof. This inlet is used for the long-term observations in mountainous sites and designed to maintain an isokinetic and laminar flow. The main flow rate was fixed at about $460 \, l \, min^{-1}$. The splitter was fixed at the end of the stainless tube. The hat of the whole air inlet and the stainless tube were both thermo-regulated in order to avoid frost and gradually regulate the temperature of the sampled air to the measurement room. The air was heated to around 20°C in order to perform aerosol in-situ measurements at a relative humidity lower than 30 %. The instrumental room temperature was regulated at around 20°C. The annual cycle of the dew point temperature varied between about -10° and +5°C.

### 2.2.2 Black carbon measurements

BC can be measured by different methods which are based on different BC properties. Petzold et al. (2013) defined a specific nomenclature for BC according to the BC quantification method. Following the recommendation of the authors, BC quantified by laser-induced incandescence, filter-based absorption and thermal-optical analysis will be referred to as refractory black carbon (rBC), equivalent black carbon (eBC) and elemental carbon (EC), respectively. More general discussion on BC without focusing on its measurement technique will be referred to as BC. The mass concentration of rBC and rBC size distribution were

measured by a single-particle soot photometer (SP2, Droplet Measurement Technology, Longmont, CO, USA). Its operating principles have been described in previous articles (Gao et al., 2007; Moteki and Kondo, 2007; Schwarz et al., 2006). In short, this instrument uses a laser-induced incandescence technique which quantifies rBC mass in single particles. A continuous intracavity laser beam (Nd:YAG; $\lambda$=1064 nm) is used to heat rBC-containing particles to their vaporization point. The measured incandescence signal of an individual rBC-containing particle can be converted to a rBC mass, using a calibration curve

obtained by recording the incandescence signal peak height of mobility size-selected fullerene soot (Alfa Aesar, lot #FS12S011) and assuming BC mass density of 1.8 g cm$^{-3}$ (Moteki and Kondo, 2010). This calibration was performed twice a year and did not change during the two-year of the measurement campaign.

The SP2 data were processed using a Python code following the method used in the SP2 Toolkit from the Paul Scherer Institute (Gysel et al., 2009). A comparison of $M_{rBC}$ resulting from the SP2 Toolkit with our Python processing is presented

in Text S1 in the Supplement. The SP2 used in this study measured rBC cores over a size range between 90 and 580 nm. However, the observed size distributions showed that an important fraction (around 12%) of $M_{rBC}$ at diameters below 90 nm is not measured by the SP2 (Fig. S2 in the Supplement). Because of these small-mode particles outside the SP2 detection range, the quantification of $M_{rBC}$ could be underestimated. To compensate the missing mass the observed rBC size distributions have been fitted daily using the sum of three individual lognormal distribution to extrapolate rBC size distribution in the range 1

to 1000 nm. The position of the three modes were constrained in the following ranges : Mode 1 : $50 < D_g < 100$ nm and $1.2 < \sigma_g < 3$; Mode 2 : $150 < D_g < 250$ nm and $1.3 < \sigma_g < 2.9$; Mode 3 : $350 < D_g < 500$ nm and $1 < \sigma_g < 3$, with $D_g$ and $\sigma_g$ the geometric mean diameter and the geometric standard deviation, respectively. Using the fitting procedure, a time-dependent missing mass correction was applied to the observed $M_{rBC}$ to calculate to overall $M_{rBC}$. The average missing mass correction factor over the campaign was $1.2 \pm 1.1$ (Mean value $\pm$ STD). More details on the SP2 data procedure can be found in the

Text S2 in the Supplement. The extent to which the uncertainty of this fitting procedure contributes to the overall $M_{rBC}$ was quantified by comparing the $M_{rBC}$ calculated from the observed BC size distribution and the fit curve over the SP2 size range. An excellent match was obtained between the measured and fitted size distribution, resulting in differences of less than 0.2 %. The combined uncertainty on the $M_{rBC}$ was estimated to be about 24.5 % by calculating the quadratic sum of the measurement uncertainties on sampling flow, anisokinetic sampling errors, and missing mass correction factor.

### 2.2.3 Aerosol properties

A Scanning Mobility Particle Sizer (SMPS), combining a differential mobility analyzer (DMA, model 3071, TSI Inc., Shoreview, USA) and a CPC (model 3772, TSI Inc., Shoreview, USA) allowed the determination of aerosols size distribution between 12.6 nm and 532.6 nm.

Aerosol scattering coefficients ($\sigma_{\text{sca}}$) at three wavelengths (450, 525 and 635 nm) were measured with an integrating nephelometer (model Aurora 3000, Ecotech Pty Ltd, Knoxfield, Australia). A calibration with carbon dioxide and filtered air was performed every three months. The instrument measures $\sigma_{\text{sca}}$ in the angular range 10-170°, and the correction of Müller et al. (2011) was used to account for the angular truncation errors.

Aerosol absorption coefficients ($\sigma_{\text{abs}}$) were measured by a seven-wavelength aethalometer (model AE33, Magee Scientific Company, Berkeley, USA, measuring wavelengths : 370, 470, 520, 590, 660, 880 and 950 nm). This instrument measures light attenuation through a filter on which aerosol sample is deposited. The aethalometer filter loading effect was corrected online by the dual-spot manufacturer correction proposed by Drinovec et al. (2015). The multiple scattering parameter used to correct the measured attenuation was set to 3.22, according to the value obtained at $\lambda$=880 nm by Yus-Díez et al. (2021) at the mountainous site of Montsec d'Ares located less than 200 km from the PDM (see Table S3 in the Supplements of Yus-Díez et al. (2021)). Uncertainty on the corrected $\sigma_{\text{abs}}$ was estimated to be 35 % (Zanatta et al., 2016). The detection limit of the aethalometer is 0.039 Mm$^{-1}$ (corresponding to an equivalent black carbon mass concentration of 0.005 $\mu$g m$^{-3}$).

### 2.2.4 Gas-phase measurements

Two different instruments have been deployed to measure carbon monoxide (CO) with a final time resolution of one hour: an IR-absorption analyser (model 48CTL, TEI Thermo Environment Instruments, New Delhi, India) placed close to the aerosol instrumentation in order to detect pollution plumes produced locally at PDM and a Cavity Ring Down Spectrometer (CRDS, model G2401, Picarro, Santa Clara, USA), located in an other building, used to measure the background carbon monoxide (CO) concentration (See Section 2.3).

A key issue in our study is the distinction between FT and PBL-influenced air masses. Optical properties of BC depend on its aging and transport pathways in the atmosphere, so it is crucial to determine whether it has been transported over the PBL or in the FT. For this purpose, we routinely monitor the diurnal cycle of radon ($^{222}$Rn) volumetric activity at PDM with a 1500-L high-sensitivity radon monitor (model D1500, ANSTO Australian Nuclear Science and Technology Organisation, Australia) (Whittlestone and Zahorowski, 1998). Radon is an inert radioactive gas emitted from ice-free soils with a half-life of 3.8 days, making it the most reliable tracer to discriminate between the FT and PBL- influenced air masses (Chambers et al., 2013).

### 2.3 Determination of intensive aerosol and BC properties

The spectral dependence of $\sigma_{\text{abs}}$ was characterized by the Absorption Ångström Exponent (AAE$_{\text{aer,450-635}}$) calculated between 450 and 635 nm as follows :

$$AAE_{aer,450-635} = \frac{-log\left(\frac{\sigma_{abs,450}}{\sigma_{abs,635}}\right)}{\left(log(\frac{450}{635})\right)} \qquad (1)$$

For this calculation, $\sigma_{abs,470}$ and $\sigma_{abs,660}$ from the aethalometer were adjusted at the wavelengths of 450 and 635 nm measured by the nephelometer using the AAE calculated from the aethalometer between 370-470 nm and 590-660 nm. $AAE_{aer,450-635}$ provides information about the chemical composition of atmospheric aerosols. Pure BC absorbs radiation across the whole solar spectrum with the same efficiency; thus, it is characterized by $AAE_{aer,450-635}$ around 1 (Bond et al., 2013). Conversely light-absorbing organic particles known as brown carbon (BrC), as well as dust particles generally have an $AAE_{aer,450-635}$ greater than 2 (Sun et al., 2007; Bergstrom et al., 2007; Schuster et al., 2016).

The wavelength dependence of $\sigma_{sca}$ can be characterized by the Scattering Ångström Exponent ($SAE_{aer,450-635}$) calculated between 450 and 635 nm, as :

$$SAE_{aer,450-635} = \frac{-log\left(\frac{\sigma_{sca,450}}{\sigma_{sca,635}}\right)}{\left(log(\frac{450}{635})\right)} \qquad (2)$$

$SAE_{aer,450-635}$ describes the relative contribution of fine and coarse mode particles (Clarke and Kapustin, 2010). Small values of $SAE_{aer,450-635}$ indicate a higher contribution of large aerosol particles (e.g. dust and sea salt), while large values of $SAE_{aer,450-635}$ indicate relatively smaller aerosol particles (Cappa et al., 2016).

The aerosol Single Scattering Albedo ($SSA_{aer,\lambda}$) was calculated at the wavelengths of $\lambda$ = 450, 525 and 635 nm using the following equation :

$$SSA_{aer,\lambda} = \frac{\sigma_{sca,\lambda}}{\sigma_{sca,\lambda} + \sigma_{abs,\lambda}} \qquad (3)$$

$SSA_{aer,\lambda}$ describes the relative importance of scattering and absorption to the total light extinction. Thus, it indicates the potential of aerosols to cool or warm the atmosphere. To calculate $SSA_{aer,\lambda}$, $\sigma_{abs}$ was first calculated at the proper wavelength $\lambda$ using the AAE calculated at the closest wavelengths ($AAE_{aer,370-470}$ to retrieve $\sigma_{abs,450}$, $AAE_{aer,520-590}$ for the $\sigma_{abs,525}$, and $AAE_{aer,590-660}$ for the $\sigma_{abs,635}$).

The $\Delta M_{rBC}/\Delta CO$ emission ratio was calculated to provide information on the combustion sources, as well as on BC wet deposition (Baumgardner et al., 2002). First, the background CO concentrations were estimated by taking the rolling 5[th] percentile of the values on a 14-day time window and then calculating a monthly mean (see fig S3 in the Supplement) based on the method of Kanaya et al. (2016). $\Delta CO$ was then calculated by subtracting the monthly background CO concentration to any measured hourly CO value. $\Delta M_{rBC}$ was considered to be equal to $M_{rBC}$, because we assume that the background BC is zero since the atmospheric lifetime of BC is known to be of a few days (Park et al., 2005). By contrast CO lifetime is estimated at several days (Bey et al., 2001).

The Mass Absorption cross-section of rBC ($MAC_{rBC}$) was determined as :

$$MAC_{rBC} = \frac{\sigma_{abs,880}}{M_{rBC}} \tag{4}$$

$M_{rBC}$ below the 5[th] and above the 95[th] percentile were filtered before $MAC_{rBC}$ calculations to reduce the influence of outliers in statistical analyses. In addition, we filtered out periods when dust were sampled at PDM for the calculation of $MAC_{rBC}$ since Yus-Díez et al. (2021) observed significant biases in the multiple scattering correction of the aethalometer AE33 during such events.

The light-absorption enhancement factor $E_{abs}$ can be determined as the $MAC_{rBC}$ value normalized by a reference value for
pure, uncoated (bare) rBC:

$$E_{abs} = \frac{MAC_{rBC}}{MAC_{bare,rBC}} \tag{5}$$

Three different methods are generally used to estimate $MAC_{bare,rBC}$: the first one is to remove the coating of BC with a thermodenuder and measure the corresponding absorption (Cappa et al., 2012; Healy et al., 2015; Yuan et al., 2021); the second one is to extrapolate measurements of $MAC_{rBC}$ as a function of the measured rBC mixing ratio (Cappa et al., 2019;
Yuan et al., 2021); and the third one consists in calculating $MAC_{bare,rBC}$ from the measured rBC size distribution using Mie's theory and the mean geometric rBC diameter (Zanatta et al., 2018; Liu et al., 2017, see fig. S4 in the Supplement). Here we used the latter method by assuming a rBC refractive index of 1.95 - 0.79i at $\lambda$=880 nm (Bond and Bergstrom, 2006). The calculation of $MAC_{bare,rBC}$ using Mie's theory assume a simplified spherical assumption of rBC morphology. However rBC may exhibit complex morphologies whose optical behavior is imperfectly predicted by Mie's theory, introducing a bias in the retrieved
$MAC_{bare,rBC}$ (Saleh et al., 2016). It might be considered that Mie's theory is suitable for estimating the absorption of highly aged rBC, which exhibit an internally mixed core-shell structure. China et al. (2015) used this method to calculate the $E_{abs}$ of rBC in a high-altitude site of the Azores Islands because the large majority (70%) of these long-range transported particles were found highly compacted. Several studies found that Mie's scattering model captures basic optical properties of BC in biomass burning plumes (Liu et al., 2017; Denjean et al., 2020). Zanatta et al. (2018) calculated $MAC_{rBC}$ of heavily coated
rBC particles from the Arctic region using Mie's theory and found consistent results with direct measurements. In addition to the morphology, the $MAC_{bare,rBC}$ calculation is also very sensitive to the refractive index of rBC core (Sorensen et al., 2018). Liu et al. (2020b) summarized the changes in MAC values induced by the use of different refractive indexes. They reported deviations from -7 % to -35 % to the $MAC_{BC}$ value of 7.5 m$^2$ g$^{-1}$ recommended by Bond and Bergstrom (2006).

Time periods with high humidity (95%) or precipitation were filtered before analysis to avoid artifacts in the sampling inlet.
Under precipitation some water droplet may indeed enter in the aerosol inlet and change both the inlet cut off diameter and the measured aerosol size distribution. This would bias all the measured aerosol properties. We also filtered periods where hourly CO concentrations exceeded 200 ppb in order to exclude local pollution events, e.g. due to snow removal of the touristic platform.

All aerosol and gas measurements were converted to standard temperature and pressure (273.15 K and 1013.25 hPa).

## 2.4 Identification of air mass origins

The Hybrid Single Particle Lagrangian Integrated Trajectory Model (HYSPLIT) (Draxler and Hess, 1997) was used to calculate air masses backtrajectories. This model uses 3-hourly atmospheric data from the Global Data Assimilation System (GDAS) of the National Center for Environmental Prediction (NCEP) in a 1°×1° spatial resolution. More information can be found on https://www.ready.noaa.gov/index.php. A backtrajectory was run at 12:00 UTC for each day, going back 72 hours, for the two years of the campaign. Precipitation rates along the back trajectories were also computed from the HYSPLIT calculations, in order to classify days when the air masses arriving at PDM encountered precipitation or not in the past 72 hours.

To discriminate FT and PBL-influenced air masses (hereafter referred as PBL/FT conditions), we followed a methodology proposed initially by Griffiths et al. (2014), assuming that the diurnal radon increase, which is typically observed at mountain sites during the daytime, is the result of transport of PBL air by thermal anabatic winds up to the summits. The method first consists in ranking the days of the sampling period by decreasing anabatic influence (details in the Supplement and in Griffiths et al. (2014)). A threshold rank (here 282, see Fig. S5 and associated text in the Supplement) can then be determined to separate days with or without anabatic influence in the daytime.

In our study, it was necessary to select the observation hours strongly influenced by the boundary layer. To do this, we selected the first 200 anabatic days in the ranking, and from these days, all the hours when radon activity was greater than the median value for the current day.

We also needed an ensemble of observation hours with minimum influence of the PBL. In the latter case, we selected hours in the non-anabatic days (i.e. ranked after 282) with radon activity below the median value for the current day.

## 3 Results

### 3.1 Meteorology and air mass classification

In the following, seasons are defined as follows: winter (December, January, February), spring (March, April, May), summer (June, July, August), and autumn (September, October, November). The meteorological conditions at PDM during the campaign are described in Figure S6 and associated text in the Supplement.

The backward trajectories performed with the HYSPLIT model on 72-hour time periods of time were assigned to six geographical zones, according to the position of their starting point (shown in Fig. 2): North-west Europe, Continental Europe, Med-Africa, Atlantic Spain, North Atlantic and Local (within a circular zone of 100 km radius around PDM). The transport to PDM was generally from the west or south, ie. from the North Atlantic and Atlantic Spain zones. It should also be noted that 99% of all atmospheric backward trajectories modelled to PDM reveal long-range transport (>100 km).

The analyses of the diurnal cycle of radon concentrations allowed to determine the FT and PBL conditions prevailing at the site following the methodology presented in Section 2.4. Details on the statistical results can be found in Table S1 in the Supplement. Over the campaign, 1149 hours were clearly identified as FT-influenced conditions, which represents 56% of the total classified hours. In winter, FT and PBL conditions occurred roughly 74% and 26% of the analyzed time, respectively,

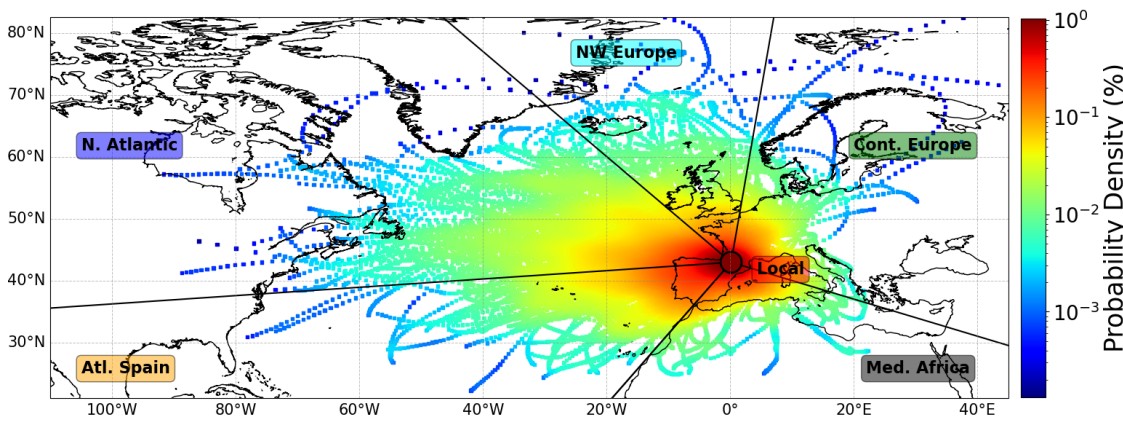

**Figure 2.** 72-hour Back trajectories of air masses measured at PDM over the measurement period 2019-2020. Geographical boundaries of the sectors used to classify the air mass back-trajectories are overlaid.

against 48% and 52% for summer, respectively. These results are broadly in agreement with the previous study by Hulin et al. (2019) at PDM, which quantified around 47 % of the days as PBL-influenced over a 10 years period. The PBL conditions occurred mostly around 15:00 UTC (see Fig. S5 in the Supplement), consistent with the dynamics at mountain sites where

plain-to-mountain winds and along-valley winds become the strongest in the afternoon (Whiteman, 2000).

### 3.2 Aerosol optical properties and classification of aerosol types

Figure 3 presents daily time series and statistics of aerosol optical properties over the two-year measurement period. The average SSA $\pm$ GSD (Geometric Standard Deviation) were 0.94 $\pm$ 0.06, 0.94 $\pm$ 0.07 and 0.95 $\pm$ 0.08 at 450, 525 and 635 nm, respectively (Fig. 3a). These values are in the range of those observed at mountain sites in Southern Europe (Bukowiecki et al.,

2016; Laj et al., 2020; Pandolfi et al., 2014). The mean value $\pm$ GSD of $\sigma_{\mathrm{abs,880}}$ was 0.27 $\pm$ 0.25 Mm$^{-1}$ (Fig. 3b), which falls in the range of 0.14 to 1.23 Mm$^{-1}$ obtained at Jungfraujoch and Montsec (Bukowiecki et al., 2016; Pandolfi et al., 2014). The average $\sigma_{\mathrm{sca}}$ $\pm$ GSD were 15.5 $\pm$ 16.1 Mm$^{-1}$, 13.4 $\pm$ 13.9 Mm$^{-1}$ and 12.2 $\pm$ 12.9 Mm$^{-1}$ at 450, 525 and 635 nm, respectively (Fig. 3c). These weak values of $\sigma_{\mathrm{abs}}$ and $\sigma_{\mathrm{sca}}$ can be explained by the remote mountain site type, where almost no aerosols are locally emitted. There was a clear seasonality of aerosol optical properties. SSA at the three wavelengths exhibited the lowest

monthly mean values in spring-summer (0.94 $\pm$ 0.02 at $\lambda$ = 525 nm) and the highest in autumn-winter (0.99 $\pm$ 0.01 at $\lambda$ = 525 nm), as shown in Fig. 3a). Simultaneously, the highest monthly mean SAE values were observed in spring-summer (1.23 $\pm$ 0.70 ) and reached a minimum in the winter (-0.25 $\pm$ 0.16) (Fig. 3d). This anticorrelation suggests a higher fraction of absorbing and fine particles relative to purely scattering and coarse particles at PDM during the spring-summer. Interestingly different trends can be observed between the summer and spring seasons. During spring 2019 the decrease of SSA is correlated with a slight

enhancement of $\sigma_{\mathrm{abs,880}}$ (Fig. 3b) and a decrease of $\sigma_{\mathrm{sca}}$ at all wavelengths. In summer the increase of $\sigma_{\mathrm{abs,880}}$ lead to values multiplied by a factor of four, while both SSA and SAE remained rather constant. All these parameters combined indicate a

similar dominant aerosol type reaching PDM but with stronger contribution in summer. This noteworthy seasonality of aerosol optical properties has previously been observed at other high mountain sites in Europe (Andrews et al., 2011; Collaud Coen et al., 2011; Laj et al., 2020; Pandolfi et al., 2018). The higher concentration of small and absorbing particles in summer at PDM could be attributed to a higher anthropogenic BC influence favored by strong vertical mixing and a higher PBL height, a higher occurrence of wildfires emitting large amounts of BC and Brown Carbon (BrC), or a lower precipitation rate.

In order to investigate these different hypotheses, a classification of the dominant aerosol type sampled at PDM was performed by using the spectral dependency of aerosol optical properties. Figure 4 shows AAE as a function of SAE, overlaid with the aerosol classification matrix from Cappa et al. (2016). Aerosols with the highest SSA values (violet points) tend to fall on the left-hand side of the plot with SAE values below 0, indicative of large particles such as marine sea salt, continental dust or highly processed/coated particles. The presence of large marine and dust aerosols is in line with backward trajectories showing a dominant origin of air masses coming from the Atlantic Ocean and Iberian Peninsula as far as North Africa (e.g. Fig. 2). Dust being a strong light absorber, it is expected to lower the mean aerosol SSA. However Fig. 4 shows that SSA for dust-dominated aerosol (classified as having AAE values above 2) are quite similar as those for remote marine aerosol (classified as having AAE values below 1). Although Europe frequently experiences African dust events (Denjean et al., 2016; Dumont et al., 2023), our results indicate that these dust events were not absorbing enough to substantially lower the aerosol SSA at PDM. This is supported by previous estimates of SSA ranging between 0.90 and 1.00 for dust particles transported in the Mediterranean region (Mallet et al., 2013; Denjean et al., 2016).

There was a natural clustering of the most light absorbing aerosols with SSA < 0.9 (pink to yellow points) on the middle of the plot, with sections on the lower side with AAE between 0.5 and 1.5, which Cappa et al. (2016) defined as the sections dominated by BC or a mix of BC and large particles. The success of aerosol classification schemes is largely dependent on uncertainties in AAE attribution for each aerosol species. Although AAE = 1 is often considered to be BC such as that in the classification by Cappa et al. (2016), observational and numerical estimates show a wide range of BC AAE from 0.6 to 1.3 (Kirchstetter et al., 2004; Liu et al., 2018) due to the variation of BC core size, coating thickness, composition and morphology (Liu et al., 2018; Zhang et al., 2018). Therefore it is possible that the large range of AAE values observed for the most light-absorbing aerosols were due to different microphysical and chemical properties of the BC sampled at PDM.

Interestingly almost none of the aerosols were classified as strong BrC and BC/BrC mixture (AAE and SAE values above 1.5), revealing a very low contribution of BrC to the aerosol absorption at PDM. An explanation could be the rapid BrC depletion within the first day after emission, by photobleaching or volatilization that has been observed in several studies (Forrister et al., 2015; Wong et al., 2019; Zeng et al., 2020). Altogether these results suggest that BC was the predominant absorption component of aerosols at PDM and controlled the variation of SSA throughout the two observation years.

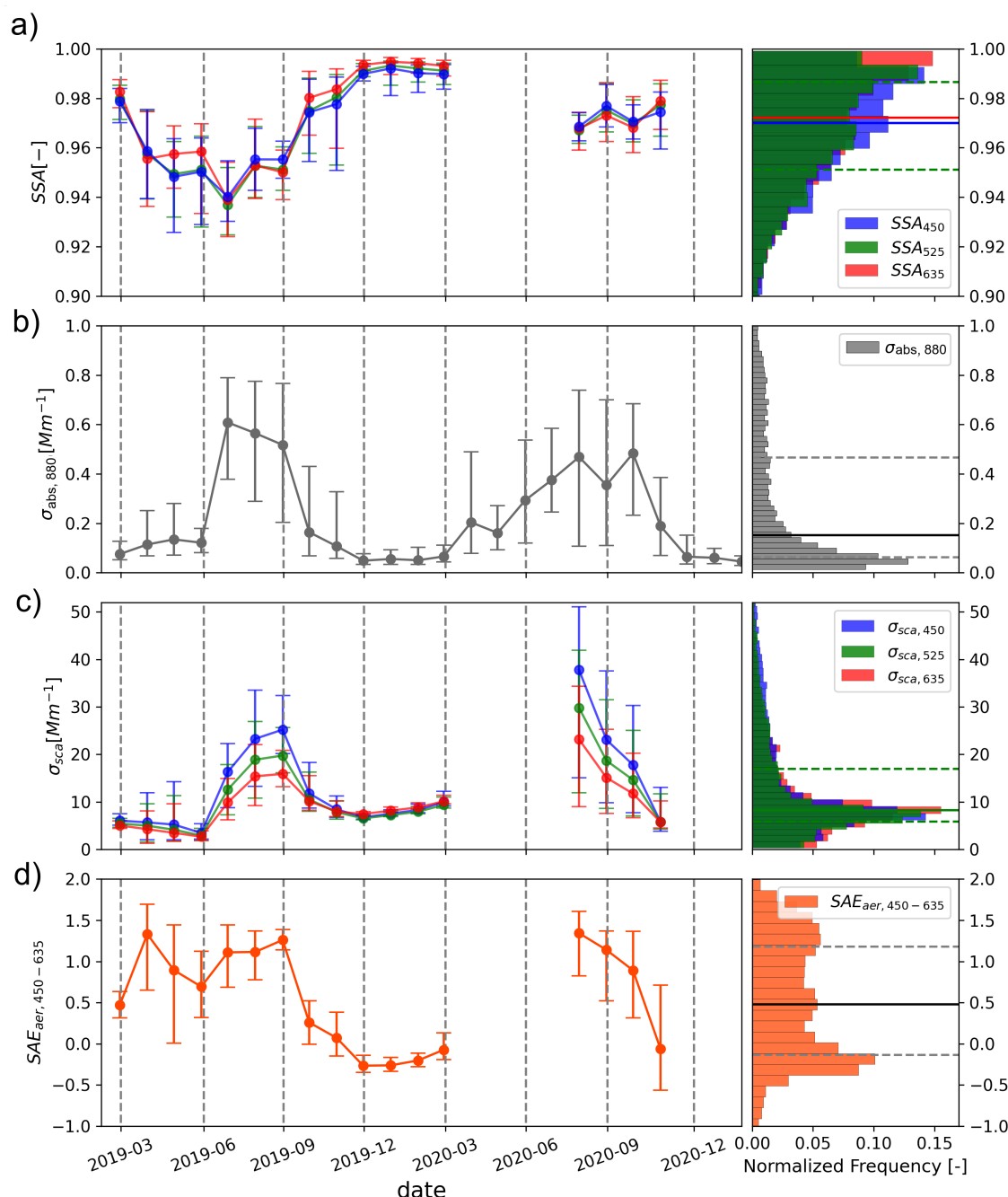

**Figure 3.** Time series (left) and statistical distributions (median, 25[th] and 75[th] percentiles, right) of aerosols optical properties measured at PDM in 2019-2020 with (a) Single Scattering Albedo at 450, 525 and 635 nm, (b) Absorption coefficient at 880 nm, (c) Scattering coefficients at 450, 525 and 635 nm and (d) Scattering Angström Exponent at 450-635 nm. The dots and bars on the time series represent the median, the 25[th] and 75[th] percentiles, respectively, with a monthly frequency. Histograms was computed using a 1-day time frequency. Vertical dashed lines represent the seasons boundaries.

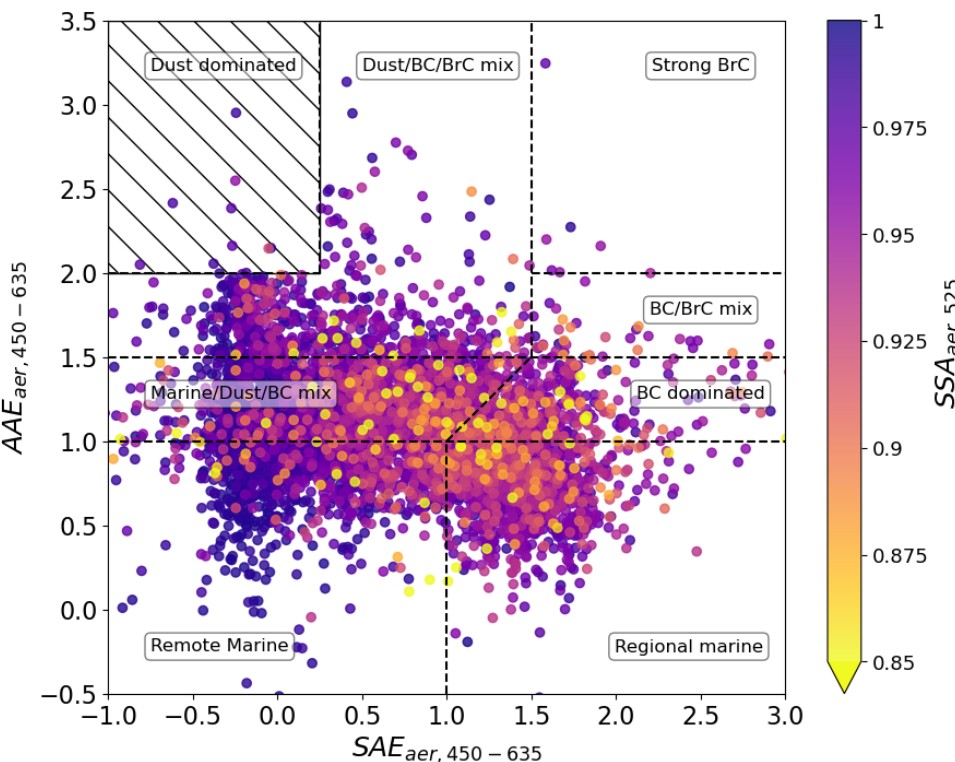

**Figure 4.** Hourly average Aerosols Absorption Angström Exponent vs. Scattering Angström Exponent calculated at 450-635 nm and colored as a function of the Single Scattering Albedo at 525 nm. The classification of aerosol type by Cappa et al. (2016) is also shown. The points in the dashed zone, representing dust events, were filtered before analyses of BC properties to avoid artifacts in the calculation of MAC$_{rBC}$.

### 3.3 rBC sources and properties

#### 3.3.1 rBC mass concentration

Figure 5 shows the time series of the microphysical and optical properties of rBC. The mean M$_{rBC}$, shown in Fig. 5a, was 34.8

± 35.7 (Mean ± STD) ng m$^{-3}$, which is a typical level for remote mountain sites. For instance, Sun et al. (2021) observed M$_{eBC}$ around 20 ng m$^{-3}$ from 9-years of measurements with a Multi-Angle Absorption Photometers (MAAP, model 5012, Thermo Scientific) at the Zugspitze-Schneefernerhaus station, Germany (2671 m asl). Motos et al. (2020) measured M$_{rBC}$ around 9 ng m$^{-3}$ in summer at Jungfraujoch, Switzerland (3580 m asl). rBC represented around 7 ± 5 % of the total aerosol number concentration measured by the SMPS over the campaign. An increase of rBC number fraction by a factor 2.5 was

found in summer (9 ± 5 %) compared to winter (4 ± 3 %). Simultaneously, $\sigma_{abs,880}$ increased by a factor 4 between winter and summer. Thus, it confirms that rBC contributed to a significant part of the aerosol absorption at PDM.

### 3.3.2 rBC emission sources

Figure 6 shows bivariate polar plots obtained by combining wind analysis and $M_{rBC}$ with 1-hour time resolution in winter and summer. The densities of $M_{rBC}$ data weighted by $M_{rBC}$ values, and normalized by the maximum density of $M_{rBC}$ were plotted as a function of wind direction and speed. The darkest areas of the wind pattern are those where the highest $M_{rBC}$ was measured with a high occurrence, whereas lightest zones exhibit lowest measured $M_{rBC}$ and/or a little occurrence of measurements. Note that locally emitted pollution at the measurement station was filtered before the analysis, limiting local $M_{rBC}$ contributions emitted from the PDM station (i.e. section 2.3).

In summer, the highest $M_{rBC}$ values were mainly associated with moderate wind speeds (above 5 m s$^{-1}$) and from the west and south west, suggesting a dominant regional transport (Fig. 6a). By contrast in winter (Fig. 6b), the highest $M_{rBC}$ occurred mainly under more static atmospheric conditions (ie. for wind speeds below 5 m s$^{-1}$) and no evident wind direction dependency. These results suggest that local-scale emissions could be a major contributor to $M_{rBC}$ in winter unlike summer. Further discussion on the role of PBL influence on $M_{rBC}$ will be addressed in Section 3.4.2.

The $\Delta M_{rBC}/\Delta CO$ emission ratio, presented in Fig. 5b, shows a wide range of values from 0 to 10 ng m$^{-3}$ ppbv$^{-1}$, with a mean value of $1.93 \pm 2.12$ ng m$^{-3}$ ppbv$^{-1}$. Summer ratios were generally higher than winter emission ratios, which could reflect either lower rBC scavenging during transport or different emission sources of rBC between seasons. Indeed, $\Delta M_{rBC}/\Delta CO$ emission ratio varies as a function of fuel types, combustion efficiencies and wet deposition by precipitation (Baumgardner et al., 2002; Taylor et al., 2014). This explains the high diversity of $\Delta M_{rBC}/\Delta CO$ emission ratios obtained in the literature worldwide, going from 0.5 ng m$^{-3}$ ppbv$^{-1}$ at Jungfraujoch, Switzerland (Liu et al., 2010), to 9 ng m$^{-3}$ ppbv$^{-1}$ in a biomass burning plume above Texas region during TexAQS 2006 campaign, USA (Spackman et al., 2010), if only studies using SP2 measurements are considered. Overall $\Delta M_{rBC}/\Delta CO$ from fossil fuel tends to exhibit lower values than those from biomass combustion (Guo et al., 2017; Pan et al., 2011; Zhu et al., 2019). To our knowledge the only available $\Delta M_{rBC}/\Delta CO$ measurements in Europe were performed during airborne measurements in the Cabauw industrial region, Netherlands, by McMeeking et al. (2010), who found very low values around 0.8 ng m$^{-3}$ ppbv$^{-1}$. The high time variability of $\Delta M_{rBC}/\Delta CO$ reflects important differences concerning the scavenging processes impacting BC and/or the relative contribution of biomass burning and fossil fuel emissions in the production of BC measured at PDM. These two different factors will be addressed in Section 3.4.

### 3.3.3 rBC mass size distribution

rBC mass median core size diameter ($D_{rBC,core}$) was quite constant during the campaign with a mean geometric diameter of $179 \pm 28$ nm (Fig. 5c). An exception occurred in early December 2019, where we detected the presence of large rBC particles with $D_{rBC,core}$ around 400 nm. However observations during this period may be the results of measurement uncertainties due to too low $M_{rBC}$ (less than 10 ng m$^{-3}$). The $D_{rBC,core}$ values obtained at PDM are generally comparable to $D_{rBC,core}$ that has been reported ranging from 180 to 225 nm for well-aged background rBC (Liu et al., 2010; McMeeking et al., 2010; Schwarz et al., 2010; Shiraiwa et al., 2008). However, our values are slightly higher than previous observations at Jungfraujoch by Motos et al. (2020) who reported $D_{rBC,core}$ ranging from 130 and 150 nm in summer and winter. Instead of fitting the SP2 observations with

a multimodal individual lognormal modes (e.g. Section 2.2.2), Motos et al. (2020) only took into account rBC particles with $D_{rBC,core} > 50$ nm (the lower detection limit of their SP2 version), which may bias the estimated SP2 mode of $D_{rBC,core}$ (Tinorua et al., in preparation).

### 3.3.4   rBC absorption

The ambient $MAC_{rBC}$ was around $9.2 \pm 3.7$ $m^2$ $g^{-1}$ at $\lambda = 880$ nm (Fig. 5d). Several studies previously reported $MAC_{BC}$ values between 8.9 and 13.1 $m^2$ $g^{-1}$ for measurements at $\lambda = 637$ nm in European mountain stations (Pandolfi et al., 2014; Yus-Díez et al., 2022; Zanatta et al., 2016). By using a AAE of unity, these values can be converted to $MAC_{BC}$ between 6.4 and 9.5 $m^2$ $g^{-1}$ at $\lambda = 880$ nm. These studies used different measurement techniques, analysis method and correction factors from ours for estimating $MAC_{BC}$ that makes difficult the comparison of $MAC_{rBC}$ derived from different instruments. Pandolfi et al. (2014)

performed a linear regression between $\sigma_{abs,637}$ measured by a MAAP (Multi-Angle Absorption Photometer) and daily $M_{EC}$ values from off-line filter-based measurements by a SUNSET OCEC Analyser. Yus-Díez et al. (2022) and Zanatta et al. (2016) retrieved $MAC_{BC}$ with these instruments by calculating the ratio between the two parameters instead of a linear regression. Because of the absence of a standard method for quantifying $M_{BC}$, the absolute uncertainties on the $MAC_{BC}$ obtained in the literature are very high ranging from $\pm 29$ to $\pm 63\%$ (Zanatta et al., 2016).

In terms of seasonality we found systematically higher values of $MAC_{rBC}$ in summer (monthly mean $\pm$ STD of $10.3 \pm 3.3$ $m^2$ $g^{-1}$) compared to winter ($8.3 \pm 3.8$ $m^2$ $g^{-1}$). Similar seasonal pattern was observed in Europe at Puy de Dôme (Zanatta et al., 2016, central France) and at Jungfraujoch (Swiss Alps) mountain sites (Motos et al., 2020). An opposite trend was observed at mountain sites affected by strong precipitation during monsoon such as the Tibetan Plateau and Himalayas regions where both $MAC_{BC}$ and $M_{BC}$ exhibit maximum values in winter or autumn (Zhao et al., 2017; Srivastava et al., 2022). The

same seasonal pattern with elevated values in winter/autumn compared to summer/spring was observed at several rural and urban sites in the PBL, which was attributed to greater emissions from residential heating combined to a lower PBL height (Zanatta et al., 2016; Kanaya et al., 2016; Yttri et al., 2007). However maximum of $MAC_{BC}$ and $M_{BC}$ in the PBL during cold periods is not a recurring observation even for a same measurement site. For instance Sun et al. (2022) showed that in Beijing (China), due to the reduction of some predominant BC sources in winter consecutive to environmental policies, the annual

cycle of $M_{BC}$ changed over the years between 2012 and 2020.

   Variations in $MAC_{rBC}$ may exist for different reasons. We first addressed the question of whether the $MAC_{rBC}$ depends on $D_{rBC,core}$ in Fig. S7 in the Supplement. There was no clear correlation between $MAC_{rBC}$ and $D_{rBC,core}$, which indicates that the variation in BC size was not the cause of the $MAC_{rBC}$ variability. This is because $D_{rBC,core}$ only varied within a relatively narrow range (the 25th and 75th percentiles around 164 and 195 nm) during the campaign. The observed $MAC_{rBC}$ values were converted

to equivalent $E_{abs}$ by dividing them by a reference MAC for pure uncoated BC ($MAC_{bare,BC}$). While values of $MAC_{bare,BC}$ are reported in the literature, estimation of campaign-specific $MAC_{bare,BC}$ allows for more robust determination of $E_{abs}$ than using values from the literature since $MAC_{bare,BC}$ is dependent of the size of uncoated BC (Bond and Bergstrom, 2006; Adachi et al., 2007, 2010; Adachi and Buseck, 2013; Cappa et al., 2012). Here the mean $MAC_{bare,BC}$ was 4.15 $m^2$ $g^{-1}$ with values ranging

from 3.90 to 4.37 m$^2$ g$^{-1}$ considering the standard deviation of the mean $D_{rBC,core}$, which is in reasonable agreement with literature assessments (Liu et al., 2020b).

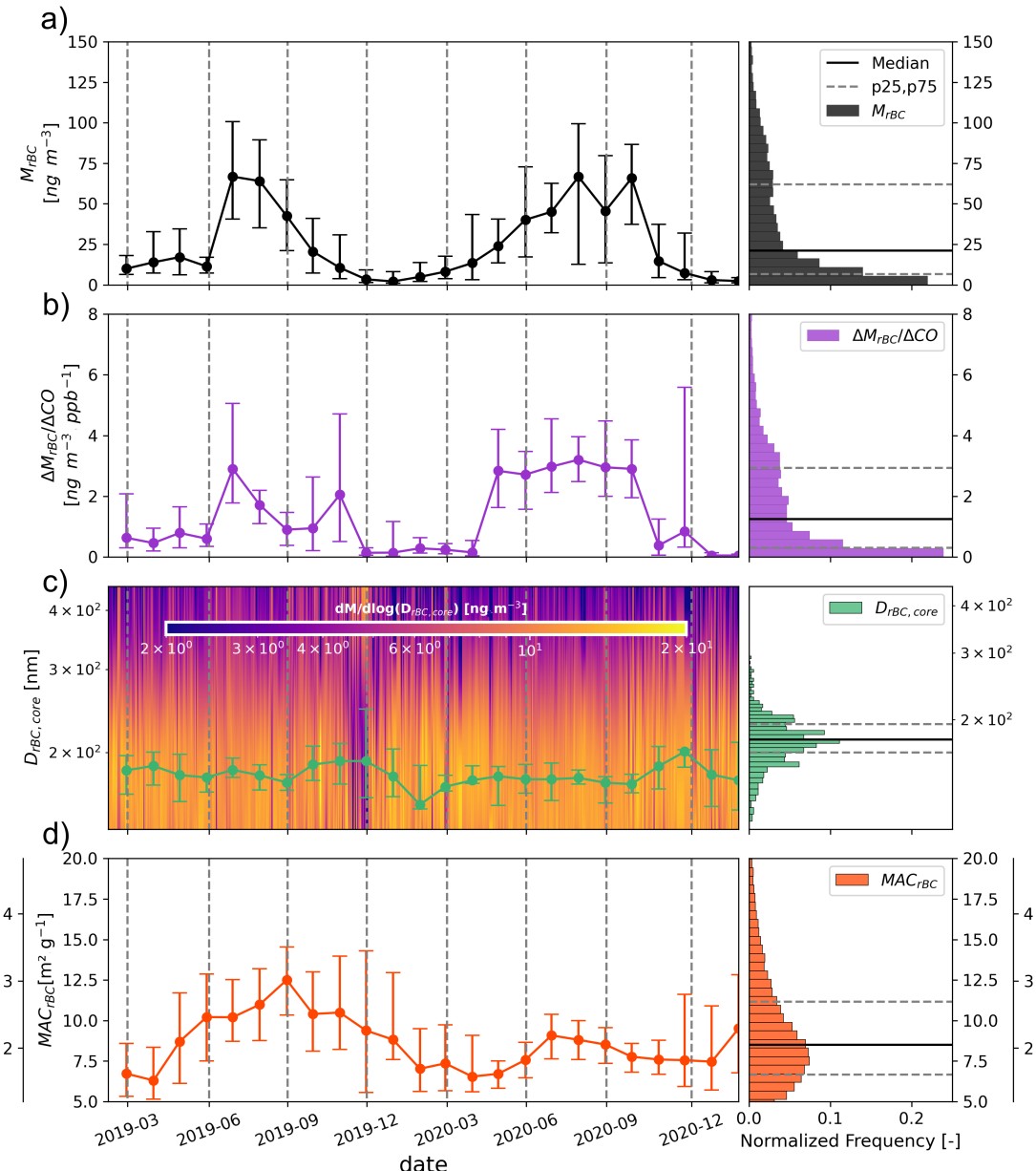

**Figure 5.** Time series (left) and statistical distributions (median, 25[th] and 75[th] percentiles, right) of rBC properties measured at PDM in 2019-2020. (a) rBC mass concentration, (b) $\Delta M_{rBC}/\Delta CO$ emission ratio, (c) rBC core mass size distribution with geometric diameter in green solid line, (d) rBC Mass Absorption Cross-Section and Absorption Enhancement at 880 nm. The dots and bars on the time series represent the median, the 25[th] and 75[th] percentiles, respectively, with a monthly frequency. Histograms was computed using a 1-day time frequency, as well as the colored background of the rBC core size distribution. Vertical dashed lines represent the seasons boundaries.

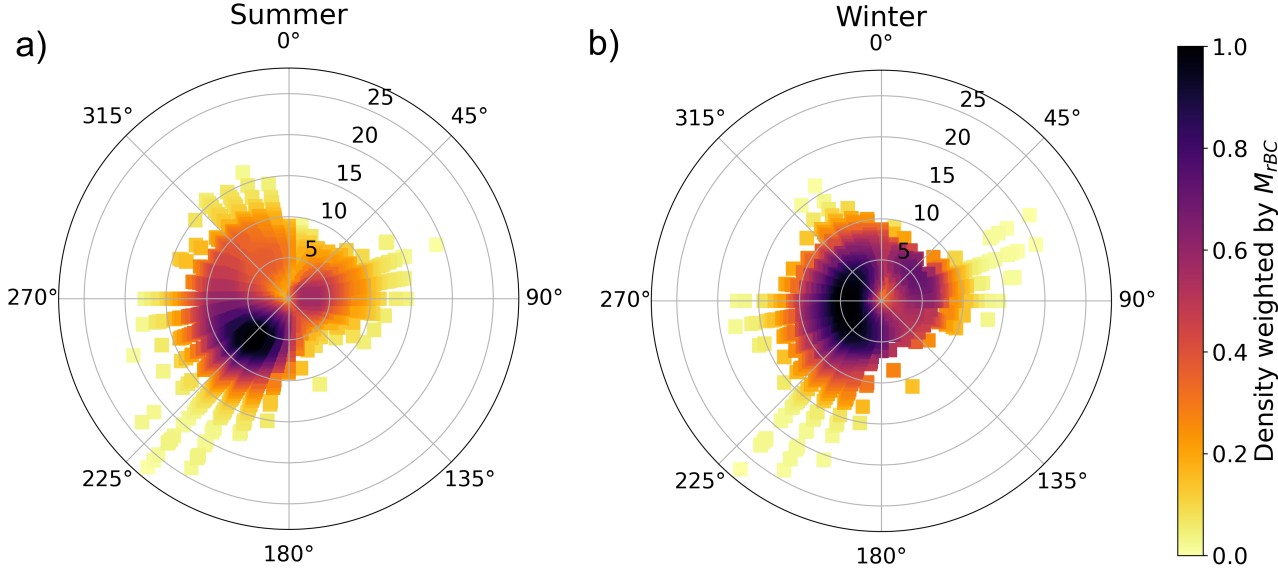

**Figure 6.** Bivariate polar plots of the density of $M_{rBC}$, weighted by $M_{rBC}$ values as a function of wind direction and speed in a) summer and b) winter. The color scale shows the $M_{rBC}$ density data weighted by $M_{rBC}$. The radial scale shows the wind speed, which increases from the centre of the plot radially outwards. Both plots use hourly data. The weighted densities was normalised by their maxima.

The $E_{abs}$ values relative to $MAC_{bare,BC} = 4.15$ m$^2$ g$^{-1}$ at $\lambda = 880$ nm were significantly greater than unity with mean value of $2.2 \pm 0.9$ (Fig. 5d). Given the remote mountain location and presumable distance from fresh rBC sources, rBC particles reaching PDM may have undergone aging and have gained a consistent coating. Previous studies found an absorption enhancement of BC due to its coating with the aging time (Yus-Díez et al., 2022; Sedlacek et al., 2022; Peng et al., 2016). The most likely cause of the strong $E_{abs}$ at PDM is a lensing effect due to the internal mixing of rBC with other particles that drives $MAC_{rBC}$ variability, though we cannot eliminate changes in rBC morphology that can result from coating onto rBC. There was a significant seasonal trend in $E_{abs}$ with higher values observed in summer, indicating that rBC reaching the PDM station has undergone longer aging processes during this season. These results are consistent with the measurements of Motos et al. (2020) at Jungfraujoch, which also indicated a strong seasonality in rBC mixing state with larger coating in summer.

Figure 7 further shows the diurnal variation of $E_{abs}$ for every season. There was a notable opposite diurnal profile between seasons in $E_{abs}$ with midday showing a minimum around 1.7 in winter, and a maximum around 2.9 in summer. Spring and autumn showed intermediate patterns with less regular $E_{abs}$ throughout the day. These observations suggest that different sources and/or processes drove the seasonal contrast in rBC properties. Nonetheless, it cannot be ruled out that some of the variability in $MAC_{rBC}$ (and $E_{abs}$) can be explained by the use of a fixed multiple scattering correction applied to calculate $\sigma_{abs}$, since this parameter has been shown to vary as a function of time and atmospheric conditions (Yus-Díez et al., 2021). The following section aims to investigate potential drivers of $E_{abs}$ variations, including rBC wet scavenging, dominant rBC sources

and transport pathways. Particular attention will be paid to winter and summer because these seasons differ greatly, whereas spring and autumn behaviors appear intermediate.

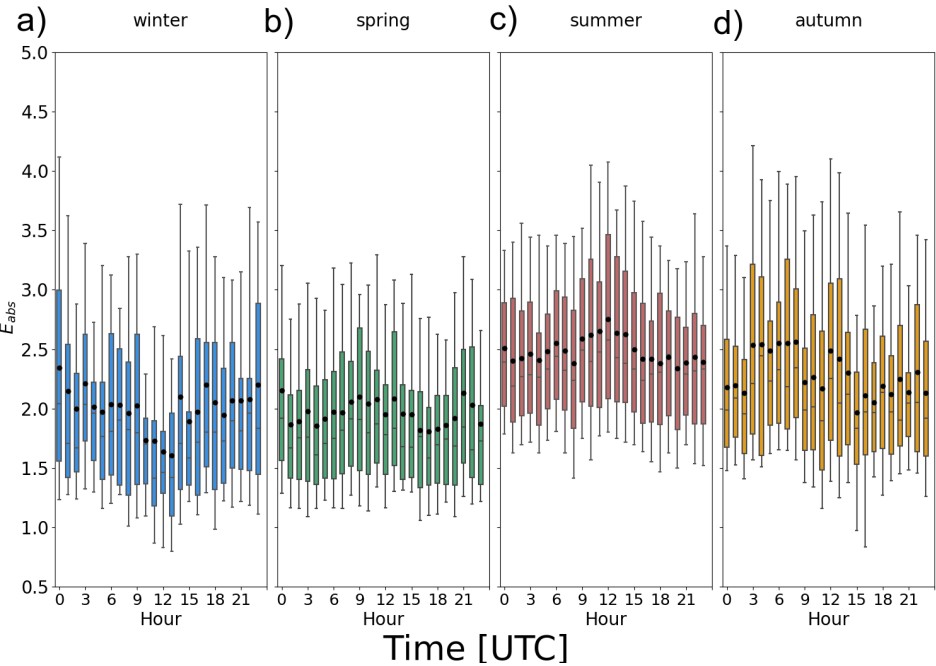

**Figure 7.** Diurnal cycles of $E_{abs}$ for each season during 2019-2020 period. Seasons are defined as follows: winter (December, January, February), spring (March, April, May), summer (June, July, August), and autumn (September, October, November). Boxes, lines, black dots and whiskers indicate 25$^{th}$ percentile, 75$^{th}$ percentile, median, mean, 10$^{th}$ percentile and 90$^{th}$ percentile, respectively.

## 3.4    Investigation of factors influencing rBC properties

### 3.4.1    The impact of wet scavenging on rBC properties

We first investigated whether $E_{abs}$ was modulated by a size-dependent rBC wet scavenging process during precipitation along their transport pathway. This hypothesis is based on the fact that the removal of particles is favored for the largest and thickly coated rBC because the activation of aerosols to cloud droplets is predominantly controlled by the particle size (Moteki et al., 2012; Mori et al., 2021; Ohata et al., 2016; Zhang et al., 2021). The wet removal of rBC was investigated by performing a cluster analysis using $\Delta M_{rBC}/\Delta CO$ data for which precipitation occurred or not along 72-h back trajectories computed by the HYSPLIT model. In order to decrease the influence of the difference sources on $\Delta M_{rBC}/\Delta CO$ compared to the effect of wet scavenging, periods for which the site was under PBL influence were filtered.

Figure 8a shows median $\Delta M_{rBC}/\Delta CO$ of 0.7 ng m$^{-3}$ ppbv$^{-1}$ for air masses affected by precipitation, against 2.1 ng m$^{-3}$ ppbv$^{-1}$ without precipitation during the transport of the air masses. The reduction of $\Delta M_{rBC}/\Delta CO$ by a factor of three suggests

that a significant removal process of rBC from the precipitation occurred along the transport pathway, apart from vertical transport from the PBL. As a reminder, time periods when PDM was under precipitations or humidity > 95 % have been filtered before the analysis. This result is confirmed by the dependence of $\Delta M_{rBC}/\Delta CO$ to RH in Fig. 8b, where a sudden decline of $\Delta M_{rBC}/\Delta CO$ appeared for highest RH > 80%, going from median $\Delta M_{rBC}/\Delta CO$ between 2.0 and 2.4 ng m$^{-3}$ ppbv$^{-1}$ for RH < 80% to a median $\Delta M_{rBC}/\Delta CO$ of $\sim 0.4$ ng m$^{-3}$ ppbv$^{-1}$ above 80% of RH. This high rBC removal by wet deposition result is in line with measurements performed in regions at similar altitudes, such as Puy de Dôme, Mt. Nanling, and Mt. Sonnblick, where wet deposition represents 30 to 70 % of the BC removing processes in the troposphere (Yang et al., 2019).

Figures 8 c-d show in contrast little influence of precipitation and RH on the rBC absorption enhancement, with a constant median $E_{abs}$ value of around $\sim 2.1$.

To better understand the negligible impact of rBC wet scavenging on $MAC_{rBC}$, we compared the measured rBC core size distribution of air masses affected or not by precipitation during their transport and under high RH conditions or not (Fig. 9). A two-fold lower $M_{rBC}$ in precipitation conditions compared to that without precipitation provides additional evidence for the dominant role of wet scavenging for rBC. The same result appeared by comparing rBC core size distribution under wet or dry conditions. However no significant change on the mean rBC core diameter was noticed between wet and dry conditions (mean $D_{rBC,core}$ of 177 and 182 nm, respectively), as well as in the presence of precipitations during the transport of rBC or not (mean $D_{rBC,core}$ of 178 and 185 nm, respectively).

This result contrasts with previous studies conducted in the FT showing a decrease in rBC size due to wet scavenging (Kondo et al., 2016; Moteki et al., 2012; Taylor et al., 2014; Liu et al., 2020a). For example, Kondo et al. (2016) found a change in $D_{rBC,core}$ between 13 and 20 nm depending on the season, while Liu et al. (2020a) and Moteki et al. (2012) measured rBC cores $\sim 32$ nm lower in air masses affected by wet removal. The insignificant effect of wet scavenging on the modal diameter of rBC core size distribution could be explained by the size of rBC core sampled at PDM that was higher than the one described in these studies. Hoyle et al. (2016) evidenced at the Jungfraujoch a threshold diameter of around 90 nm above which a particle activates to a droplet upon cloud formation. The majority of rBC sampled at PDM exhibited $D_{rBC,core}$ above this critical diameter. In addition droplet activation of an aerosol particle occurs when the supersaturation of the surrounding water vapor exceeds a critical value of supersaturation. Thus, it is not likely for freshly emitted BC particles to act as cloud condensation nuclei due to their hydrophobic nature unless the water vapor supersaturation is higher than 2% (Wittbom et al., 2014), far beyond the actual supersaturation (0.1–0.6%) in ambient air. Furthermore, the rBC wet removal by impaction is also a size-dependent process which could has been responsible for the removal of small rBC particles (Croft et al., 2010). However interstitial scavenging affects mostly particles smaller than 100 nm (Pierce et al., 2015), which have a very low presence at PDM (rBC mean diameter of 180 nm). As the ambient supersaturation varies depending on the environment, it is difficult to conclude whether the insensitivity of rBC size distribution and $E_{abs}$ to precipitation occurrence was solely due to the presence of large rBC particles at the sampling site or to a high ambient supersaturation in the precipitating clouds. Further measurements of simultaneous rBC wet removal and effective supersaturation are needed to test all these assumptions.

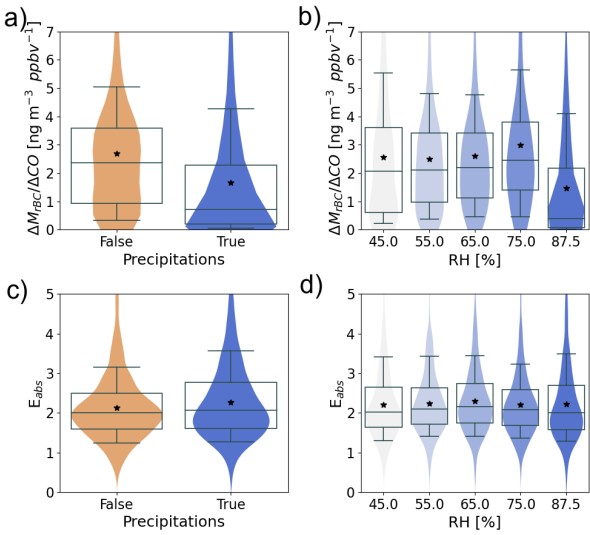

**Figure 8.** $\Delta M_{rBC}/\Delta CO$ emission ratio and $E_{abs}$ vs. a) and c) precipitation along the air mass back trajectory calculated with HYSPLIT model and b) and d) Relative Humidity measured at PDM. Violin plots represent the probability density function of each parameter. Statistics of the boxplots are the same as Fig. 7. PBL conditions were filtered before analysis.

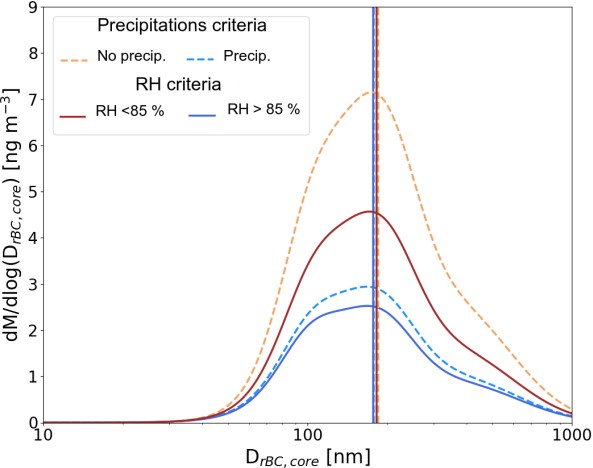

**Figure 9.** Mass size distributions of rBC core measured by the SP2 as a function of the presence or not of precipitation along the path of airmasses arriving at PDM, and whether the relative humidity was over 85 % or not. Vertical lines show the geometric mean rBC core diameter colored by the criteria described in the legend.

### 3.4.2 The contrasted seasonal influence of FT and PBL on rBC properties

Figure 10 shows the rBC properties classified by FT and PBL conditions (methodology in Section 2.5) and by seasons. As explained in Section 3.3.2, $\Delta M_{rBC}/\Delta CO$ ratio depends on the condition of combustion (fuel type, efficiency) and wet deposition by precipitation (Baumgardner et al., 2002; Taylor et al., 2014). We observed in Section 3.4.1 a large decrease of $\Delta M_{rBC/\Delta CO}$ when precipitation occurred during the transport of the air masses. In order to investigate the influence of rBC sources on rBC properties, precipitation events (air masses for which precipitation occurred along 72-h back trajectories) were removed in this section.

In winter, we measured higher $M_{rBC}$ values and variability in PBL conditions (39.5, 30.0 and 105 ng m$^{-3}$ for the median, 25$^{th}$, and 75$^{th}$ percentiles, respectively) than in FT conditions (33.5, 10.4 and 45.4 ng m$^{-3}$ for the median, 25$^{th}$, and 75$^{th}$ percentiles, respectively) (Fig. 10a). Furthermore the diurnal cycle of $M_{rBC}$ in winter PBL conditions showed an enhancement in the daytime (Fig. S8 in Supplement). This trend is consistent with intrusions of pollutants transported from PBL sources through convective mixing. During the night, pollution from the surface is trapped by the low height of the PBL and cannot reach the PDM. At the same time, the cleaner air transported in the FT may contribute to the dilution of $M_{rBC}$ at the PDM. The higher $\Delta M_{rBC}/\Delta CO$ in PBL conditions (2.5 $\pm$ 2.3 ng m$^{-3}$ ppbv$^{-1}$ for the mean $\pm$ STD) than in FT conditions (0.6 $\pm$ 0.1 ng m$^{-3}$ ppbv$^{-1}$ for the mean $\pm$ STD) may indicate additional sources from biomass combustion from the valley (Fig. 10b), which could be attributed to either residential wood heating or stubble-burning that is still a common practice in the Pyrenees (González-Olabarria et al., 2015). Figure 10c shows that PBL conditions were associated with lower $E_{abs}$ values (1.5 $\pm$ 0.3 for the mean $\pm$ STD) than FT conditions (1.9 $\pm$ 0.4 for the mean $\pm$ STD). Therefore, $E_{abs}$ was strongly modulated by atmospheric dynamics in winter.

During summer vertical transport from the PBL occurred about half of the days analyzed in this study. Surprisingly, the $M_{rBC}$ did not vary between BL and FT influence with values of 75.4 $\pm$ 33.2 ng m$^{-3}$ (Mean $\pm$ STD) and 80.2 $\pm$ 46.6 ng m$^{-3}$ respectively, meaning that the thermally driven PBL injection did not significantly impact $M_{rBC}$ measured at PDM (Figure 10d). This contrasts with our winter observations and most previous surface measurements at mountain sites, where the daytime PBL development has been shown to enhance aerosol mass concentration (Herrmann et al., 2015; Motos et al., 2020). The summer $M_{rBC}$ values at PDM are twice as high as those observed in winter, which indicates a massive additional regional transport of rBC in the FT and a lower contribution of rBC from PBL injection. $\Delta M_{rBC}/\Delta CO$ in PBL and FT conditions were close to each other, with values around 2.8 $\pm$ 1.6 ng m$^{-3}$ ppbv$^{-1}$ (Mean $\pm$ STD) and 3.3 $\pm$ 1.7 ng m$^{-3}$ ppbv$^{-1}$, respectively (Fig. 10e). This result indicates that the FT exhibited a significant background load of rBC at the continental scale, thus limiting the relative influence of PBL injection on $M_{rBC}$ during summer. The resulting $E_{abs}$ was remarkably similar for PBL vs. FT air mass categories (Fig. 10f). The high rBC loading transported in the FT, coupled with the higher $\Delta M_{rBC}/\Delta CO$ observed in the summertime (Fig. 10d and e), could be due to a strong influence of biomass burning emissions on the background FT in Europe. The majority of trajectories reaching PDM in summer have crossed the Iberian Peninsula and, previously, North Africa and North America (Fig. S9 in the Supplement). In these regions large fire events frequently occur, which may explain the high concentrations of strongly absorbing rBC observed at PDM during summer. This hypothesis is supported by Petetin et al.

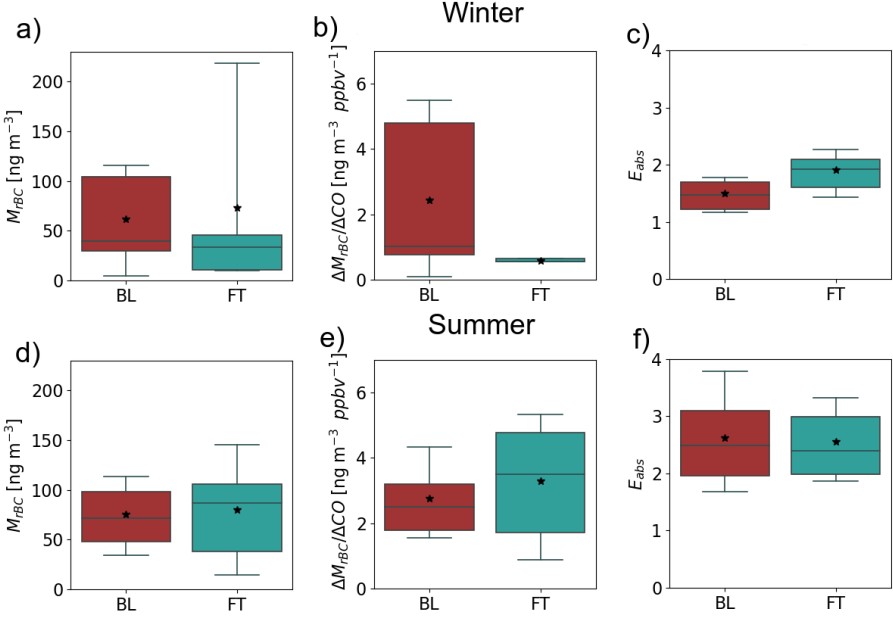

**Figure 10.** (a) rBC mass concentrations, (b) $\Delta M_{rBC}/\Delta CO$ emission ratio and (c) $E_{abs}$ as a function of the predominant influence at PDM in winter. The same for summer are given in d), e) and f). Red boxplots represents PBL conditions and green boxes are FT conditions. Precipitation events were filtered before analyses.

(2018) who showed that biomass burning aerosol accounts for about 43 - 81% of the CO concentration in lower FT in summer using in-situ airborne observations of CO from the IAGOS (In-service Aircraft for a Global Observing System) program. The ubiquitous presence of dilute biomass burning in the FT and its significant contribution to aerosol mass loading was also established using airborne measurements of ozone and precursor source tracers from the NASA Atmospheric Tomography mission (Bourgeois et al., 2021; Schill et al., 2020). Additional measurements of the aerosol chemical composition and in particular of a tracer of biomass burning in the atmosphere such as levoglucosan should be performed at PDM to confirm this.

A question remains about the cause of the diurnal variation of $E_{abs}$ in summer (Fig. 7c). As shown in Fig. 11, the $E_{abs}$ increase was not temporally correlated with the wind direction change from West-South-West to South, as evidenced by the 2-h delay between the two events. Furthermore, while the $E_{abs}$ increase occurs when $\Delta M_{rBC}/\Delta CO$ decreases, the $E_{abs}$ drop in the afternoon was not accompanied by an increase in $\Delta M_{rBC}/\Delta CO$. Then increase of $E_{abs}$ in the morning was most likely due to further aging and the appearance of heavily coated rBC rather than a change in rBC emission source. Several studies highlighted the major role of photochemical processes and extensive secondary aerosol generation to promote the light absorption enhancement of BC (Knox et al., 2009; Krasowsky et al., 2016; Liu et al., 2019; Wang et al., 2017; Xu et al., 2018; Yus-Díez

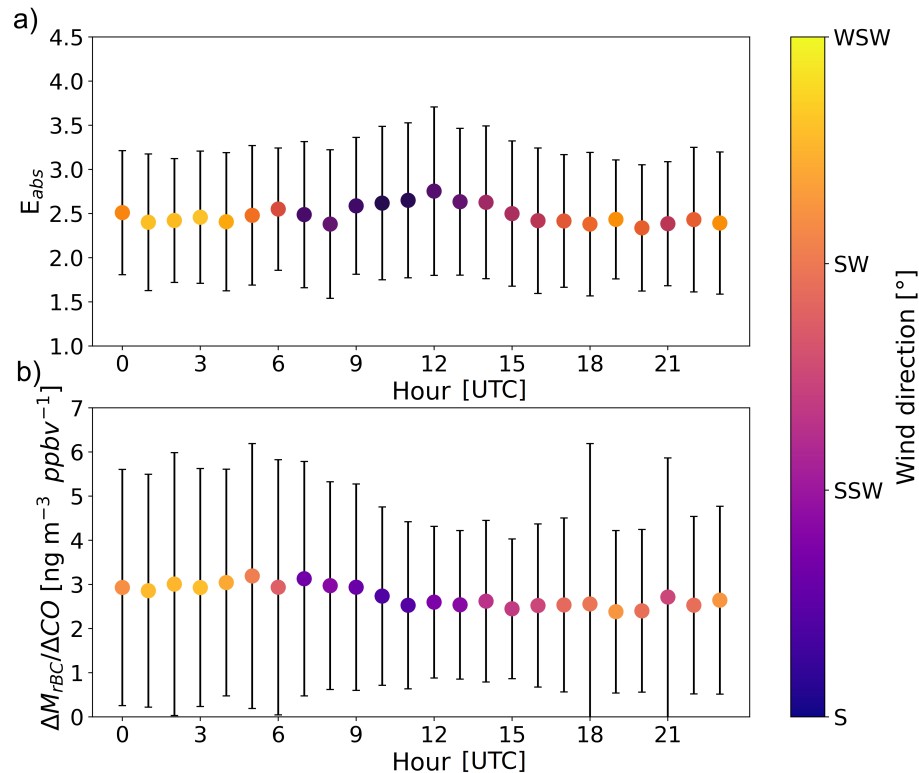

**Figure 11.** Hourly variation of (a) $E_{abs}$ and (b) $\Delta M_{rBC}/\Delta CO$ values in summer. Dots represent mean values and whiskers are one standard deviation. Dots are colored as a function of the wind direction.

et al., 2022). At PDM the enhanced $E_{abs}$ at noon was accompanied by a shift of the aerosol accumulation mode towards larger sizes, which may be due to the condensation of gaseous species on aerosol particles (Figure S10 in the Supplement). Simultaneously, a strong elevation of particle number concentration in the diameter range 10-30 nm can be observed, revealing new particle formation most likely produced by photochemical reactions at this time of the day. It is thus possible that rBC particles
became more coated via condensation of species produced by photochemical reactions at noon. However, it cannot be ruled out that the evolution of aerosol size distribution is a poor indicator of the rBC mixing state.

## 4  Summary and implications for climate models

Continuous two-year measurements of refractory BC (rBC) properties and additional aerosol characteristics have been performed at the high-altitude mountain site Pic du Midi in the French Pyrenees. The classification of the dominant aerosol type
using the spectral aerosol optical properties indicates that rBC is the predominant absorption component of aerosols at PDM

and controls the variation of SSA throughout the two years. The lower SSA in summer ($\sim$0.93) than in winter ($\sim$0.97) is correlated with a higher rBC number fraction, whereas the influence of BrC and dust was found negligible.

One key parameter to constrain BC absorption and associated radiative forcing in climate models is the refractive index of BC, and in particular the resulting $MAC_{BC}$. It was not clear if BC at high-altitude mountain sites should have a thicker or thinner coating than in urban or plain sites or even should be coated at all. On the one hand, the longer BC lifetime and the low temperature in the free troposphere (FT) favor thicker coating due to enhanced condensation of low-volatility compounds in colder environment. On the other hand, the low concentrations of particles and gaseous precursors in the FT may limit the coating processes. Our two-year long observations show that the overall net effect is a strong absorption enhancement with a mean $E_{abs}$ value of 2.2 $\pm$ 0.9.

The value of 7.5 m$^2$ g$^{-1}$ at $\lambda = 550$ nm of Bond and Bergstrom (2006) is the most common $MAC_{BC}$ used in climate models. The recommendation was based on a compilation of experimental results for freshly generated BC at and near sources obtained earlier than the early-2000s. Nevertheless this value is largely under the $MAC_{rBC}$ found in this study (9.2 m$^2$ g$^{-1}$ at $\lambda = 880$ nm, which can be converted to 14.7 m$^2$ g$^{-1}$ at $\lambda = 550$ nm assuming AAE = 1). The review by Moteki (2023) has also come to a similar conclusion. The reasons behind this bias should be better understood, in the light of observations such as those provided in the present study.

This study has notably shown the high variability of rBC properties measured in a remote site, where rBC have undergone long-range transport and aging. Certain causes of the large variability in $MAC_{rBC}$ have been eliminated and highlighted:

– Wet deposition is considered to be the main sink of BC, constraining its lifetime and size distribution, and thus its atmospheric concentration and optical properties. Our direct $\Delta M_{rBC}/\Delta CO$ measurements show the important role of wet deposition as a sink of rBC with around 67 % removed in the atmosphere by precipitation. However, we found a negligible impact of rBC wet removal process on both rBC size distribution and $E_{abs}$. This result may be due to the combination of large rBC particles reaching PDM ($D_{rBC,core}$ around 180 nm) and high critical supersaturation in precipitating clouds. The BC wet removal process was found to be one of the most misrepresented process in the representation of BC in models (Textor et al., 2006; Yu et al., 2019), leading to overestimated BC tropospheric concentrations and lifetime and *in fine*, a higher simulated radiative forcing (Samset et al., 2014; Schwarz et al., 2013). Substantial controversial and ambiguous issues in the wet scavenging processes of BC are apparent in current studies (Yang et al., 2019). Our results suggest that a bulk wet deposition parameterization (which does not account for particle size dependent scavenging) could realistically represent the actual BC wet scavenging at this site.

– rBC core was found to have a mean $D_{rBC,core}$ of 179 nm $\pm$ 28 nm, being reasonably independent of the season and day. There was no clear relationship between $MAC_{rBC}$ and $D_{rBC,core}$, which indicates that the variation in rBC core size was not responsible for the $MAC_{rBC}$ variability. Similar observations of rBC core size distribution in the atmosphere provided observational evidence of the stable distribution with a mode centered of around 200 nm approximately one day after emission (Liu et al., 2010; Schwarz et al., 2010; Shiraiwa et al., 2008). This self-similarity could greatly simplify the

representation of $MAC_{BC}$ in model simulations since a description of BC mixing state becomes the determinant factor of model performance when estimating BC optical properties and radiative forcing.

– Different time scales of air movements and atmospheric processes affect $MAC_{rBC}$ throughout the year. $MAC_{rBC}$ values were found higher in summer (geometric mean of 10.3 m$^2$ g$^{-1}$), when the influence of regional-scale motions dominates the rBC load, than in winter (geometric mean of 8.3 m$^2$ g$^{-1}$), when the influence of local-scale motions outweighs the rBC load. There are three possible explanations for this. (i) The plumes traveling in the FT tend to have a longer lifetime providing sufficient time for rBC aging during transport. In winter this results in a strong diurnal variability of $M_{rBC}$ ($E_{abs}$) with higher (lower) values in the middle of the day linked to the injection of rBC originating from the planetary boundary layer (PBL). However the aging timescale can not be the only explanatory factor since thermally driven PBL injection did not significantly impact $M_{rBC}$ and $E_{abs}$ in summer and higher values have been observed in summer than in winter for similar FT conditions. (ii) The source of rBC emission was different between the winter and summer seasons. Combining $\Delta M_{rBC}/\Delta CO$ with air mass transport analysis, we observed additional sources from biomass burning in summertime leading to higher $M_{rBC}$ and $E_{abs}$. (iii) Different aging processes occur between seasons, such as photochemical activity that could explain the observed amplification of light absorption by rBC around noon. However, the latter effect could not be rigorously demonstrated in this study.

The complexity and diversity of BC mixing states in the real atmosphere cannot be represented in climate models, and therefore these models generally use simplified schemes. A fixed e-folding timescale (1–3 days) is commonly used as the turn-over time for converting fresh BC particles into aged ones (Myhre et al., 2013). In addition, atmospheric models necessarily approximate the full complexity and diversity of BC composition, which can lead to mismatches with observed $E_{abs}$ (Fierce et al., 2020). The findings presented here suggest that different dynamic processes governing rBC light absorption occur during the day and night, and between summer and winter. A parameterization of BC aging explicitly based on aerosol microphysical processes, in which the conversion rate is considered to vary depending on the environmental conditions (e.g., temperature, photochemical activity,...) and some key species (e.g., aerosol, coating precursors,...) may be required to acurately represent the true variability of $MAC_{BC}$.

*Data availability.* Aerosol microphysical and optical properties are freely available at http://ebas.nilu.no/ (NILU, 2018). CO data are available on the ICOS platform at https://www.icos-cp.eu/. rBC data are available upon request to the authors.

*Author contributions.* ST and CD designed the study, developed the analysis protocols, and wrote the initial manuscript. PN contributed to the data analysis. FG and VP provided data and methods to analyse them. TB, FG, EL, ST, VP and CD contributed to the measurement campaign. All authors reviewed the final manuscript.

*Competing interests.* The authors declare that they have no conflict of interest.

*Acknowledgements.* This work received funding from the French national program LEFE/INSU and Météo-France. Observation data were collected at the Pyrenean Platform for Observation of the Atmosphere P2OA (http://p2oa.aero.obs-mip.fr). P2OA facilities and staff are funded and supported by the University Paul Sabatier Toulouse 3, France, and CNRS (Centre National de la Recherche Scientifique). We especially thank the staff of the Pic du Midi platform (Observatoire Midi-Pyrénées) for their technical assistance. We acknowledge the SNO ICOS-France and ACTRIS-France forsupporting greenhouse gases and aerosol observations at PDM, data collection, processing and dissemination. We also acknowledge the use of the HYSPLIT model and READY websitefrom the NOAA Air Ressources Laboratory (http://ready.arl.noaa.gov). The authors would like to thank the reviewers for their useful comments, which helped to improve the quality of the document. A final acknowledgement goes to Dr. Martin Gysel Beer, who brought his expertise to review this manuscript.

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
