# Peer review of "Higher absorption enhancement of black carbon in summer shown by two year measurements at the high-altitude mountain site of Pic du Midi Observatory in the French Pyrenees"

_EGUsphere, 2023_

## Author Comment (AC1)

**Answer to referees**

**Higher absorption enhancement of black carbon in summer shown by two year measurements at the high-altitude mountain site of Pic du Midi Observatory in the French Pyrenees**

We thank the three reviewers for evaluating the manuscript and providing us constructive and useful comments. Referees have common concerns which we addressed to the best of our possibilities:

- In order to better highlight the objectives and the main results of the paper (1) we modified the title, (2) clearly listed the scientific questions addressed in the introduction section, (3) modified the conclusion into a section "Summary and implications for climate models", (4) moved Figure 2 from the meteorological section to the Supplement, (5) moved Figure S7 and Figure S9 from the Supplement to the main text in Sections 3.4.1 and 3.4.2, respectively.

- We clarified and completed the section on aerosol optical properties by adding a more complete description of the observed parameters and by reworking the design of Figures 3 and 5.

- We added two elements in the Supplement describing the processing of the SP2 measurements: a Section S1 comparing BC mass concentrations obtained from our software on PYTHON and the PSI SP2 Toolkit running on IGOR and a Section S2 to explain the processing of the rBC core size distribution. A Section S5 about the discrimination of Free Tropospheric/ Planetary Boundary Layer conditions has been added in the Supplement.

- We changed the "BC" nomenclature to "rBC" to follow the recommendations by Petzold et al. (2013) for measurements performed by a SP2.

Please find below reviewer comments in black and our responses in blue. The line numbers in the responses refer to the new version of the paper.

**Anonymous Referee #2**

**COMMENT #1**:Instrumentation: In Sec. 2.2.1, the air was heated to keep the samplings dry. Could this measure alter the properties of BC, e.g., leading to some mass loss of volatile substances in the coating materials of BC particles?

**REPLY :** We followed the WMO-GAW and ACTRIS guidelines, which recommend a relative humidity below 40 % for sampling in order to avoid water uptake and be able to compare physical and optical measurements (GAW Report No. 227:WMO/GAW Aerosol Measurement Procedures, Guidelines and Recommendations, 2nd Edition2016). The Whole Air Inlet was heated to 20 °C, which is low enough to ensure that the sampling step did not affects the BC mixing state.

**COMMENT #2:** It will be also useful to examine the coated size of BC measured by the SP2 for the discussion about the mixing state of BC.

**REPLY :** We agree with the reviewers that a closure with the rBC coating measurement would reinforce our results on absorption enhancement However, due to a technical issue on the low gain of the scattering channel of the SP2, we could not provide rBC mixing state measurements.

**COMMENT #3:**Eq. (1) and Eq. (2) are wrong, it should be log (450/635) instead of log(450)/log(635).

**REPLY :** We thank the reviewer for this correction which we have taken into account.

**COMMENT #4:**Line 276: Based on your results, what can be the main influence that caused a higher ΔBC/ΔCO in summer than in winter when considering the larger amount of precipitation in summertime?

**REPLY :** Even after filtering precipitation days, we found a higher $\Delta M_{rBC}/\Delta CO$ in summer than in winter (cf. l. 333-334 :"Summer ratios were generally higher than winter emission ratios, which could reflect either lower rBC scavenging during transport or different emission sources of rBC between seasons..").

This means that the seasonal variation of $\Delta M_{rBC}/\Delta CO$ is probably due to different BC sources. The higher $\Delta M_{rBC}/\Delta CO$ in summer is most probably due to biomass burning emissions, which produce a higher $\Delta M_{rBC}/\Delta CO$ than fossil fuel emissions (Guo et al., 2017; Pan et al., 2011; Zhu et al., 2019). Further explanations on $\Delta M_{rBC}/\Delta CO$ variations was provided in lines 371-346. The argument of the biomass burning influence in summer is reinforced by the study of Dupuy et al. (2020) who found that Europe is strongly impacted by wildfires.

**COMMENT #5:**Line 313-314: "It is expected that the BC particles reaching PDM would be aged and relatively thickly coated." some evidence is needed here, a previous study which also conducted experiments on a mountain site influenced by the PBL intrusion (doi: 10.5194/acp-19-6749-2019) may support your results.

**REPLY :** For clarity reasons, the sentence in l. 389-392 was modified and references has been added:
" Given the remote mountain location and presumable distance from fresh BC sources, BC particles reaching PDM may have undergone aging and have gained a consistent coating. Previous studies found an absorption enhancement of BC due to its coating with the aging time (Peng et al., 2016; Sedlacek et al., 2022; Yus-Díez et al., 2022)."

**COMMENT #6:**The authors use "the size of BC core sampled at PDM was higher than other studies" to explain "BC wet scavenging did not significantly affect the size of BC-containing particles," which is not convincing. A previous study performed on a mountain site during wintertime in Beijing reported larger BC core than the present study but showed significantly lowered BC core size after wet scavenging due to larger BC cores were preferentially wet removed (doi: 10.1029/2019GL083171). Please give more explanations.

**REPLY :** In order to explain the impact of rBC wet scavenging on rBC size distribution, we compared the measured rBC core size distribution (see Fig. 9, which has been moved from the Supplement to the main text) of air masses affected or not by precipitation during their transport. The resulting rBC core size distribution were similar in both cases, which suggest that there was not a preferential wet removal of bigger BC cores during their transport to PDM.

The study of Ding et al. (2019) was conducted at a mountain site frequently under the influence of anthropogenic emissions coming from sources closer than the one influencing PDM. BC-containing particles may have different size and Kappa values than those measured at the PDM. Furthermore, due to different meteorological condition, supersaturation in clouds may significantly differs between the two sites. The cloud supersaturation is a key parameter impacting the critical activation diameter of Cloud Condensation Nuclei (CCN). Further simultaneous measurements of rBC mixing state and effective supersaturation in precipitating clouds are needed to test these assumptions.

**COMMENT #7:** The wet scavenging process of BC due to PBL cloud during its vertical transport within PBL around midday may partly explain the relationship between the diurnal variation of Eabs and ΔBC/ΔCO in summer (doi: 10.1029/2020JD033096)

**REPLY :** We thank Reviewer #2 for this comment. In order to eliminate the effect of rBC wet removal during PBL intrusions on $E_{abs}$ and ΔBC/ΔCO, Fig. 8 has been modified by filtering periods when PDM was under PBL conditions. Similar results as before have been obtained with lower $\Delta M_{rBC}/\Delta CO$ obtained under precipitation, which suggests that rBC has not been removed during its vertical transport.

Minor:

**COMMENT #1:** Line 171: "1,95 should be 1.95"
**REPLY :** The notation has been corrected.

**COMMENT #2:** Line 173: what do you mean by "artifacts"
**REPLY :** An explanation has been added in lines 217-218 :
"Under precipitations some water droplet may indeed enter in the aerosol inlet and change both the inlet cut off diameter and the measured aerosol size distribution. This would bias all the measured aerosol properties. "

**COMMENT #3:** BL and PBL need to be uniform.
**REPLY :** The abbreviation PBL has been retained and harmonized in the manuscript.

**COMMENT #4:** Abstract, line 18: This sentence was confusing.

**REPLY :** The sentence in lines 18-20 has been modified and now reads :

"On the contrary, in summer, $M_{rBC}$ showed no diurnal variation despite more frequent PBL conditions, implying that $M_{rBC}$ fluctuations are rather dominated by regional and long-range transport in the FT."

References:

Ding, S., Zhao, D., He, C., Huang, M., He, H., Tian, P., Liu, Q., Bi, K., Yu, C., Pitt, J., Chen, Y., Ma, X., Chen, Y., Jia, X., Kong, S., Wu, J., Hu, D., Hu, K., Ding, D., & Liu, D. (2019). Observed Interactions Between Black Carbon and Hydrometeor During Wet Scavenging in Mixed-Phase Clouds. *Geophysical Research Letters, 46*(14), 8453-8463. https://doi.org/10.1029/2019GL083171

Dupuy, J., Fargeon, H., Martin-StPaul, N., Pimont, F., Ruffault, J., Guijarro, M., Hernando, C., Madrigal, J., & Fernandes, P. (2020). Climate change impact on future wildfire danger and activity in southern Europe : A review. *Annals of Forest Science, 77*(2), 35. https://doi.org/10.1007/s13595-020-00933-5

Guo, Q., Hu, M., Guo, S., Wu, Z., Peng, J., & Wu, Y. (2017). The variability in the relationship between black carbon and carbon monoxide over the eastern coast of China : BC aging during transport. *Atmospheric Chemistry and Physics, 17*(17), 10395-10403. https://doi.org/10.5194/acp-17-10395-2017

Pan, X. L., Kanaya, Y., Wang, Z. F., Liu, Y., Pochanart, P., Akimoto, H., Sun, Y. L., Dong, H. B., Li, J., Irie, H., & Takigawa, M. (2011). Correlation of black carbon aerosol and carbon monoxide in the high-altitude environment of Mt. Huang in Eastern China. *Atmospheric Chemistry and Physics, 11*(18), 9735-9747. https://doi.org/10.5194/acp-11-9735-2011

Peng, J., Hu, M., Guo, S., Du, Z., Zheng, J., Shang, D., Levy Zamora, M., Zeng, L., Shao, M., Wu, Y.-S., Zheng, J., Wang, Y., Glen, C. R., Collins, D. R., Molina, M. J., & Zhang, R. (2016). Markedly enhanced absorption and direct radiative forcing of black carbon under polluted urban environments. *Proceedings of the National Academy of Sciences, 113*(16), 4266-4271. https://doi.org/10.1073/pnas.1602310113

Sedlacek, A. J. I., Lewis, E. R., Onasch, T. B., Zuidema, P., Redemann, J., Jaffe, D., & Kleinman, L. I. (2022). Using the Black Carbon Particle Mixing State to Characterize the Lifecycle of Biomass Burning Aerosols. *Environmental Science & Technology, 56*(20), 14315-14325. https://doi.org/10.1021/acs.est.2c03851

Yus-Díez, J., Via, M., Alastuey, A., Karanasiou, A., Minguillón, M. C., Perez, N., Querol, X., Reche, C., Ivančič, M., Rigler, M., & Pandolfi, M. (2022). Absorption enhancement of black carbon particles in a Mediterranean city and countryside : Effect of particulate matter

chemistry, ageing and trend analysis. *Atmospheric Chemistry and Physics, 22*(13), 8439-8456. https://doi.org/10.5194/acp-22-8439-2022

Zhu, C., Kanaya, Y., Yoshikawa-Inoue, H., Irino, T., Seki, O., & Tohjima, Y. (2019). Sources of atmospheric black carbon and related carbonaceous components at Rishiri Island, Japan : The roles of Siberian wildfires and of crop residue burning in China. *Environmental Pollution, 247*, 55-63. https://doi.org/10.1016/j.envpol.2019.01.003

---

## Author Comment (AC2)

**Answer to referees**

**Higher absorption enhancement of black carbon in summer shown by two year measurements at the high-altitude mountain site of Pic du Midi Observatory in the French Pyrenees**

We thank the three reviewers for evaluating the manuscript and providing us constructive and useful comments. Referees have common concerns which we addressed to the best of our possibilities:

- In order to better highlight the objectives and the main results of the paper (1) we modified the title, (2) clearly listed the scientific questions addressed in the introduction section, (3) modified the conclusion into a section "Summary and implications for climate models", (4) moved Figure 2 from the meteorological section to the Supplement, (5) moved Figure S7 and Figure S9 from the Supplement to the main text in Sections 3.4.1 and 3.4.2, respectively.

- We clarified and completed the section on aerosol optical properties by adding a more complete description of the observed parameters and by reworking the design of Figures 3 and 5.

- We added two elements in the Supplement describing the processing of the SP2 measurements: a Section S1 comparing BC mass concentrations obtained from our software on PYTHON and the PSI SP2 Toolkit running on IGOR and a Section S2 to explain the processing of the rBC core size distribution. A Section S5 about the discrimination of Free Tropospheric/ Planetary Boundary Layer conditions has been added in the Supplement.

- We changed the "BC" nomenclature to "rBC" to follow the recommendations by Petzold et al. (2013) for measurements performed by a SP2.

Please find below reviewer comments in black and our responses in blue. The line numbers in the responses refer to the new version of the paper.

**Anonymous Referee #3**

**Major comments**

Tinorua et al. provides a description of an important black carbon data set collected at a high-altitude mountain site in France – these sorts of data sets and measurements are indeed essential to improving climate models. However, the impact of this data set is not compellingly described in the manuscript – there are some sections bogged down in numbers and lists, some paragraphs that making sweeping general statements without the rigorous analysis / discussion that I would expect of an ACP paper, and perhaps most importantly, the implications of the findings on climate models (as described in the article introduction) are limited to a brief 6-line paragraph at the end of the manuscript.

These problems of narrative, hypothesis generation and testing, and rigor can even be seen in the title, which is overly generic – I would like to see the authors more precisely focus and describe their key findings and make sure these findings are well supported in the paper, clear in the title and that the methods are described in more detail. In summary, I think the manuscript has a lot of

potential after major revisions to sharpen the focus. I look forward to this paper being published in ACP after making these adjustments.

There are also some grammatical / style issues throughout that would benefit from an additional proofread, and the time series figures spanning the full 2-year campaign are generally overwhelming. I would recommend that some of these figures be moved to the SI and that focused vignettes of the data be presented in the main manuscript. Additionally, the Supporting Information lacks a lot of detail and description; it is simply a collection of figures, which limits its utility.

**Specific Comments**

**COMMENT #1 :** Title: I recommend making the title more descriptive of the findings. It's too generic as written.

**REPLY :** The title has been changed as follows:

"Higher absorption enhancement of black carbon in summer shown by two year measurements at the high-altitude mountain site of Pic du Midi Observatory in the French Pyrenees"

**COMMENT #2 :** Line 54: Can you define what is meant by "short periods" – currently subjective

**REPLY :** The sentence in lines 56-57 has been completed as follows :

"However, most of these measurements were performed in the Planetary Boundary Layer (PBL) and over short periods from a few hours to as long as a season."

**COMMENT #3 :** Figure 1: Map resolution is poor – can you make this clearer?

**REPLY :** Figure 1 has been edited. It now has a better resolution. Maps at the regional and local scales have also been added.

**COMMENT #4 :** Lines 89-91: I would like to see a bit more description of the inlet and its design – what is the total flow rate and diameter? What are the particle losses? If there is a citation that describes this inlet in more detail perhaps it could be provided.

**REPLY :**

A Whole Air Inlet (WAI) is installed at the PDM as recommended by ACTRIS for measurement sites frequently in clouds and/or freezing conditions. This inlet samples clouds droplets and interstitial aerosol particles up to a diameter of around 10 µm.

Section 2.2.1 in lines 96-102 has been modified as follows:

"All particle-measuring instruments sampled air taken in parallel from a whole air inlet, located 2 m above the building roof. This inlet is used for the long-term observations in mountainous sites and designed to maintain an isokinetic and laminar flow. The main flow rate was fixed at about 460 l min$^{-1}$. The splitter was fixed at the end of the stainless tube. The hat of the whole air inlet and the stainless tube were both thermo-regulated in order to avoid frost and gradually regulate the temperature of the samples air to the measurement room. The air was heated to around 20°C in order to perform aerosol in-situ measurements at a relative humidity lower than 30 %. The

instrumental room temperature was regulated at around 20°C. The annual cycle of the dew point temperature varied between about -10° and +5°C."

**COMMENT #5 :**Line 101-102: The process of developing the size distributions shown in Figure S1 in the Supplement are not described – I don't see any descriptive text in the SI, either. Can you provide some more detail in the SI? Feel like there is a big jump here. How are you measuring size distribution? How do you know that all of the ultrafine particles are indeed BC? I think I am just missing a few steps here because I don't have the relevant expertise / background.

**REPLY :** We agree that we did not provide sufficient elements to understand the issue. We have reformulated the sentence and added a Section S2 in the Supplement to describe the processing of rBC size distribution measured by the SP2 and the estimated missing mass fraction.

The sentence in lines 118-120 was revised as :

"However, the observed size distributions showed that an important fraction (around 12%) of $M_{BC}$ at diameters below 90 nm is not measured by the SP2 (Figure S1 in the Supplement)"

The following text has been added to the Supplement :

**"Text S2 : Information about the rBC size distribution processing**

The SP2  measures rBC cores from mass equivalent diameters of 90 to 580 nm. Fig. S2 shows the two-year average of the daily rBC cores size distributions. It can be noticed in Fig. S2 that the number size distribution measured by the SP2 did not cover the full size range of rBC at PDM. This is particularly true for the rBC particles below 90 nm, where the major fraction of the $M_{rBC}$ was missed by the SP2. In order to estimate the missing rBC mass fraction undetected by the SP2 (e.g. the mass size distribution under 90 nm and over 580 nm), the daily rBC size distributions were fitted with a sum of three  lognormal functions as :

$$M_{rBC} = dM/dln(D_{rBC}) = \sum_{i=0}^{3} \left( \frac{M_i}{\sqrt{2\pi}\ln(\sigma_{g,i})} \exp\left[ \frac{-\ln^2(d/d_{g,i})}{2\ln^2(\sigma_{g,i})} \right] \right)$$

with $M_i$, $d_{g,i}$ and $\sigma_{g,i}$ representing the rBC mass concentration, the geometric mean diameter and the geometric standard deviation of the mode i, respectively. The same function with two modes has been used to fit the number size distribution.

The fitting parameters were constrained in the following ranges : Mode 1 : $50 < d_{g,1} < 100$ nm and $1.2 < \sigma_{g,1} < 3$; Mode 2 : $150 < d_{g,2} < 250$ nm and $1.3 < \sigma_{g,2,} < 2.9$; Mode 3 : $350 < d_{g,3} < 500$ nm and $1 < \sigma_{g,3} < 3$. "

**COMMENT #6 :**Section 2.2.3: I see now that you are describing the SMPS in the 10-1000 nm size range – would recommend re-ordering the manuscript so you aren't introducing results before you have mentioned the methods. My question on how you know the size distribution below 90 nm is BC still stands.

**REPLY :** Section 2.2.3 focuses on the measurement of aerosol properties. The SMPS was used to measure aerosol size distribution between 12.6 nm and 532.6 nm size range. We used a SP2 to measure the rBC size distribution and it is described in section 2.2.2.

**COMMENT #7 :**Lines 124-126: I do not believe these in preparation materials were provided for review – can you provide more details on the calculation of the C value in the response to reviewers?

**REPLY :** The following sentence has been added in Section 2.2.3, lines 143-145:

"The multiple scattering parameter used to correct the measured attenuation was set to 3.22, according to the value obtained at λ=880 nm by Yus-Díez et al. (2021) at the mountainous site of Montsec d'Ares located less than 200 km from the PDM. "

**COMMENT #8 :**Section 2.3: I find the description of the equations (1-3) to be confusing to follow. You mention on line 143 the calculation of the aerosol absorption coefficient at 635 nm using another AAE calculation but the justification for this approach seems to be missing. I believe this is linked to the nephelometer wavelength ranges but its difficult to follow the narrative.

**REPLY :** We agree that there was missing elements about the method used to calculate AAE between 450 and 635 nm. Therefore, we have added the measurement wavelengths of the aethalometer in Section 2.2.3 and we have completed the text in Section 2.3, lines 161-165, as follows:

"The spectral dependence of $\sigma_{ap}$ was characterized by the Absorption Angstrom Exponent ($AAE_{aer,450-635}$) calculated between 450 and 635 nm as follows :

$$AAE_{aer,450-635} = \frac{-\log\left(\frac{\sigma_{ap,450}}{\sigma_{ap,635}}\right)}{\log\left(\frac{450}{635}\right)}$$

For this calculation, $\sigma_{ap,470}$ and $\sigma_{ap,660}$ from the aethalometer were adjusted at the wavelengths of 450 and 635 nm measured by the nephelometer using the AAE calculated from the aethalometer between 370-470 nm and 590-660 nm. "

**COMMENT #9 :**Line 161: You are citing material that to my knowledge is not available for review – please provide details even if it is confidentially to reviewers only. I would like to see some of these details from the in preparation manuscript if possible.

**REPLY :** Some details has been added in Section 2.3, lines 196-198:

"In addition, we filtered out periods when dust were sampled at PDM for the calculation of $MAC_{rBC}$ since Yuz-Diez et al. (2021) observed significant biases in the multiple scattering correction of the aethalometer AE33 during such events."

**COMMENT #10 :**Lines 167-171: I find the justification of choice for MAC_bare to be lacking – what are the implications of your choice? Is this still standard practice given that you are citing a paper from 2006? Given the interest in the subject matter I require more convincing that this is the appropriate method.

**REPLY :** A text in Section 2.3, lines 207-215, has been added to justify the use of Mie theory to calculate $MAC_{bare,rBC}$:

"The calculation of $MAC_{bare,rBC}$ using Mie's theory assume a simplified spherical assumption of rBC morphology. However rBC may exhibit complex morphologies whose optical behavior is

imperfectly predicted by Mie's theory, introducing a bias in the retrieved $MAC_{bare,rBC}$ (Saleh et al., 2016). It might be considered that Mie's theory is suitable for estimating the absorption of highly aged rBC, which exhibit an internally mixed core-shell structure. China et al. (2015) used this method to calculate the $E_{abs}$ of rBC in a high-altitude site of the Azores Islands because the large majority (70%) of these long-range transported particles were found highly compacted. Several studies found that Mie's scattering model captures basic optical properties of BC in biomass burning plumes (Liu et al., (2017), Denjean et al., (2020), Zanatta et al., (2018)) calculated $MAC_{rBC}$ of heavily coated rBC particles from the Arctic region using Mie's theory and found consistent results with direct measurements. "

**COMMENT #11 :** Figure 2: I'm not sure this figure has a lot of value in the main manuscript given that you have already described the findings in the text in Section 3.1. I would rather see this figure in the SI and instead have you provide more details on the methods section, which I found to be lacking.

**REPLY :** We agree with this comment. Figure 2 has been moved to the SI.

**COMMENT #12 :** Figure 4: The time series are difficult to follow because you are plotting the full two years. I am not sure this is the most effective way to communicate this information. Perhaps you can re-arrange to highlight the seasonality you describe in the text and move the full time series to the SI.

**REPLY:** We thank the reviewer for the suggestion . We decreased the time resolution of Figure 4 and Figure 6 (now Figure 3 and 5) to one month.

**COMMENT #13 :** Lines 253-254: If this is a key conclusion of the work then this should be better reflected in abstract / title of the paper. Also given that it was unexpected, can you probe more deeply why you think this was observed and provide more detail that you are confident that it isn't an artifact of your sampling methods or calculations? Would just like to see this better and more specifically justified.

**REPLY:** We are confident in the sampling method that follows the ACTRIS and WMO-GAW guidelines. The uncertainties on the calculation of optical parameters and rBC concentration have been quantified. We clarified and completed the section on aerosol optical properties by adding a more complete description of the observed parameters and by reworking the design of Figures 3 and 5.

**COMMENT #14 :** Lines 297-298: I feel like you could be more precise describing how different techniques and correction factors influence the differences. This feels overly generalized as written. Additionally, if you have confidence in your very high MAC value, then what are the implications of this?

**REPLY:** The paragraph comparing $M_{rBC}$ and $MAC_{rBC}$ has been modified and completed in Section 3.3, l. 358-367:

"The ambient $MAC_{rBC}$ was around $9.2 \pm 3.7$ m² g$^{-1}$ at $\lambda=880$ nm (Fig. 5d). Several studies previously reported $MAC_{rBC}$ values between 8.9 to 13.1 for measurements at $\lambda=637$ nm in European mountain stations (Pandolfi et al., 2014; Yus-Díez et al., 2022; Zanatta et al., 2016). By using a AAE of unity, these values can be converted to $MAC_{rBC}$ between 6.4 and 9.5 m² g$^{-1}$ at $\lambda=880$ nm. These studies used different measurement techniques, analysis method and correction factors from ours for estimating $MAC_{BC}$ that makes difficult the comparison of $MAC_{rBC}$ derived from different instruments. Pandolfi et al. (2014) performed a linear regression between $\sigma_{ap,637}$ values measured by a Multi-Angle Absorption Photometer (MAAP) and daily $M_{EC}$ values from off-line filter-based measurements by a SUNSET OCEC Analyzer. Yus-Díez et al. (2022) and Zanatta et al. (2016) retrieved $MAC_{BC}$ with these instruments by calculating the ratio between the two parameters instead of a linear regression. Because of the absence of a standard method for quantifying $M_{BC}$, the absolute uncertainties on the $MAC_{BC}$ obtained in the literature are very high ranging from $\pm30$ to 70% (Zanatta et al., 2016)."

The implication of our results on BC representation in the climate models has been detailed in the last Section "Summary and implications for climate models" .

**COMMENT #15 :**Figure 7: You should have a consistent scale for panels (a) and (b).

**REPLY :** The scales of Figure 7 (now Fig. 6) have been changed.

**COMMENT #16 :**Line 320: Please avoid sensational language like "remarkable". Do you mean a notable diurnal profile?

**REPLY :** Yes, it was a linguistic confusion. This was changed in lines 398-399, as follows:

"There was a notable opposite diurnal profile between seasons in $E_{abs}$ with midday showing a minimum around 1.7 in winter, and a maximum around 2.9 in summer."

**COMMENT #17 :**Section 4.1 and wet scavenging: Conclusions about the significance of precipitation I think would benefit from a more rigorous statistical test vs descriptive observations that you have provided. Additionally, this whole section reads as quite speculative. This is important given that this is one of your main conclusions of note.

**REPLY :** The Figure (now Fig. 9) showing the influence of precipitation to the rBC core size distribution has been added to the main text for clarity. We also modified the text in section 4.1 to include values and statistics, lines 414-428:

"Figure 8a shows median $\Delta M_{rBC}/\Delta CO$ of 2.1 ng m$^{-3}$ ppbv$^{-1}$ for air masses affected by precipitation, against 0.7 ng m$^{-3}$ ppbv$^{-1}$ without precipitation during the transport of the air masses. The reduction of $\Delta M_{rBC}/\Delta CO$ by a factor of three suggests that a significant removal process of rBC from the precipitation occurred long the transport pathway, apart from vertical transport from the PBL. This result is confirmed by the dependence of $\Delta M_{rBC}/\Delta CO$ to RH in Fig. 8b, where a sudden decline of $\Delta M_{rBC}/\Delta CO$ appeared for highest RH>80%, going from median $\Delta M_{rBC}/\Delta CO$ between 2.0 and 2.4 ng m$^{-3}$ ppbv$^{-1}$ for RH<80% to a median $\Delta M_{rBC}/\Delta CO$ of ~ 0.4 ng m$^{-3}$ ppbv$^{-1}$ above 80% of RH.

Figures 8c-d show in contrast little influence of precipitation and RH on the rBC absorption enhancement, with a constant median $E_{abs}$ value of around ~ 2.1.

To better understand the negligible impact of rBC wet scavenging on $MAC_{rBC}$, we compared the measured rBC core size distribution of air masses affected or not by precipitation during their transport and under high RH conditions or not (Figure 9). A two-fold lower $M_{rBC}$ in precipitation conditions compared to that without precipitation provides additional evidence for the dominant role of wet scavenging for rBC. The same result appeared by comparing rBC core size distribution under wet or dry conditions. However wet scavenging did not significantly altered the modal diameter of rBC core size distribution. "

**COMMENT #18 :** Lines 348-350: Can you clarify why precipitation events were removed? Because of the influence on the ΔBC/ΔCO ratio? Was slightly unclear to me.

**REPLY :** The objectives of the analysis of $\Delta M_{rBC}/\Delta CO$ ratio in Section 3.4.2 is to investigate the sources of rBC transported in the free troposphere and coming from the boundary layer. In order to avoid potential variation of $\Delta M_{rBC}/\Delta CO$ ratio due to wet deposition rather than a difference in rBC source, precipitation events (Air masses for which precipitation occurred along 72-h back trajectories computed by the HYSPLIT model) were removed in the section 3.4.2

.A text in Section 3.4.2, lines 442-447 has been added to precise the reasons why precipitation events were removed:

"As explained in Section 3.3, $\Delta M_{rBC}/\Delta CO$ ratio depends on the condition of combustion (fuel type, efficiency) and wet deposition by precipitation (Baumgardner et al., 2002; Taylor et al., 2014). We observed in section 3.4.1 a large decrease of $\Delta M_{rBC}/\Delta CO$ when precipitation occurred during the transport of the air masses. In order to investigate the influence of rBC sources on rBC properties, precipitation events (air masses for which precipitation occurred along 72-h back trajectories) were removed in this section."

**COMMENT #19 :** Lines 378-379: Can you describe what these additional measurements would be? And the more precise source apportionment?

**REPLY :** We modified the sentence in lines 476-477 to precise which additional measurements could allow to validate our hypothesis :

"Additional measurements of the aerosol chemical composition and in particular of a tracer of biomass burning in the atmosphere such as levoglucosan should be performed at PDM to confirm this. "

**COMMENT #20 :** Lines 420-427: To me these are the most important lines in the manuscript – the implications of your findings. I am a little disappointed that this is relegated to one brief paragraph and that your major conclusion re: wet scavenging is not as thoroughly assessed in the paper as is likely warranted given the conclusion. I would like to see the implications section for climate models more rigorously discussed.

**REPLY:** We added the implications of our results for climate models application in a last section "Summary and implications for climate models" as follows:

[revised manuscript text omitted]

Schwarz, J. P., Samset, B. H., Perring, A. E., Spackman, J. R., Gao, R. S., Stier, P., Schulz, M., Moore, F. L., Ray, E. A., & Fahey, D. W. (2013). Global-scale seasonally resolved black carbon vertical profiles over the Pacific. *Geophysical Research Letters, 40*(20), 5542-5547. https://doi.org/10.1002/2013GL057775

Schwarz, J. P., Spackman, J. R., Gao, R. S., Perring, A. E., Cross, E., Onasch, T. B., Ahern, A., Wrobel, W., Davidovits, P., Olfert, J., Dubey, M. K., Mazzoleni, C., & Fahey, D. W. (2010). The Detection Efficiency of the Single Particle Soot Photometer. *Aerosol Science and Technology, 44*(8), 612-628. https://doi.org/10.1080/02786826.2010.481298

Shiraiwa, M., Kondo, Y., Moteki, N., Takegawa, N., Sahu, L. K., Takami, A., Hatakeyama, S., Yonemura, S., & Blake, D. R. (2008). Radiative impact of mixing state of black carbon aerosol in Asian outflow. *Journal of Geophysical Research: Atmospheres, 113*(D24). https://doi.org/10.1029/2008JD010546

Taylor, J. W., Allan, J. D., Allen, G., Coe, H., Williams, P. I., Flynn, M. J., Le Breton, M., Muller, J. B. A., Percival, C. J., Oram, D., Forster, G., Lee, J. D., Rickard, A. R., Parrington, M., & Palmer, P. I. (2014). Size-dependent wet removal of black carbon in Canadian biomass burning plumes. *Atmospheric Chemistry and Physics, 14*(24), 13755-13771. https://doi.org/10.5194/acp-14-13755-2014

Textor, C., Schulz, M., Guibert, S., Kinne, S., Balkanski, Y., Bauer, S., Berntsen, T., Berglen, T., Boucher, O., Chin, M., Dentener, F., Diehl, T., Easter, R., Feichter, H., Fillmore, D., Ghan, S., Ginoux, P., Gong, S., Grini, A., … Tie, X. (2006). Analysis and quantification of the diversities of aerosol life cycles within AeroCom. *Atmospheric Chemistry and Physics, 6*(7), 1777-1813. https://doi.org/10.5194/acp-6-1777-2006

Yang, Y., Fu, Y., Lin, Q., Jiang, F., Lian, X., Li, L., Wang, Z., Zhang, G., Bi, X., Wang, X., & Sheng, G. (2019). Recent Advances in Quantifying Wet Scavenging Efficiency of Black Carbon Aerosol. *Atmosphere, 10*(4), Article 4. https://doi.org/10.3390/atmos10040175

Yu, P., Froyd, K. D., Portmann, R. W., Toon, O. B., Freitas, S. R., Bardeen, C. G., Brock, C., Fan, T., Gao, R.-S., Katich, J. M., Kupc, A., Liu, S., Maloney, C., Murphy, D. M., Rosenlof, K. H., Schill, G., Schwarz, J. P., & Williamson, C. (2019). Efficient In-Cloud Removal of Aerosols by Deep Convection. *Geophysical Research Letters*, *46*(2), 1061-1069. https://doi.org/10.1029/2018GL080544

Yus-Díez, J., Bernardoni, V., Močnik, G., Alastuey, A., Ciniglia, D., Ivančič, M., Querol, X., Perez, N., Reche, C., Rigler, M., Vecchi, R., Valentini, S., & Pandolfi, M. (2021). Determination of the multiple-scattering correction factor and its cross-sensitivity to scattering and wavelength dependence for different AE33 Aethalometer filter tapes : A multi-instrumental approach. *Atmospheric Measurement Techniques*, *14*(10), 6335-6355. https://doi.org/10.5194/amt-14-6335-2021

Yus-Díez, J., Via, M., Alastuey, A., Karanasiou, A., Minguillón, M. C., Perez, N., Querol, X., Reche, C., Ivančič, M., Rigler, M., & Pandolfi, M. (2022). Absorption enhancement of black carbon particles in a Mediterranean city and countryside : Effect of particulate matter chemistry, ageing and trend analysis. *Atmospheric Chemistry and Physics, 22*(13), 8439-8456. https://doi.org/10.5194/acp-22-8439-2022

Zanatta, M., Gysel, M., Bukowiecki, N., Müller, T., Weingartner, E., Areskoug, H., Fiebig, M., Yttri, K. E., Mihalopoulos, N., Kouvarakis, G., Beddows, D., Harrison, R. M., Cavalli, F., Putaud, J. P., Spindler, G., Wiedensohler, A., Alastuey, A., Pandolfi, M., Sellegri, K., … Laj, P. (2016). A European aerosol phenomenology-5 : Climatology of black carbon optical properties at 9 regional background sites across Europe. *Atmospheric Environment*, *145*, 346-364. https://doi.org/10.1016/j.atmosenv.2016.09.035

Zanatta, M., Laj, P., Gysel, M., Baltensperger, U., Vratolis, S., Eleftheriadis, K., Kondo, Y., Dubuisson, P., Winiarek, V., Kazadzis, S., Tunved, P., & Jacobi, H.-W. (2018). Effects of mixing state on optical and radiative properties of black carbon in the European Arctic. *Atmospheric Chemistry and Physics, 18*(19), 14037-14057. https://doi.org/10.5194/acp-18-14037-2018

---

## Author Comment (AC3)

**Answer to referees**

**Higher absorption enhancement of black carbon in summer shown by two year measurements at the high-altitude mountain site of Pic du Midi Observatory in the French Pyrenees**

We thank the three reviewers for evaluating the manuscript and providing us constructive and useful comments. Referees have common concerns which we addressed to the best of our possibilities:

- In order to better highlight the objectives and the main results of the paper (1) we modified the title, (2) clearly listed the scientific questions addressed in the introduction section, (3) modified the conclusion into a section "Summary and implications for climate models", (4) moved Figure 2 from the meteorological section to the Supplement, (5) moved Figure S7 and Figure S9 from the Supplement to the main text in Sections 3.4.1 and 3.4.2, respectively.

- We clarified and completed the section on aerosol optical properties by adding a more complete description of the observed parameters and by reworking the design of Figures 3 and 5.

- We added two elements in the Supplement describing the processing of the SP2 measurements: a Section S1 comparing BC mass concentrations obtained from our software on PYTHON and the PSI SP2 Toolkit running on IGOR and a Section S2 to explain the processing of the rBC core size distribution. A Section S5 about the discrimination of Free Tropospheric/ Planetary Boundary Layer conditions has been added in the Supplement.

- We changed the "BC" nomenclature to "rBC" to follow the recommendations by Petzold et al. (2013) for measurements performed by a SP2.

Please find below reviewer comments in black and our responses in blue. The line numbers in the responses refer to the new version of the paper.

**Anonymous Referee #1**

SUMMARY

The work of Tinorua et al. provides interesting dataset on black carbon properties at a high-mountain site in Europe. These sorts of data are rare and of great interest. The aim of the manuscript is to understand the variability of black carbon properties as function of season, dynamics of the boundary layer and wet removal. Although the current dataset might allow investigating these processes, the presentation and discussion of the results prevents the authors to clearly communicate their message. The language and nomenclature are often problematic to the understanding of the text. Which require a thoughtful revision. Often, the authors jump to conclusions very fast, without a proper description of the observed parameters and with a superficial use of references. As a consequence, the processes leading to the observed changes of rBC properties are often unclear. I suggest the authors to clarify their goals, reduce to a minimum the non-essential discussion and elaborate more in details their hypothesis. I also advise caution when discussing "photochemical processes" and "hygroscopicity", which cannot be investigated with the current dataset. In its current status, the manuscript is not suitable for publication. However, I invite Tinorua et co-authors

to consider the major comments and a multitude of specific, yet not minor, comments for resubmission after major changes.

MAJOR COMMENTS

First, the manuscript would benefit from a deep revision of the language, which often results non-scientific and approximative. The authors are also invited to revise the format of citations, acronyms, and units and the grammar. See specific comments.

Nomenclature is extremely important. The authors should make sure to provide the correct information especially when using abbreviations and acronyms. 1) When dealing with optical properties, it is essential to always declare the wavelength. This is not always done, in the text and especially in the figures. In some cases, it thus results difficult to understand at what wavelengths the measurements are performed. 2)Every soot-measuring technique is based on different properties of the aerosol; hence, instrument specific nomenclature must be used. Soot measured with SP2 should be named rBC (refractory black carbon). Soot measured with filter-based photometers should be named eBC (equivalent black carbon). Soot measured with thermal-optical method should be named EC (elemental carbon). This nomenclature is not applied to the data presented here and to the results of other works. Please revise all the nomenclature and resulting abbreviation following Petzold et al. (2013).

**REPLY :**A deep revision of the manuscript has been done in order to correct typographical and linguistic errors as best as possible. The wavelengths of each optical parameters have been added in the text and the figures. The acronym "BC" has been replaced by "rBC" throughout the entire manuscript when talking about BC measured by the SP2.

The SP2 offers the possibility to quantify the mixing state (PSD detector and time-lag) and "composition" (colour ratio) of rBC. Unfortunately, these analyses are not performed, although it might help understanding ageing process and absorption enhancement, influence of different sources and potentially wet removal. Could the author explain why mixing-state and colour-ratio were not presented in the manuscript?

**REPLY :** We agree with Reviewer #1 that a closure between the rBC coating and absorption enhancement would reinforce our interpretation. However, due to a technical issue on the low gain of the scattering channel of the SP2, we could not provide rBC mixing state.

Concerning the color ratio, its analysis provides information on the presence of iron oxyde contained in dust particles (Liu et al., 2018; Moteki et al., 2017; Yoshida et al., 2016). To our knowledge, this parameter can not be used to assess the rBC composition.

The discussion of results and its interpretation is often superficial. This is particularly true in Section 3.1 and 3.2, where a detailed variability of aerosol and BC properties is provided but not discussed with the appropriate literature context. The text reads like a list of numbers followed by a list of references, while the reasons causing the variability is often explained with short and generic sentences like "*It has been attributed to the seasonal variation of the continental boundary layer height, long-range transport events (e.g. Saharan dust outbreaks, coal burning from eastern Europe) and biomass burning both from forest fires in summer and domestic heating in winter.*" I suggest the author rethinking all the results section to improve their data interpretation and to set clear scientific objectives.

The figures based on time series are not particularly helpful. If the authors aim to discuss the seasonal variability it is advisable to use a longer time stamp (1 month or 2 weeks). In order to provide evidence of correlations between the various properties I also suggest using scatter plots.

**REPLY:** Section 3.2 has been completed to better describe the seasonal variability of aerosol optical properties and explain the cause of this variability.

The sentence quoted by the reviewer aimed to provide the reasons causing the variability mentioned in the literature. This has been reworded.

The time resolution of Figure 3 and 5 has been decrease to provide a better visualization of the seasonal variations. We preferred to use boxplot in Figures 7, 8 and 10 instead of scatterplots for clarity reasons due to the large amount of data.

SPECIFIC COMMENTS

L30: Merge the two statements, not clear what "this" refers to.

**REPLY :** The statements were merged as suggested the reviewer.

L36-37: please add a reference.

**REPLY :** The reference of Matsui et al., (2018) has been added.

L39-40: the definition is correct, but it is not described how Mac is measured. A short description of the methodology is needed since later on (L43) the instrumental influence is mentioned.

**REPLY :** The following sentence has been added in line 39-41 :

" The $MAC_{BC}$ can be calculated either by dividing the measured absorption coefficient of BC by its mass concentration or by using Mie's Theory and the BC size distribution and coating thickness as input variables. "

L53: too many references, select the most relevant to deliver your message.

**REPLY :** We included numerous references to show the high quantity of studies highlighting the lensing effect, ie. the absorption enhancement of BC due to its coating.

L58: what it is meant with "multiplied by two"?

**REPLY :**The sentence has been replaced in lines 60-62 by:

" López-Moreno et al., (2014) have shown by running several regional climate models that the occurrence of winter warm events in the Spanish Pyrenees will gradually increase until 2080. This includes an increase in the number of warm days and nights and the number of snow/ice melting days at altitudes above 2000 m above sea level (asl). "

L66-76: part of this sub-paragraph can be moved into the methodology (ABL-Topoindex).

**REPLY :** We believe it is essential to mention in the introduction the topographic features influencing BC observations at the PDM site in order to introduce the specific scientific questions investigated in the paper.

Listing of the sections is not needed. I suggest rewriting the current paragraph focusing on the goals of your work.

A paragraph describing the main scientific questions of the paper was added at the end of the introduction:
"This paper aims to provide comprehensive picture of the seasonal and diurnal variability of rBC properties at PDM, and to explore the processes driving these properties. Specifically, in the indicated sections, the following questions are addressed:

1. What are the air mass transport pathways impacting PDM ?
2. What is the seasonal variability of aerosol optical properties and dominant aerosol types ? What is the specific contribution of rBC to aerosol absorption ?
3. How do the microphysical and optical properties of rBC vary on a seasonal and daily basis ?
4. What are the roles of wet deposition, source and transport pathway in driving rBC absorption ?"

L90: replace "sucked" with "sampled"

**REPLY :** This has been modified

L94: DMT is not based any longer in Boulder, but in Longmont

**REPLY :** This has been modified.

L93-113: Although being relatively tedious, nomenclature is important. BC measured via laser-induced incandesce technique is normally referred as rBC (refractory black carbon). I suggest reading Petzold et al. (2013) for more details. Considering this technicality, I also recommend the authors to replace "BC" with "rBC" in the text and in all abbreviations ($M_{rBC}$, $D_{rBC}$, etc…) when referring to their or other SP2 measurements. BC can be used for more generic discussion in the introduction.

**REPLY :** The nomenclature has been modified according to Petzold et al. (2013). A text in the beginning of Section 2.2.2, lines 104-108 has been added to present the different nomenclatures relative to BC:

"BC can be measured by different methods which are based on different BC properties. Petzold et al. (2013) defined a specific nomenclature for BC according to the BC quantification method. Following the recommendation of the authors, BC quantified by laser-induced incandescence and thermal-optical analysis will be referred to as refractive black carbon (rBC) and elemental carbon (EC), respectively. More general discussion on BC without focusing on its measurement technique will be referred to as BC."

L100: out of curiosity, did the authors ever compared the results obtained with the Python code and the SP2 Toolkit?

A new Section S1 has been added in the supplement about the SP2 data processing :
"**Text S1: Data processing for retrieving SP2 mass concentrations**

The Paul Scherrer Institute's SP2 toolkit is a softwave developed using IGOR to provide quantitative analysis of rBC mass concentration. However, this software is not suitable to analyze large amounts of data. During the PDM campaign, more than 1.2 To of data has been recorded. Processing it with the PSI toolkit would be too much time-consuming. This is why we developed a

software on Python. The data analysis was validated by comparing our $M_{rBC}$ to the one obtained by the SP2 toolkit.

Figure S1 presents a comparison between the $M_{rBC}$ retrieved with our Python program in blue, and the $M_{rBC}$ calculated with the PSI SP2 toolkit in red. The output $M_{rBC}$ from the toolkit does not take into account the rBC mass fraction below and above the SP2 size detection range corresponding to $90 < D_{rBC} < 580$ nm. Therefore $M_{rBC}$ without correction of the missing mass fraction is presented here. Globally over the 3 days, the two processing yielded $Mr_{BC}$ values in agreement taking into account the 14% of uncertainties on $Mr_{BC}$ (shaded areas), with a mean $M_{BC}$ of $101.1 \pm 14.2$ and $82.3 \pm 11.5$ ng m$^{-3}$ for our method and the SP2 toolkit, respectively. The SP2 toolkit seems to generate more $M_{rBC}$ peaks compared to our method, which smooths a bit more the time series. Such high peaks of $M_{rBC}$ don't seem realistic, given the situation of the site (remote station, without the presence of local rBC sources). The values provided by the PSI toolkit may be more noisy that the Python software due to a different selection of invalid individual signals. This could include signals exhibiting the maximum of their incandescence peak completely off-centered in the detection window of the SP2, incorrect sample flowrate, or an underestimation of the baseline of the incandescence peak height was underestimated, leading to overestimated individual masses,…). These different possibilities have not been explored in detail."

[Figure]

*Figure S1: 72-h comparison between MBC calculated with the PSI SP2 toolkit and MBC calculated with the Python program used in this study. Data was 10-min averaged on the period from 26th to 28$^{th}$ July, 2020.The top left panel shows $M_{BC}$ time series with the shaded area representing the $M_{BC}$ uncertainties, and the associated histogram on the right-hand side with the median $M_{BC}$ and its uncertainties represented by the solid and dashed lines, respectively. The bottom panel shows the bias ($M_{BC}$ from our processing minus $M_{BC}$ from the SP2 toolkit) and its associated histogram on the right-hand side. MBC data was here measured for rBC cores between 90 to 580 nm, without correction of the missing mass fraction.*

L102: I do not see an increase of mass concentration at diameter smaller than 90 nm in Figure S1. Please reformulate or verify the top panel of Figure S1. Figure S1 shows both mass and number size distribution, but only mass is described.

**REPLY :** We agree that we did not provide sufficient elements to understand the issue. We have reformulated the sentence and added a Section S2 in the Supplement to describe the processing of rBC size distribution measured by the SP2 and the estimated missing mass fraction.

The sentence in lines 118-119 was revised as :

"However, the observed size distributions showed that an important fraction (around 12%) of $M_{BC}$ at diameters below 90 nm is not measured by the SP2 (Figure S2 in the Supplement)"

The following text has been added to the Supplement :

**"Text S2 : Information about the rBC size distribution processing**

The SP2 measures rBC cores from mass equivalent diameters of 90 to 580 nm. Fig. S2 shows the two-year average of the daily rBC cores size distributions. It can be noticed in Fig. S2 that the number size distribution measured by the SP2 did not cover the full size range of rBC at PDM. This is particularly true for the rBC particles below 90 nm, where the major fraction of the $M_{rBC}$ was missed by the SP2. In order to estimate the missing rBC mass fraction undetected by the SP2 (e.g. the mass size distribution under 90 nm and over 580 nm), the daily rBC size distributions were fitted with a sum of three lognormal functions as :

$$M_{rBC} = dM/dln(D_{rBC}) = \sum_{i=0}^{3} \left( \frac{M_i}{\sqrt{2\pi}\ln(\sigma_{g,i})} \exp\left[ \frac{-\ln^2(d/d_{g,i})}{2\ln^2(\sigma_{g,i})} \right] \right)$$

with $M_i$, $d_{g,i}$ and $\sigma_{g,i}$ representing the rBC mass concentration, the geometric mean diameter and the geometric standard deviation of the mode i, respectively. The same function with two modes has been used to fit the number size distribution.

The fitting parameters were constrained in the following ranges : Mode 1 : $50 < d_{g,1} < 100$ nm and $1.2 < \sigma_{g,1} < 3$; Mode 2 : $150 < d_{g,2} < 250$ nm and $1.3 < \sigma_{g,2} < 2.9$; Mode 3 : $350 < d_{g,3} < 500$ nm and $1 < \sigma_{g,3} < 3$. "

L103: "detection range", not "detection window".

**REPLY :** This has been modified.

L106-113: Please define what "dg" and "σg" mean. Assuming these are the geometric mean and geometric standard deviation, how these were defined, empirically? For mode 1. The SP2 lower size quantification limit was 90 nm. Does it mean that the lognormal fit is applied to the 90-100 nm diameter range to derive mode1?

**REPLY :** A definition of "$d_g$" and "$\sigma_g$" has been added in the text.

Please find details on the rBC size distribution processing in the previous answer.

Figure S1 shows the size distribution of rBC, but on what time scale? With what temporal resolution was the MBC-correction calculated? Would it change during different conditions (PBL, FT, winter, summer, etc…)?

**REPLY :** Figure S2 (following the new numbering) shows the two-year average rBC size distributions to illustrate the missing mass fraction of BC undetected by the SP2. However in the paper we used the daily average rBC size distribution to quantify the missing mass factor and correct the $M_{rBC}$ measured by the SP2. The variations of the missing mass fraction with the season, dynamic conditions, etc. will be addressed in a paper in preparation for submission on AMT.

The relative standard deviation of the correction factor is approximately 90%, this lets me thing that non-negligible variability was observed during the measuring period. Could the authors have used a time dependent correction factor instead of constant one for the full dataset?

**REPLY :** Please refer to the previous answer. Lines 125-126 have been reformulated as follows:

"The average missing mass correction factor over the campaign was 1.2 ± 1.1 (Mean value ± STD)."

Considering the temporal variability of MBC-correction, I would like to see how MAC correlate with the correction factor.

**REPLY :** Figure R1 shows the $MAC_{rBC}$ as a function of the $M_{rBC}$ correction factor, noted $R_{fit/meas}$, equal to the ratio between the fitted mass concentration and the measured mass concentration. There is no particular correlation between the two.

[Figure]

*Figure 1: S8: MACBC as a function of $R_{fit/meas}$,core over the campaign.*
*Each point represents 1 day average data.*

L115: please provide the model, manufacturer, company, and country for the TSI instruments, as it is nicely done for the other instruments.

**REPLY :** The details for the TSI instruments have been added in the text.

L121: List the measuring wavelengths.

**REPLY :** This has been added.

L125: since is not yet published, the Cref value used in the present work should be described a bit better (location of the measurement, reference instrument, wavelength) and compared to previous studies. Since the manuscript is in preparation, and not submitted the year is not relevant.

**REPLY :** The following sentence has been added in Section 2.2.3, lines 143-145:

"The multiple scattering parameter used to correct the measured attenuation was set to 3.22, according to the value obtained at $\lambda$=880 nm by Yus-Díez et al. (2021) at the mountainous site of Montsec d'Ares located less than 200 km from the PDM ."

L126a: I strongly do not recommend the use of "MBC" for the BC mass concentration derived from the aethalometer data. First, the correct nomenclature should be equivalent black carbon (eBC; Petzold et al., 2013). Second, the mass concentration derived from SP2 measurements is also abbreviated as MBC. As a result, it become tremendously confusing to understand how MB is derived in the rest f the paper. Update the use of nomenclature.

**REPLY:** We changed the nomenclature used for aethalometer measurements by "mass concentration of equivalent black carbon"

L126b: Were the MeBC and $\sigma$ap limits corrected with Cref? At what wavelenght these values were derived, this is particoularly important (especially for $\sigma$ap). If I take 0.0215 Mm−1 and 0.005 μg m−3 I obtaine a MAC (or a mass attenuation coefficient) of 4.3 m2/g, please revise these values. And set the limit of AE33 based on absortion coefficient rather than MeBC, since you have a more reliable instrument (SP2) to measure the mass of rBC.

**REPLY :**

The corresponding line has been corrected in lines 146-147:

"The detection limit of the aethalometer is 0.039 Mm$^{-1}$ (corresponding to an equivalent black carbon mass concentration of 0.005 μg m$^{-3}$). "

L130-132: when providing the information about the instruments try do be consistent with the rest of the paper and provide (model, manufacturer, company, and country), as done for the aerosol instruments

**REPLY :** These information have been added for all the instruments.

L133 I suggest removing ΔBC/ΔCO in this section, since it comes out of the blue without any context and it is anyway explained later in the text.

**REPLY :** $\Delta M_{rBC}/\Delta CO$ has been removed from this section as suggested by the reviewer.

L144: The authors should explain clearly that AAE was calculated between 450-635 nm to match the wavelength range of the Nephelometer. Since the measuring wavelengths of the Aethalometer are not listed, it becomes harder for the reader to understand why $\sigma$ap660 was adjusted to 635 nm.

**REPLY :** We agree that there were missing elements about the method used to calculate AAE between 450 and 635 nm. Therefore, we have added the measurement wavelengths of the aethalometer in Section 2.2.3 and we have completed the text in Section 2.3 as follows, lines 161-165:

"The spectral dependence of $\sigma_{ap}$ was characterized by the Absorption Angstrom Exponent ($AAE_{aer,450-635}$) calculated between 450 and 635 nm as follows :

$$AAE_{aer,450-635} = \frac{-\log\left(\frac{\sigma_{ap,450}}{\sigma_{ap,635}}\right)}{\log\left(\frac{450}{635}\right)}$$

For this calculation, $\sigma_{ap,470}$ and $\sigma_{ap,660}$ from the aethalometer were adjusted at the wavelengths of 450 and 635 nm measured by the nephelometer using the AAE calculated from the aethalometer between 370-470 nm and 590-660 nm. "

L141-151: I believe a short explanation on what these optical properties represent is needed here. SSA, What SSA, AAE and SAE represent, why they are climatically relevant?

**REPLY :** We thank the reviewer for these suggestions and have added an explanatory sentence after formulas (1) and (2) to explain the meanings of AAE, SAE and SSA respectively :

l 165-169 : "$AAE_{aer,450-635}$, provides information about the chemical composition of atmospheric aerosols. Pure BC absorbs radiation across the whole solar spectrum with the same efficiency; thus, it is characterized by $AAE_{aer,450-635}$ around 1 (T. C. Bond et al., 2013). Conversely light-absorbing organic particles called as brown carbon (BrC) or dust particles typically have $AAE_{aer,450-635}$ greater than 2 (Bergstrom et al., 2007; Schuster et al., 2016; H. Sun et al., 2007).

L 177-179 : "$SAE_{aer,450-635}$ describes the relative contribution of fine and coarse mode particles (Clarke & Kapustin, 2010). Small values of $SAE_{aer,450-635}$ indicate a higher contribution of large aerosol particles (e.g. dust and sea salt) , while large values of SAE indicate relatively smaller aerosol particles (Cappa et al., 2016)."

L 182-183 : "$SSA_{aer,\lambda}$ describes the relative importance of scattering and absorption to the total light extinction. Thus, it indicates the potential of aerosols to cool or warm the atmosphere."

L155-157: I do not agree with the nomenclature choice. If $\Delta BC/\Delta CO$ is the ratio of MrBC over $\Delta CO$, it should be simply called MrBC/$\Delta CO$, as done by previous studies cited in the result section (Liu et al., 2010; McMeeking et al., 2010).

**REPLY :** We understand the reviewer's point of view. The nomenclature of $\Delta BC/\Delta CO$ has been changed to $\Delta M_{rBC}/\Delta CO$ . We decided to keep the "$\Delta$ "because $\Delta M_{rBC}$ refers to the excess rBC mass concentrations, ie. above the background levels. In our particular case of a remote site without consequent rBC sources, the background levels of $M_{rBC}$ are negligible and this why we can directly use $M_{rBC}$ as $\Delta M_{rBC}$.

L160: MBC under (resp. over) 160 the 5th (resp. 95th) percentile? Rephrase.

**REPLY :** The sentence in l. 195 has been corrected as follows :

"$M_{rBC}$ below the 5th and above the 95th percentile were filtered before $MAC_{rBC}$ calculations to reduce the influence of outliers in statistical analyses."

L161: I suggest giving more explanation about the influence of dust on absorption. Often, the authors do not provide adequate context to very specific statements, assuming that every reader has a deep knowledge of the treated topic.

**REPLY :** Some details has been added in Section 2.3, l. 196-198 :

"In addition, we filtered out periods when dust were sampled at PDM for the calculation of $MAC_{rBC}$ since Yuz-Diez et al. (2021) observed significant biases in the multiple scattering correction of the aethalometer AE33 during such events."

L167-170. Provide some references for each method.

**REPLY :** The following references were added : Cappa et al. (2012), Healy et al. (2015), Shiraiwa et al. (2010) for the thermodenuder method, Cappa et al. (2019), Yuan et al. (2021) for the extrapolation of $MAC_{BC}$ as a function of the BC coating mixing ratio, and Liu et al. (2017) and Zanatta et al. (2018) for Mie calculations.

L171: correct "1,95" in "1.95". Moreover, I strongly recommend reading Liu et al. (2020), who showed that, despite being widely used, 1.95-0.79i might not be representative of realistic condition. The authors are invited to verify the sensitivity of their calculated MACbare as function of different refractive index. As a matter of fact, Figure S3 showed a maximum MACbare below 5 m2/g which considerably lower than MAC of fresh and bare Bc presented by Bond (7.5m2/g). I imagine that Eabs presented here might be overestimated.

**REPLY :** The widely used $MAC_{bare,BC}$ value of 7.5 m² g$^{-1}$ recommended by Bond & Bergstrom (2006) is provided at 550 nm. If we assume an AAE of pure BC equal to 1, this corresponds to a $MAC_{bare,BC}$ around 4.7 m² g$^{-1}$ at 880 nm, which is in the range of the values shown in Fig. S3.

L183-195: for non-expert readers, this subsection might result of difficult understanding. Since the analysis is important, I suggested providing more details on how the ranking is calculated (more technical aspects could go in the supplementary). As it is, FigureS4 does not really help understanding the anabatic ranking, since zero context is provided in the supplementary.

**REPLY:** The text concerning the ranking method in the main body of the paper has been simplified for non-expert readers. In compensation, further detail on the ranking procedure, the determination of the threshold rank, and the interpretation of the "anabatic radon" diagnostic, is now given in the Supplement, as a text accompanying Fig.S5 (formerly Fig. S4). However the procedures are not described in full detail since this would be strictly redundant with the original description by Griffiths et al. (2014), which is given as reference.

Text part of the Section 2.4, l. …:

"To discriminate FT and PBL-influenced air masses (hereafter referred as PBL/FT conditions), we followed a methodology proposed initially by Griffiths et al. (2014), assuming that the diurnal radon increase, which is typically observed at mountain sites during the daytime, is the result of transport of PBL air by thermal anabatic winds up to the summits. The method first consists in ranking the days of the sampling period by decreasing anabatic influence (details in the Supplement and in Griffiths et al. (2014)). A threshold rank (here 282, see Fig. S5 and associated text in the

Supplement) can then be determined to separate days with or without anabatic influence in the daytime.

In our study, it was necessary to select the observation hours strongly influenced by the boundary layer. To do this, we selected the first 200 anabatic days in the ranking, and from these days, all the hours with radon activity was greater than the median value for the current day.

We also needed an ensemble of observation hours with minimum influence of the PBL. In the latter case, we selected hours in the non-anabatic days (i.e. ranked after 282) with radon activity below the median value for the current day."

Text S5 in the Supplement:

"In the present study, the method used to discriminate anabatic vs. non-anabatic days follows the method by Griffiths et al. (2014) based on radon measurements, and the recognition that the anabatic influence can be measured by the amplitude of a diurnal radon cycle, properly phased with a maximum in the afternoon. The method mainly consists in ranking days by decreasing anabatic influence. All details of the ranking algorithm are given in Griffiths et al. (2014), but in overview (citing the authors) "the procedure involves computing the diurnal composite of the set of all observed days and then removing days from the set in the order which most quickly reduces the mean square amplitude of the set's composite diurnal cycle.

In our study, the procedure to compute the ranking strictly follows the steps described in Griffiths et al. (2014), except on this only point: as input data for the ranking procedure, these authors use the absolute deviation between the hourly radon data and the daily mean of the current day. In our case, we alternatively used the relative deviation (i.e. the absolute deviation normalized by the current-day mean). This considerably improved the result because the radon regional background at PDM is suspected to be much more variable than at the Jungfraujoch (see discussion below).

Then, a diagnostic value called "anabatic radon" is calculated for each day, which represents (in short) the average deviation of radon above a nocturnal background (see full detail of the calculation in Griffiths et al. (2014), which we again strictly followed). Anabatic radon mostly decreases with increasing anabatic rank (Fig. S5), at least up to a threshold rank (282) corresponding to its first minimum. Days ranked before this threshold are considered as anabatically-influenced (or more simply called "anabatic days"), and the days after this rank are considered as non-anabatic.

After this rank, anabatic radon values should expectedly be zero. This is obviously the case neither in Fig. S5, nor in the similar graph by Griffiths et al. (2014 – their Fig.3). The reason is that intraday radon variations due to any reason but anabatic transport, may occur out of phase with the thermally-driven cycle. Because of incoherent phasing, such variations contribute little to the set's composite diurnal cycle, and as a consequence, such days appear far in the ranking. But such variations may nevertheless be above the background (i.e. the minimum value) of the current day, and produce non negligible values of the "anabatic radon" diagnostic – which has thus little sense for non anabatic days.

A question arises, however, why anabatic radon appear more noisy in our Fig. S5 than in the similar graph in Griffiths et al. (their Fig.3). We have no definitive explanation to this, but may speculate that radon sources at the regional scale around Pic du Midi are more heterogeneous and intense than around the Jungfraujoch. Supporting this idea are radon exhalation maps by soils presented in

(Karstens et al. (2015) or (Quérel et al. (2022), showing radon hot spots in the western Iberian peninsula, in the French Massif Central, and (to a lesser extent) locally in the Pyrenees. In such conditions, the radon background at Pic du Midi may be much more variable than at JFJ, and other transport processes than anabatic transport may thus contribute more strongly to radon variability at PDM. This would deserve specific investigation, but is out of the scope of the present study."

[Figure]

*Figure S5: Daily anabatic radon as a function of the day anabatic rank (see text for details). Each dot represent an observation day ranked from the most anabatically-influenced day (left) to the least one (right). The vertical pink line represents the cut-off rank before which days can be considered as PBL-influenced. The insert shows diurnal composites of radon activity anomaly according to different ranges of ranks, using the same color code as in the main plot.*

L191: I find the note particularly disturbing. Please avoid statements like "make no sense".

**REPLY:** This note actually brought little information but was confusing. It has been removed in the revised manuscript.

The fluctuation after rank 282 are not negligible and more noisy than in Griffiths et al. (2014). Please try to argue what might be the natural causes leading to the radon fluctuation. Could it be that these values are false negatives? Could the radon ranking be verified as function of water vapour as done in Griffiths et al. (2014)?

**REPLY:** A discussion on this specific point has been added in the text in the Supplement accompanying Fig. S5. This method may unlikely produce false negatives, because radon variations in phase with the thermal cycle should positively contribute to the amplitude of the set's composite

cycle. Such days should thus fall in the upper part of the ranking. But for the same reason the method is subject to false positives, when radon variation caused by any reason but anabatic transport fall in phase with the thermal cycle.

The ranking method was applied at PDM with water vapor by Hulin et al. (2019), in absence of radon data for their studied period. They obtained results (their Fig. 8) that were similar to ours in Fig. S5, in term of both (normalized) threshold rank and shape of the anabatic composite cycles. But radon at PDM has less seasonal and day-to-day variability than water vapor, and is thus less ambiguous as BL tracer than water vapor.. Griffiths et al. also pointed that the algorithm applied to water vapor is more subject to false positives.

L192-195: I am not sure to properly understand this final selection. The periods under the influence of PBL presented later are based on hourly selection and not daily selection (for ranking below 200 in the "anabatic-subset"), right? The opposite was done for FT influence. I expect the PBL-periods to occur preferentially during day-time, while FT-periods during night-time. Is the analysis only considering day-time or it does include also night-time?

**REPLY :** The text on the selection of either BL- or FT-influenced observation hours has been rephrased and, we hope, clarified. In summary, we made two distinct hourly selections among two previous  selections of either anabatic of non-anabatic days. As BL-influenced hours are defined as hours with radon above the median of the current day among a selection of most anabatic days, these hours will clearly fall in the daytime. In contrary,  FT-influenced hours may fall either during the day of the night. This is illustrated by Fig. R2.

[Figure]

*Figure 2: Density of FT/PBL classified hours*

L202: m.s−1. Remove the dot.

**REPLY :** This has been modified.

L223-230: SSA at what wavelength? In figure 4a there are values well below 0.93. Is a monthly minimum, a season minimum? Please explain better. The simultaneous increase of SAE and absorption does not automatically indicate that absorbing particles are small in size. It must be kept in mind that BC is co-emitted with other fine aerosol species such as sulfate. I fund however interesting that the maximum peak of absorption does not correspond with a minimum of SSA. The reasons beyond the seasonal variability are actually not explained ("It has been attributed to the seasonal variation of the continental boundary layer height, long-range transport events and biomass burning both from forest fires in summer and domestic heating in winter." is a very generic statement).

**REPLY :** We agree with these very useful remarks which helped us for the interpretation of the aerosols optical properties. We have modified Figure 3 in order to better highlight the monthly variation of aerosol optical properties. In addition the scattering coefficient measured at 450, 525 and 635 nm have been added to Figure 3 in help in the interpretation of the SSA variations.

The text has been changed as follows, lines 270-285:

"There was a clear seasonality of aerosol optical properties. SSA at the three wavelengths exhibited the lowest monthly mean values in spring-summer ($0.94 \pm 0.02$ at $\lambda = 525$ nm) and the highest in autumn-winter ($0.99 \pm 0.01$ at $\lambda = 525$ nm, as shown in Fig. 3a). Simultaneously, the highest monthly mean SAE values were observed in spring-summer ($1.23 \pm 0.70$ ) and reached a minimum in the winter ($-0.25 \pm 0.16$) (Fig. 3b). This anticorrelation suggests a higher fraction of absorbing and fine particles relative to purely scattering and coarse particles at PDM during the spring-summer. Interestingly different trends can be observed between the summer and spring seasons. During spring 2019 the decrease of SSA correlated with a slight enhancement of $\sigma_{ap,880}$ (Fig. 3d) and decrease of $\sigma_{sca}$ at all wavelengths. In summer the increase of $\sigma_{ap,880}$ lead to values multiplied by a factor of four, while both SSA and SAE remained rather constant. All these parameters combined indicate a similar dominant aerosol type reaching PDM but with stronger contribution in summer. This is further confirmed by the simultaneous increase of $M_{rBC}$ in summer shown in Fig. 5.

This noteworthy seasonality of aerosol optical properties has previously been observed at other high mountain sites in Europe (Andrews et al., 2011; Collaud Coen et al., 2011; Laj et al., 2020; Pandolfi et al., 2018). The higher concentration of small and absorbing particles in summer at PDM could be attributed to a higher anthropogenic BC influence favored by strong vertical mixing and a higher PBL height, a higher occurrence of wildfires emitting large amounts of BC and Brown Carbon (BrC), or a lower precipitation rate.

In order to investigate these different hypotheses, a classification of the dominant aerosol type sampled at PDM was performed by using the spectral dependency of aerosol optical properties."

L259: check reference format

**REPLY :** This has been modified.

L260: why "MBCs"?

**REPLY :** This has been modified.

L260: Jungfraujoch name.

**REPLY :** It is a typo error and this has been modified.

L260-265: The seasonal variability of BC mass and absorption is opposite to background and polluted stations, where higher values are observed during winter compared to summer (among others: Yttri et al., 2007; Zanatta et al., 2016). The authors should explain this difference and potentially exploit it to introduce the analysis performed in the following sections of their works.

**REPLY :** In several mountainous sites of Jungfraujoch, Zugspitze-Schneefernerhaus (ZSF), the patterns of BC mass and absorption also shows minimums during winter and maximum during summer (Motos et al., 2020; Sun et al., 2021). The elevated sites presented by Zanatta et al. (2016) also exhibited maximum BC concentration and absorption during spring or summer.

The paragraph comparing $M_{BC}$ and $MAC_{rBC}$ has been modified and completed in Section 3.3, l. 358-378:

"The ambient $MAC_{rBC}$ was around $9.2 \pm 3.7$ m² g⁻¹ at $\lambda$=880 nm (Fig. 5d). Several studies previously reported $MAC_{rBC}$ values between 8.9 to 13.1 for measurements at $\lambda$=637 nm in European mountain stations (Pandolfi et al., 2014; Yus-Díez et al., 2022; Zanatta et al., 2016). By using a AAE of unity, these values can be converted to $MAC_{rBC}$ between 6.4 and 9.5 m² g⁻¹ at $\lambda$=880 nm. These studies used different measurement techniques, analysis method and correction factors from ours for estimating $MAC_{BC}$ that makes difficult the comparison of $MAC_{rBC}$ derived from different instruments. Pandolfi et al. (2014) performed a linear regression between $\sigma_{ap,637}$ values measured by a Multi-Angle Absorption Photometer (MAAP) and daily $M_{EC}$ values from off-line filter-based measurements by a SUNSET OCEC Analyzer. Yus-Díez et al. (2022) and Zanatta et al. (2016) retrieved $MAC_{BC}$ with these instruments by calculating the ratio between the two parameters instead of a linear regression. Because of the absence of a standard method for quantifying $M_{BC}$, the absolute uncertainties on the $MAC_{BC}$ obtained in the literature are very high ranging from $\pm30$ to 70% (Zanatta et al., 2016).

In terms of seasonality we found systematically higher values of $MAC_{rBC}$ in summer (monthly mean $\pm$ STD of 10.3$\pm$ 3.3) compared to winter (8.3 $\pm$ 3.8). Similar seasonal pattern was observed in Europe at Puy de Dôme (central France) and at Jungfraujoch (Swiss Alps) mountain sites (Motos et al., 2020; J. Sun et al., 2021; Zanatta et al., 2016). An opposite trend was observed at mountain sites affected by strong precipitation during monsoon such as the Tibetan Plateau and Himalayas regions where both $MAC_{BC}$ and $M_{BC}$ exhibit maximum values in winter or autumn (Srivastava et al., 2022; Zhao et al., 2017). The same seasonal pattern with elevated values in winter/autumn compared to summer/spring was observed at several rural and urban sites in the PBL, which was attributed to greater emissions from residential heating combined to a lower PBL height (Kanaya et al., 2016; Yttri et al., 2007; Zanatta et al., 2016). However, maximum of $MAC_{BC}$ and $M_{BC}$ concentration in the PBL during cold periods is not a recurring observation even for a same measurement site. For instance J. Sun et al. (2022) showed in Beijing (China) that, due to the reduction of some predominant BC sources in winter consecutive to environmental policies, the annual cycle of $M_{BC}$ changed over the years between 2012 and 2020."

L268: "Seasonal differences between the origin of highest MBC are thrown into relief,"…not sure what it is meant here.

**REPLY :** This sentence has been deleted.

L273: discussion discussed. Avoid repetitions.

REPLY : The sentence was changed and now reads:

« Further discussion on the role of PBL influence on $M_{BC}$ will be addressed in Section 3.4.2. »

L268-272: From my point of view, Figure 7 shows that 1) the wind patterns are similar in winter and summer; 2) high MBC are associated with low wind speed; 3) and that there is a north scarred signal in winter and southern signal in summer. With the MBC scale and so many points, I cannot identify any clear correlation between wind direction and BC concentration, so I do not agree with the statement "highlighting different BC geographical sources" Similar reasoning can be done for Figure S5, where the overall origin of the air masses lays in the same western sector in both seasons. To improve the visualization and interpretation of the data, I suggest organizing the wind direction in broader classes (10-20 degrees) and normalize the MBC to its maximum. This modification might help identify a correlation between wind direction and BC concentration

REPLY : We are very grateful to the reviewer for his suggestion concerning the polar graph. Instead of plotting the 1-hour data on the windrose, we partitioned the windrose and calculated the density of the $M_{rBC}$ data for every wind speed-direction 'bin', weighted by the $M_{rBC}$ values. Thus, the color of each point represents the density of $M_{rBC}$ measurements and their values.

We added a description of the representation and modified our interpretation in l. 321-331 as follows :

"Figure 6 shows bivariate polar plots obtained by combining wind analysis and $M_{BC}$ with 1-hour time resolution in winter and summer. The densities of $M_{rBC}$ data weighted by $M_{rBC}$ values, and normalized by the maximum $M_{rBC}$ were plotted as a function of wind direction and speed. The darkest areas of the wind pattern are those where the highest $M_{rBC}$ was measured with a high occurrence, whereas lightest zones exhibit lowest measured $M_{rBC}$ and/or a little occurrence of measurements. Note that locally emitted pollution at the measurement station was filtered before the analysis, limiting local $M_{rBC}$ contributions emitted from the PDM station (i.e. section 2.3).

In summer, the highest $M_{rBC}$ values were mainly associated with moderate wind speeds (above 5 m s$^{-1}$) and from the west and south west, suggesting a dominant regional transport (Fig. 6a). By contrast in winter (Fig 6b), the highest $M_{rBC}$ occurred mainly under more static atmospheric conditions (ie. for wind speeds below 5 m s$^{-1}$) and no evident wind direction dependency. These results suggest that local-scale emissions could be a major contributor to $M_{rBC}$ in winter unlike summer. Further discussion on the role of PBL influence on $M_{rBC}$ will be addressed in Section 3.4.2"

L270: please define summer and winter, this applies elsewhere in the text.

REPLY : A sentence has been added at the beginning of the results sections as follows:

"In the following, seasons are defined as follows: winter (December, January, February), spring (March, April, May), summer (June, July, August), and autumn (September, October, November) "

L275-285: So, what it is the conclusion of this analysis?

REPLY : Separate paragraphs has been created in Section 3.3 to better highlight the conclusions.

L320: All paper is based on winter and summer differences. I suggest removing non-essential information like the daily cycle in autumn and spring. Considering that little to no explanation is

given about the diurnal-seasonal change, I cannot fully understand the relevance or the aim of this analysis. As said already, the authors should try to motivate the observed variability, giving context and explanation. Section 3.3 suffers, in its entirety, of this problem.

**REPLY :** The purpose of Section 3.3 is to describe the diurnal and season variation of BC properties, whereas Section 3.4 aims to investigate more deeply the causes of the observed variability.

As shown above, we added the main questions addressed in each section at the end of Section 1. In addition a sentence has been added at the end of section 3.3, lines 403-404, to explain the focus on winter and summer season in the following sections:

"Particular attention will be paid to winter and summer because these seasons differ greatly, whereas spring and autumn appear intermediate."

L325:Section 4 is still part of the results, right? So it should be Section 3.x. Please correct.

**REPLY :** We modified the name of this section by: "3.4 Investigation of factors influencing rBC properties"

L327: As it is shown in the following section, BC mass concentration and BC/CO ratio drastically change (at least in winter) due to anabatic injection from the PBL. Under the influence of PBL injection of fresh BC, wet removal has a smaller impact of BC properties compared to free tropospheric conditions. If the authors excluded periods affected by precipitation in Section 4.2, period under the influence of PBL should be excluded here. This additional filter will reduce the number of atmospheric variable and, perhaps, improve the interpretation of the results.

**REPLY :** We thank the reviewer for his suggestion. We agree with the reviewer that filtering periods under PBL influence is more rigorous to exclude the potential impact of PBL injections on the higher observed $\Delta M_{rBC}/\Delta CO$ when the airmass did not undergone precipitation events and with a low humidity. We applied this filter and changed the Figure 8 accordingly. The new figure is similar as the previous one, leading to the same interpretation.

L356: biomass burning influence. rBC emitted by different sources might show a difference in properties. If the biomass plumes were fresh, the authors should be able to see a difference in the size distribution, and, potentially, in the colour-ratio (ratio of BB over NB channel of the SP2).

**REPLY :** As explained above, the color ratio only provides information on the presence of dust particles (D. Liu et al., 2018; Moteki et al., 2017; Yoshida et al., 2016). To our knowledge, the color-ratio can not be used to assess the rBC composition.

We agree with the reviewer concerning the influence of the age of rBC from biomass burning emissions on its size distribution. However, no significant rBC size distribution variability between the seasons has been observed despite the different $\Delta M_{rBC}/\Delta CO$. This can be explain since even if the PDM was under the influence of different BC sources throughout the campaign, rBC from long-range transport appears to be the dominant source of BC and this is reflected in the globally constant rBC size distributions. Furthermore, the time resolution of the rBC size distribution (of a day) does not allow to see the impact of PBL/FT conditions (1-h of time resolution)."

L357: Be consistent with cross-references…heather is "Fig.X" or "Figure X"

**REPLY :** This was modified.

L362: this is most likely due to the lower concentration of BC observed in the PBL in the summer period.

**REPLY :** Lines 458-463 have been changed in line with the reviewer's remarks and now reads:

"Surprisingly the thermally driven PBL injection did not significantly impact $M_{rBC}$ measured at PDM (Figure 10d). This contrasts with our winter observations and most previous surface measurements at mountain sites, where the daytime PBL development has been shown to enhance aerosol mass concentration (Herrmann et al., 2015; Venzac et al., 2009). The summer $M_{rBC}$ values at PDM are twice as high as than those observed in winter, which indicates a massive additional regional transport of rBC in the FT and a lower contribution of rBC from PBL injection. "

L348-390: why the size distribution of rBC is not shown here? It might help with the data interpretation..

**REPLY :** Providing BC size distributions require extensive processing of the raw SP2 data. We have managed to analyze two years of SP2 measurements (representing more than 1.2 To of data) by using a time resolution for processing BC size distributions of one day. This time resolution do not allow to investigate the influence of PBL/FT on BC size distribution.

L369: what is "this evidence". Reduce the use of "this", it makes difficult to understand what the authors refer to.

**REPLY :** The sentence of lines 467-470 has been changed and now reads :

"The high rBC loading transported in the FT, coupled with the higher $\Delta M_{rBC}/\Delta CO$ observed in the summertime (Fig. 10d and e),could be due to a strong influence of biomass burning emissions on the background FT in Europe."

L371: IAGOS…Always explain every abbreviation

**REPLY :** The signification of IAGOS has been added.

L375-379: long unclear sentence, rephrase.

**REPLY :** The sentences in lines 468-471 have been modified and now reads :

"The majority of trajectories reaching PDM in summer have crossed the Iberian Peninsula and, previously, North Africa and North America (Fig. S7 in the Supplement). In these regions large fire events frequently occur, which may explain the high concentrations of strongly absorbing rBC observed at PDM during summer."

L384-390: Are the SMPS data filtered for FT and PBL conditions? Please specify. If this is not the case, PBL aerosol injection might potentially explain the concentration increase of smaller particles (PBL influence timing is exactly the same FigureS8). Overall, the statement is mostly speculative since the authors cannot prove the occurrence of coagulation and condensation on rBC cores. I thus would not call it "evidence" but rather "hypothesis". Moreover, the SP2 is capable of providing coating thickness (via the position sensitive detector) and a simpler proxy for mixing-degree (scattering- incandescence time lag). Could the authors explain why these two analyses were not applied?

**REPLY :** Due to a technical issue on the low gain of the scattering channel of the SP2, we could not provide BC mixing state measurements.

**"Evidence"** has been replaced by "hypothesis" and the end of Section 3.4.2 (lines 485-490) has been modified to focus on the shift of the dominant mode of the aerosol size distribution, which was potentially due to the condensation of preexisting particles on BC particle during the morning:

"At PDM the enhanced $E_{abs}$ at noon was accompanied by a shift of the aerosol accumulation mode towards larger sizes, which may be due to the condensation of species on aerosol particles (Figure S10 in the Supplement). Simultaneously, a strong elevation of particle number concentration in the diameter range 10-30 nm can be observed, revealing new particle formation most likely produced by photochemical reactions at this time of the day. It is thus possible that rBC particles became more coated via condensation of species produced by photochemical reactions at noon. However, it cannot be ruled out that the evolution of aerosol size distribution is a poor indicator of the rBC mixing state."

L399-403: In the present work no evidence is provided on interaction with snow, coating thickness, lifetime, condensation rate or gaseous precursors. Only results obtained by the present study should be discussed in the conclusion section. This part is mostly speculative and I suggest removing it.

**REPLY :** The aim of this paragraph is to present our result on mean $E_{abs}$ values. We think it is important to remind readers the importance of $MAC_{rBC}$ for climate issues and put our results in the broad literature context.

L405: I would like to see if these results might change by removing the PBL periods.

**REPLY :** The remark has been answered above.

L406-407: avoid the use of references in the conclusions. Especially 4 in a row.

**REPLY :** References in the conclusion have been removed.

L415: What is the "evidence" exactly. Please elaborate.

**REPLY :** "evidence" has been replaced by (Lines 20-21) : "Combining $\Delta M_{rBC}/\Delta CO$ with air mass transport analysis, we observed additional sources from biomass burning in summertime leading to higher $M_{rBC}$ and $E_{abs}$."

L423-427: ageing time scale and its impact on cloud activation and optical properties of BC is not treated in the present work. Saying that wet removal is independent from size and mixing state, and that hygroscopicity is not treated properly in models is a bold statement…I recommend caution. Same goes for the following statement.

**REPLY :** A Section 4 has been added which discuss on the implication of the results for climate models.

F1: This figure might benefit some editing. Besides the low resolution, I suggest removing the picture (although beautiful) and introduce a double map with a continental and regional scale. More info could be provided within the figure such as coordinates, altitude, managing institute, ACTRIS name, station type (mountain, background…), instrument list…

**REPLY :** Figure 1 has been edited. It now has a better resolution, and regional and local scales maps has been added.

F2: I do not think that Figure 2 is needed. The text in section 3.1 describes well enough the general meteorological conditions. Since day-by-day variability is not discussed (and there is no need), I suggest removing the full figure

F6: Figure 6, as Figure 2 and 4, suffers from the choice of using a daily temporal resolution. Since the authors are mostly discussing the seasonal variability, a longer time scale (month) will help visualizing the seasonal changes.

**REPLY :** We thank the reviewer for his suggestions.

The figure 2 has been moved in Supplementary materials and its temporal resolution has been reduced.

The temporal resolution in Figures 2, 4 and 6 has been reduced to monthly median values.

F10b-e: axis is Eabs, caption is MACbc, correct.

**REPLY :** This has been corrected.

FS6: I suggest plotting this graph with daily or weekly temporal resolution.

**REPLY :** The temporal resolution of Figure S6 has been changed to a daily time resolution.

FS9 Shouldn't the points have the same colour in the top and bottom panels?

**REPLY :** The color code is already the same for the top and bottom panels. There is a very slight difference of colors between the two panels due to different amount of $E_{abs}$ and $\Delta M_{rBC}/\Delta CO$ data, leading to different mean wind direction for each hour .

FS10: usually nucleation mode is defined as D<10 nm

**REPLY :** "Nucleation mode" was replaced by "Lower Aitken mode".

---

## Author Response (AR2)

**Answer to the referee**
**Higher absorption enhancement of black carbon in summer shown by two year measurements at the high-altitude mountain site of Pic du Midi Observatory in the French Pyrenees**
**(Preprint egusphere-2023-570)**

Please find below the reviewer's comments in black and our responses in blue. The line numbers in the responses refer to the new version of the paper.

I am glad that Tinorua and co-authors incorporated some of my previous suggestions. In some parts, the manuscript's quality and readability improved since the first submission. However, this version still suffers from very similar flaws compared to the first draft. The presentation of the results is chaotic since there is little cohesion between the text and the figures. I notice an overall superficiality in justifying technical choices, presenting the data and interpreting the results. Finally, the results are not clearly summarized in the conclusion section, which is a very confusing mix of results, references and speculations. Considering the number of specific comments that can be found as follows (editing, grammar, nomenclature, sequence of figures, etc…), I have the impression that the manuscript was not carefully controlled before resubmission. Despite the high interest in the dataset, I cannot recommend the publication of the manuscript in ACP. I leave the choice of full rejection or resubmission after major changes to the editor, waiting for the pending reviews from other referees.

We thank the reviewer for his deep work of revision. A large part of the comments deal with the SP2 data processing to retrieve the rBC mass concentration. We acknowledge that the PSI SP2 toolkit is the most common tool used to process SP2 data. However, for the purposes of our study, a Python-based treatment process was developed, in order to meet our specific needs (processing very large quantities of data, transparency of data processing, using Linux,...). Other studies like Taylor et al. (2015)  have also developed their own data processing. Although it would be interesting to make an inter-comparison of the different methods, this is not the aim of this paper.
Before the second submission, the paper was reviewed by all the co-authors and we did our best to provide the most accurate version possible.

MAJOR COMMENTS

MULTIPLE SCATTERING CORRECTION
The C value was changed from 3.63 to 3.22 according to Yus-Díez et al. (2021). Considering the importance of this change and the direct effect on absorption coefficient and MAC, the choice should be explained better, and should not be based, solely, on the fact that Montsec d'Ares is a mountaintop site 200 km away from PDM. More specific comments:
- In the main text of Yus-Díez et al. (2021), I could find values between 2.51 and 2.36 for Montsec d'Ares. These values are calculated using a different type of filter and normalized against the MAAP at one single wavelength.
- Finding the value 3.22 in Yus-Diez requires some work since is available only in a table in the supplementary.
- In Yus-Diez, the value of 4.05 is specific for 880 nm and is retrieved by comparing the Aethalometer with an offline polar photometer, which is not commercially available nor widely diffused.
- 3.22 is a second factor obtained using the offline polar photometer working as a MAAP. Since no detailed information is provided in the supplementary of Yus-Díez et al. (2021), I believe that 3.22

is calculated normalizing the absorption of the offline polar photometer to the absorption of the MAAP.

- The use of a wavelength-dependent C, is justified in this specific case since SSA values are quite high at PDM. In this SSA region, C values might drastically increase.

Unfortunately, none of these points are addressed in the manuscript nor in the supplementary. So, the authors are required to describe and justify more in detail their choices. This comment should be taken very seriously since the same problem was observed in the first round of reviews.

**REPLY:** -The instrumental set-up did not allow us to determine the value of C, as we would have had to deploy another absorption-measuring instrument. We therefore used a C value obtained from the literature, as is done in the large majority of publications using aethalometer data. We chose a value obtained close to PDM (less than 200 km), on a similar type of site (remote mountain site) and with SSA values close to those observed at PDM. As highlighted by the reviewer, the high SSA values measured at the PDM supports the use of a C value adapted for 880 nm, which can be found in the supplementary material of Yus Diez et al. (2021). The precise information on where to find this information in Yuz-Diez et al. (2021) has been added in lines 146-149:

"The multiple scattering parameter used to correct the measured attenuation was set to 3.22, according to the value obtained at λ=880 nm by Yus-Diez et al. (2021) at the mountainous site of Montsec d'Ares located less than 200 km from the PDM (see Table S3 in the Supplements of Yus-Diez et al. 2021)."

- If a wavelength-dependent C is used at 880 nm, I imagine that the σap used to calculate SSA at 635, 525 and 450 nm are corrected with a wavelength-dependent C. Same applies to AAE and SAE.
- First submission: The mean σap,880 was 0.27 Mm−1 (L220). Second submission: The mean σap,880 was 0.27 Mm−1 (L267). So, are the data corrected for a C value of 3.22 or not?
L267: Since C was decreased from 3.63 to 3.22. I expect a value of σap higher than the previous submission (also 0.27 Mm-1). Please verify your numbers.

**REPLY:** We noticed a mistake on the C value written in the fist submission. In facts, the C value applied in the first and the second submission was the same value of 3.22. This is why the $\sigma_{abs,880}$ and SSA values did not change between the first and the second submission. Both AAE and SAE are independent of C. AAE is calculated as an absorption ratio, and there is no absorption in SAE calculation.

NOMENCLATURE
Work needs to be done to harmonise nomenclature:
- Although this might be interpreted as a single little mistake, it is an irritating one. The authors use "refractive BC" instead of "refractory BC" in the full text.
- Harmonise the nomenclature for scattering and absorption…. σsca -σabs or σsp -σap

**REPLY:** We apologize for these nomenclature mistakes which have been corrected. We changed $\sigma_{ap}$ to $\sigma_{abs}$.

BANDRATIO
In my previous review, I mentioned that colour ratio might be used (potentially) to distinguish direct biomass-burning events. Although the analysis of colour ratio usually provides very noisy results, which are hard to interpret, it could be used to identify the potential influence of BB (Schwarz et al., 2006; Dahlkötter, 2014). So, the author's statement "the color ratio only provides information on the presence of dustparticles" is fundamentally wrong. In the revised manuscript I did not expect to see a full colour-ratio analysis over two years, but, at least, a better justification on why it was not

**REPLY:** In their study, Schwarz et al., (2006) used the scattered/ incandescence peak light intensity ratio in order to characterize the rBC mixing state. However, the color ratio – defined as the broad-band incandescence signal over the narrow-band incandescence signal – has not been used to

analyze the rBC emission source or composition. Dahlkotter et al (2014) found different color ratios between rBC from a forest fire plume and rBC from a Creek fire plume. Nevertheless, as shown by (Moteki et al., 2010) the color ratio is also a function of the size and the shape of rBC particles. Thus, from our point of view the analyses of the color ratio for a proper source apportionment is delicate and less documented than the $\Delta M_{rBC}/\Delta CO$ ratio analysis.

PYTHON CODE
The SP2 community urgently needs open-source software to treat the data.
**REPLY:** We fully agree with the reviewer that there is a strong need of a SP2 data processing tool compatible with all operating systems. Our goal was to develop a Python program because the PSI SP2 toolkit was not suitable for our study, given the large volume of data to be processed and the compatibility with LINUX, which is used at Météo-France.

So, the authors should consider a more careful evaluation of the two codes. In this regard, I have some more comments:
- As written Text S1 suggest that the PSI toolkit is wrong while the python code is right. Indeed, the IGOR code has its limits, but it has been around for more than a decade, so I would be careful with some statements.
- When addressing the differences between the two codes, the authors should be able to properly identify the causes. Without a proper evaluation/comparison of the two codes, the concluding statements could also be rewritten as: "The Python code might be less sensitive than the PSI toolkit due to a different selection of valid individual signals. …"
- "These different possibilities have not been explored in detail". To be truly honest, this must have been done before the submission of the current manuscript. From my point of view, this is a crucial mistake from the authors.
**REPLY:** - We agree with the reviewer that a more robust and deeper work on the biases between all the SP2 toolkits – not only the PSI – should be done. However, this work requires additional analysis and statistical tools which are out of the scope of this paper.

- We changed the end of the Text S1 by using more nuanced arguments :
"The values provided by the PSI toolkit may be more noisy than the Python software due to different filters applied to the individual signals, a difference in the flowrate sampling, or a different estimation of the baseline of the incandescence peak height, leading to biases in individual masses."

- The mass difference can be mostly attributed to the calibration curve. I would be very curious to see a comparison between number concentration.
**REPLY:** We have quantified the part of the uncertainties due to the calibration curve (see Figure 1). By considering the average mass size distribution of the campaign, a difference of 8 % on the total $M_{rBC}$ was found between the two calibration curves. The bias in Figure S1 seems to be systematic and may be partly due to the calibration curve. However it does not explains the different amplitudes of $M_{BC}$ variations, which may be related to the flow rate sampling or the baseline calculation.

- If the igor code counts invalid signal as real particles, shouldn't the toolkit concentration be higher? Figure S1 shows the opposite.
- According to the values provided in the supplementary, analysing the same dataset with the IGOR toolkit would lead to a 20% lower MrBC and to a 20% higher MAC. I would not consider this difference to be negligible. If we consider that C values decreased from 3.63 to 3.22 (absorption increase of roughly 10%), the software and constant choices will introduce a MAC uncertainty of 30%.

**REPLY:** The PSI SP2 toolkit exhibits both positive and negative biases compared to our data treatment, leading to an overall lower mean $M_{rBC}$.

- As explained in the previous answers, the $\sigma_{abs,880}$ did not change between the first and the second submissions because the C values applied are the same.

[Figure]

*Figure 1: Calibration curve of the rBC mass concentration data processing using the PSI SP2 toolkit (in red) or the Python program of this study (in green).*

ABSORPTION ENHANCEMENT AND MAC

As already raised in the first review, I have my doubts about the relevance of Eabs, as calculated and treated here. It is worth mentioning absorption enhancement if the available data allow quantifying the mixing state of BC…since the lensing effect is a direct consequence of coating formation. Without data on coating thickness, no real optical closure could be presented. I thus suggest the authors to: 1) focus on MAC variability in the full paper rather than EABS; 2) dedicate a very short paragraph, listing all possible uncertainty (RI in primis), to the overall Eabs that might characterize on average PDM.

**REPLY:** Several studies cited in the paper in line 210 calculated $E_{abs}$ using a $MAC_{bare,rBC}$ value estimated from the Mie theory. The text from line 211 to line 219 already discusses the impact of the morphology assumption on $MAC_{bare,rBC}$ and the relevance of using Mie theory in its calculation. In the section 3.3.4 lines 390-392, the calculated $MAC_{bare,rBC}$ value are comparable with the values obtained in the literature and summarized by Liu et al. (2020).

We added a text in lines 215-218 to discuss the refractive index impact on $MAC_{bare,rBC}$:

"In addition to the morphology, the $MAC_{bare,rBC}$ calculation is also very sensitive to the refractive index of rBC core (Sorensen et al., 2018). Liu et al. (2020) summarized the changes in MAC values induced by the use of different refractive indexes. They reported deviations from -7 % to -35 % to the $MAC_{BC}$ value of 7.5 m² g⁻¹ recommended by Bond and Bergström (2006)."

SUMMARY AND IMPLICATIONS FOR CLIMATE MODELS

As already mentioned in the first round of reviews, the conclusions are too speculative. From my point of view, the new section worsened compared to the first submission. I strongly advise the authors to:
- Strictly and precisely describe their conclusive results. As it is, it is extremely hard to separate the results of this paper from previous works.
- Avoid any long discussions on global modelling and related parametrization of ageing, scavenging and absorption.

**REPLY:** The paragraph about implications for climate models was added following a suggestion of an other reviewer (RC #3 of the first round :"Lines 420-427: To me these are the most important lines in the manuscript – the implications of your findings. I am a little disappointed that this is relegated to one brief paragraph and that your major conclusion re: wet scavenging is not as thoroughly assessed in the paper as is likely warranted given the conclusion. I would like to see the implications section for climate models more rigorously discussed." )

UNITS

Many figures feature an unusual notation for units. As an example, "ng.m-3". From my experience, the use of a dot as a unit separator is unusual. None of the recently published ACP manuscripts presents this type of notation.

**REPLY:** The notation of the units has been changed.

SPECIFIC COMMENTS

L54: explain what Eabs is
**REPLY**: A description of $E_{abs}$ is already provided in lines 53-54:
"Numerous studies have demonstrated that coating of BC with non-absorbing materials is accompanied by an enhancement of light absorption ($E_{abs}$) through the so-called lensing effect "

L54-56: please, try to avoid such a long listing of references. My former supervisor would call this "lazy bibliography work". Try to identify the works most pertinent works needed to send your message and help the reader identify who did what.
**REPLY:** We mentioned many references to highlight the large number of studies.

L71-76: This description might fit better in Section 2.1
**REPLY:** This part has been moved to the Section 2.1.

L77: remove "in the indicated sections"
**REPLY:** This has been removed.

L92-93: Is this campaign called "Hygroscopic properties of black carbon"? Is this important information? If yes, please mention it in the abstract or introduction.
**REPLY:** The campaign is called h-BC for hygroscopic properties of black carbon. Event though no hygroscopic properties are presented is this paper, data from this paper were collected during this campaign.
The campaign has been mentioned in the introduction in line 70-71 as follows:
"This study presents two-year continuous measurements of BC and aerosol properties conducted during the Hygroscopic properties of Black Carbon (h-BC) campaign at the high-altitude long-term monitoring station Pic du Midi (PDM)."

L100: sampled air
**REPLY:** This has been modified.

L101: inside the room or in the inlet?
**REPLY:** The air was heated to 20 °C inside the inlet and maintained at this temperature inside the room.

L114 give a reference for the density.
**REPLY:** We added the reference of Moteki and Kondo (2010).

L123-125: Here I have the same question as in the first round of comments. Mode1 is extrapolated from the Sp2 measurements in the 90-100 nm range? If this is true, is it reasonable to fit a lognormal curve on 10 nm?
**REPLY:** As explained in the paper in lines 123-134, the rBC size distribution measured between 90 and 580 nm has been fitted with a sum of three modes, in order to minimize the differences between the measurements and the fit (see Fig. S2 a). The first mode was constrained between 50 and 100 nm, because these limits allow the representation of the first mode peaking at around 130 nm in the sum of the three modes.
A study on the impact of the fitting procedure – and in particular the representation of the rBC particles under 90 nm of diameter -  on $M_{rBC}$ is actually in preparation for submission in AMT journal.

L142: …,880,950 nm. Be consistent with line 137.
**REPLY:** This has been modified.

L217: The histogram in Figure S6 shows the dominance of periods with RH above 90%. I am wondering how many days have been removed from the two years period.
**REPLY:** We filtered the hours when relative humidity was above 95 %, which represented 24 % of the total hours in the campaign.

L253: please use the sectors indicated above.
**REPLY:** This has been modified.

L264-284: There is incongruency between the sequence of properties discussed in the text and presented in figure 3. If I am not wrong, panels b and c are not discussed here. So, to be consistent, $\neg$ap should be Figure 3b and $\neg$sp should be Figure 3c.
F3: I would not fill the gaps between march and august 2020 for panel a, b and e. SAE is shortly discussed and AAE is not mentioned in the text. I guess they can both be removed from figure 3, since they are included in Figure 4 and discussed after. It is particularly not nice to see the wavelengths in legend not in decreasing or incresing order 635-450-525, please correct.
**REPLY:** The order of the panels has been changed in Figure 3, as well as the legends in the SSA panel. The line between March and August has been removed.
The AAE panel has been removed in line with the reviewer comment. The SAE panel has been kept in the paper since its temporal variations are discussed in lines 275 and 280.

L279-280: Remove the sentence about BC. When speaking of BC you can recall absorption. Try to keep a linear sequence of topics and figures.
**REPLY:** The sentence about BC has been removed.

L285-309 and F4: I have the impression that the graph does not provide a clear "speciation" of the aerosol optical properties. If the authors want to draw some evident conclusions from this analysis, the time resolution of the AAE, SAE and SSA should be decreased to at least 1 day. Alternatively,

the Cappa method could be applied to summer-winter (on daily resolution)9, wet-dry and BL-FT cases (on hourly resolution)

**REPLY:** As shown in section 3.1, PDM is influenced by air masses from various locations and by dynamic and chemical processes on fine time scales. As a result, the aerosol optical properties at PDM cant be very different from one day to another. The hourly variations of $E_{abs}$ presented in section 3.4 and Fig. 7 suggest that a time resolution of one hour is appropriate to study the aerosol optical properties at the PDM. So we believe that an hourly time resolution for the classification of the dominant aerosol type is more appropriate.

S3.3: Maybe this is a problem of my PDF reader, but the numbering of subsections is missing, as in the first submission

**REPLY:** The subsections have been numbered.

F6: since the concentration is normalized, only one colour scale is needed

**REPLY:** Figure 6 has been edited.

L399-400: the diurnal variability of Eabs is poorly described. As it is, it does not provide crucial information and can be easily removed. Nonetheless, it might reflect, in terms of MrBC or BC/CO the daily cycle of BL and FT, as also shown by the authors in the reply to my first review. Excluding spring and summer, the diurnal analysis could be modified and used to introduce section 3.4.2. Clearly, this change will require some rethinking and additional work.

**REPLY:** Figure 7 was added to the paper to highlight the seasonal variability of $E_{abs}$ and in particular the opposite diurnal pattern between winter and summer. It is these diurnal and seasonal variations in $E_{abs}$ shown in the figure that motivated the approach chosen in section 3,4 to assess the factors influencing these variations.

The paragraph in lines 398-403 has been reorganized as follows:

"Figure 7 further shows the diurnal variation of $E_{abs}$ for each seasons. There was a notable opposite diurnal profile between seasons in $E_{abs}$ with midday showing a minimum around 1.7 in winter, and a maximum around 2.9 in summer. Spring and autumn showed intermediate patterns with less regular $E_{abs}$ throughout the day. These observations suggest that different sources and/or processes drove the seasonal contrast in rBC properties. The following section aims at investigating potential drivers of $E_{abs}$ variations, including rBC wet scavenging, dominant rBC sources and transport pathways. Particular attention will be paid to winter and summer because these seasons differ greatly, whereas spring and autumn behaviors appear intermediate"

L413-414: I am glad that the authors implemented my comments. However, I think that the statement here is not exact. The goal of removing BL periods is to decrease the influence of air masses with different BC/CO ratios caused by different sources.

**REPLY:** We thank the reviewer for his suggestion. We changed the sentence in line with the reviewer's suggestion as follows in lines 411-412 :

"In order to decrease the influence of the difference sources on $\Delta M_{rBC}/\Delta CO$ compared to the effect of wet scavenging, periods for which the site was under PBL influence were filtered."

L415: Looking at Figure 8a, a value of 2.1 ng m−3 ppbv−1 is associated with precipitation-free back trajectories. In the second round of reviews, this sort of mistake should be avoided.

**REPLY:** We corrected this mistake and thank the reviewer for its awareness.

L418: please remind the readers that measurements that occurred at RH above 90% are removed.

**REPLY:** A sentence has been added in line 416-417:

"As a reminder, time periods when PDM was under precipitations or humidity > 95 % have been filtered before the analysis."

L425-429:It would be nice to provide some numbers here. Otherwise is hard to compare with the studies cited (provide some numbers here too, please)

**REPLY:** The sentences in lines 429-437 have been modified and some numbers have been added: "However no significant change on the mean rBC core diameter was noticed between wet and dry conditions (mean $D_{rBC,core}$ of 177 and 182 nm, respectively), as well as in the presence of precipitations during the transport of rBC or not (mean $D_{rBC,core}$ of 177 and 182 nm, respectively). This result contrasts with previous studies showing a decrease in rBC size due to wet scavenging (Kondo et al., 2016; Moteki et al., 2012; Taylor et al., 2014; Liu et al., 2020a). For example, Kondo et al. (2016) found a change in $D_{rBC,core}$ between 13 and 20 nm depending on the season, while Liu et al. (2020) and Moteki et al. (2012) measured rBC cores ~ 32 nm lower in air masses affected by wet removal. The insignificant effect of wet scavenging on the modal diameter of rBC core size distribution could be explained by the size of rBC core sampled at PDM that was higher than the one described in these studies."

L434-441: The authors are considering only nucleation scavenging. Rightfully, fresh BC particles are hydrophobic, thus non-cloud-active. However, fresh and aged BC particles could be removed by wet scavenging below the cloud by impaction scavenging or inside the cloud by interstitial scavenging too. So, I would not indulge in a long discussion about supersaturation, when there might have been various competing removal mechanisms.

**REPLY:** Nucleation scavenging was found to be the most efficient process in the rBC removal (Jacobson, 2012). Interstitial scavenging affects mostly particles smaller than 100 nm (Pierce et al., 2015), which have a very low presence at PDM (rBC mean diameter of 180 nm). Thus, we think it is important to focus on the impact of rBC nucleation scavenging on $E_{abs}$. Following the relevant reviewer's remark, a sentence has been added to include the impaction process in the discussion in lines 442-445: "Furthermore, the rBC wet removal by impaction is also a size-dependent process which could has been responsible for the removal of small rBC particles (Croft et al., 2010). However interstitial scavenging affects mostly particles smaller than 100 nm (Pierce et al., 2015), which have a very low presence at PDM (rBC mean diameter of 180 nm)"

F8: Eabs is not discussed in the text. And I agree with this choice. But, why it is still shown in the figure? I would remove panel c and d and potentially replace it with Drbc, see following comment.

**REPLY:** $E_{abs}$ was discussed in the text in lines 423-424: "Figures 8 c-d show in contrast little influence of precipitation and RH on the rBC absorption enhancement, with a constant median $E_{abs}$ value of around ∼ 2.1."

F9: there is no need for two panels, show the absolute or normalized concentration. It is unclear to me how the RH lines are defined…RH>85% during precipitation period and RH<85% in no precipitation period? Potentially, Figure 9 could be merged with figure 8 removing the Eabs panels. Why it is "ng.m-3"? Remove this omnipresent point from units.

**REPLY:** The legend in Fig. 9 has been modified. The RH criteria is based on RH measurements at the PDM while precipitation criteria is based on the presence of precipitation along the HYSPLIT back-trajectories. Normalized plot has been deleted. The unit has been corrected.

L444: check the subscript for ΔmrBC/ΔCO

**REPLY:** This has been corrected.

L444-445: I already expressed my doubt on ΔMrBC calculated as MrBC. Figure 10a clearly shows that there is substantial variability in MrBC values depending on BL conditions. Especially in BL

conditions. So, I still think that ΔMrBC should be calculated as the difference between the background concentration and the current concentration. In any case, as already mentioned, if you decide to keep the current calculation, the ratio must be called MrBC/ ΔCO. Otherwise, this is misleading. Moreover, I cannot find the number associated with the results shown in Figure 10c.

**REPLY: :** Most studies have calculated $\Delta M_{rBC}$ as $M_{rBC}$ and kept the $\Delta$ in the notation (Choi et al., 2020; Kanaya et al., 2016; Kondo et al., 2016; Pani et al., 2019). They justified their choice by the shorter lifetime of rBC in the atmosphere – several days -  compared to the CO lifetime – a few months (Bey et al., 2001; Park et al., 2005). Although we understand the reviewer's point of view, we decided to use the same notation as in the literature.

The sentence in lines 462-465 has been modified according to the new numbering of Figure 10 and numbers have been added :

"The higher $\Delta M_{rBC}$ /$\Delta CO$ in PBL conditions (2.5 ± 2.3 ng m$^{-3}$ ppbv$^{-1}$ for the mean ± STD) than in FT conditions (0.6 ± 0.1 ng m$^{-3}$ ppbv$^{-1}$ for the mean ± STD) may indicate additional sources from biomass combustion from the valley (Fig. 10b), which could be attributed to either residential wood heating or stubble-burning that is still a common practice in the Pyrenees (González-Olabarria et al., 2015)."

L446-448: No values given for Eabs. Eabs should be shown in panel c and not b. Please try to maintain the same sequence in the text and in the figures/panels.

**REPLY:** Some values in lines 467-468 has been provided as follows :

"Figure 10c shows that PBL conditions were associated with lower $E_{abs}$ values (1.5 ± 0.3 for the mean ± STD) than FT conditions (1.9 ± 0.4 for the mean ± STD). "

The numbering in Figure 10 has been corrected.

L449-450: Please reformulate the sentence reporting the mean or median concentration….as written, it is weird

**REPLY:** The sentence in lines 456-458 has been modified as follows:

"In winter, we measured higher $M_{rBC}$ values and variability in PBL conditions (39.5, 30.0 and 105 ng m$^{-3}$ for the median, 25th, and 75th percentiles, respectively) than in FT conditions (33.5, 10.4 and 45.4 ng m$^{-3}$ for the median, 25$^{th}$, and 75$^{th}$ percentiles, respectively) (Fig. 10a)."

L452-454: unclear, please rephrase.

**REPLY:** The sentence in lines 460-461 has been modified as follows:

"During the night, pollution from the surface is trapped by the low height of the PBL and cannot reach the PDM. At the same time, the cleaner air transported in the FT may contribute to the dilution of $M_{rBC}$ at the PDM."

L459-460: Again, no values are provided.

**REPLY:** Some values has been added in lines 468-470 as follows:

"Surprisingly, the $M_{rBC}$ did not vary between BL and FT influence with values of 75.4 ± 33.2 (Mean ± STD) ng m$^{-3}$ and  80.2 ± 46.6 ng m$^{-3}$ respectively, meaning that the thermally driven PBL injection did not significantly impact $M_{rBC}$ measured at PDM (Figure 10d)

L468-478: rBC loading and BC/OC are shown in Fig 10d and f not d and e

**REPLY:** This has been corrected according to the new numbering.

L479-491: I still have doubts about this subsection. First, I do not find a clear reason explaining the variability of eabs. Second, Section 3.4.2 is supposed to discuss FT/BL dynamics, which is not treated in this part of the text and in Figure 11. In my opinion, all these parts should be removed.

**REPLY:** We have shown in section 3.4.2 that in summer, the influence of FT/BL does not explain

the $E_{abs}$ variability, and the diurnal aerosol size distribution shows a predominant transport in the FT in summer. Thus, an analysis on $\Delta M_{rBC}/\Delta CO$ ratios has been conducted to investigate the dominant rBC sources transported in the FT, which could explain the $E_{abs}$ variability. Thus, the paragraph in lines 489-501 and Figure 11 aims to provide further explanations on processes playing a role in the high $E_{abs}$ values in summer.

REFERENCES

Dahlkötter, F.: Airborne observations of black carbon aerosol layers at mid-latitudes, Technische Universität München, 2014.
Schwarz, J. P., Gao, R. S., Fahey, D. W., Thomson, D. S., Watts, L. A., Wilson, J. C., Reeves, J. M., Darbeheshti, M., Baumgardner, D. G., Kok, G. L., Chung, S. H., Schulz, M., Hendricks, J., Lauer, A., Kärcher, B., Slowik, J. G., Rosenlof, K. H., Thompson, T. L., Langford, A. O., Loewenstein, M., and Aikin, K. C.: Single-particle measurements of midlatitude black carbon and light-scattering aerosols from the boundary layer to the lower stratosphere, J. Geophys. Res. Atmospheres, 111, D16207, https://doi.org/10.1029/2006JD007076, 2006.
Yus-Díez, J., Bernardoni, V., Močnik, G., Alastuey, A., Ciniglia, D., Ivančič, M., Querol, X., Perez, N., Reche, C., Rigler, M., Vecchi, R., Valentini, S., and Pandolfi, M.: Determination of the multiple-scattering correction factor and its cross-sensitivity to scattering and wavelength dependence for different AE33 Aethalometer filter tapes: a multi-instrumental approach, Atmospheric Meas. Tech., 14, 6335–6355, https://doi.org/10.5194/amt-14-6335-2021, 2021.

BIBLIOGRAPHY

Bey, I., Jacob, D. J., Logan, Jennifer. A., & Yantosca, R. M. (2001). Asian chemical outflow to the Pacific in spring : Origins, pathways, and budgets. *Journal of Geophysical Research: Atmospheres*, *106*(D19), 23097-23113. https://doi.org/10.1029/2001JD000806

Choi, Y., Kanaya, Y., Park, S.-M., Matsuki, A., Sadanaga, Y., Kim, S.-W., Uno, I., Pan, X., Lee, M., Kim, H., & Jung, D. H. (2020). Regional variability in black carbon and carbon monoxide ratio from long-term observations over East Asia : Assessment of representativeness for black carbon (BC) and carbon monoxide (CO) emission inventories. *Atmospheric Chemistry and Physics*, *20*(1), 83-98. https://doi.org/10.5194/acp-20-83-2020

Croft, B., Lohmann, U., Martin, R. V., Stier, P., Wurzler, S., Feichter, J., Hoose, C., Heikkilä, U., van Donkelaar, A., & Ferrachat, S. (2010). Influences of in-cloud aerosol scavenging parameterizations on aerosol concentrations and wet deposition in ECHAM5-HAM. *Atmospheric Chemistry and Physics*, *10*(4), 1511-1543. https://doi.org/10.5194/acp-10-1511-2010

Jacobson, M. Z. (2012). Investigating cloud absorption effects : Global absorption properties of black carbon, tar balls, and soil dust in clouds and aerosols: CLOUD ABSORPTION EFFECTS. *Journal of Geophysical Research: Atmospheres*, *117*(D6), n/a-n/a. https://doi.org/10.1029/2011JD017218

Kanaya, Y., Pan, X., Miyakawa, T., Komazaki, Y., Taketani, F., Uno, I., & Kondo, Y. (2016). Long-term observations of black carbon mass concentrations at Fukue Island, western Japan, during 2009–2015 : Constraining wet removal rates and emission strengths from East Asia. *Atmospheric Chemistry and Physics*, *16*(16), 10689-10705. https://doi.org/10.5194/acp-16-10689-2016

Kondo, Y., Moteki, N., Oshima, N., Ohata, S., Koike, M., Shibano, Y., Takegawa, N., & Kita, K. (2016). Effects of wet deposition on the abundance and size distribution of black carbon in East Asia. *Journal of Geophysical Research: Atmospheres*, *121*(9), 4691-4712. https://doi.org/10.1002/2015JD024479

Moteki, N., Kondo, Y., & Nakamura, S. (2010). Method to measure refractive indices of small nonspherical particles : Application to black carbon particles. *Journal of Aerosol Science*, *41*(5), 513-521. https://doi.org/10.1016/j.jaerosci.2010.02.013

Pani, S. K., Ou-Yang, C.-F., Wang, S.-H., Ogren, J. A., Sheridan, P. J., Sheu, G.-R., & Lin, N.-H. (2019). Relationship between long-range transported atmospheric black carbon and carbon monoxide at a high-altitude background station in East Asia. *Atmospheric Environment, 210,* 86-99. https://doi.org/10.1016/j.atmosenv.2019.04.053

Park, R. J., Jacob, D. J., Palmer, P. I., Clarke, A. D., Weber, R. J., Zondlo, M. A., Eisele, F. L., Bandy, A. R., Thornton, D. C., Sachse, G. W., & Bond, T. C. (2005). Export efficiency of black carbon aerosol in continental outflow : Global implications. *Journal of Geophysical Research: Atmospheres, 110*(D11). https://doi.org/10.1029/2004JD005432

Pierce, J. R., Croft, B., Kodros, J. K., D'Andrea, S. D., & Martin, R. V. (2015). The importance of interstitial particle scavenging by cloud droplets in shaping the remote aerosol size distribution and global aerosol-climate effects. *Atmospheric Chemistry and Physics*, *15*(11), 6147-6158. https://doi.org/10.5194/acp-15-6147-2015

Schwarz, J. P., Gao, R. S., Fahey, D. W., Thomson, D. S., Watts, L. A., Wilson, J. C., Reeves, J. M., Darbeheshti, M., Baumgardner, D. G., Kok, G. L., Chung, S. H., Schulz, M., Hendricks, J., Lauer, A., Kärcher, B., Slowik, J. G., Rosenlof, K. H., Thompson, T. L., Langford, A. O., … Aikin, K. C. (2006). Single-particle measurements of midlatitude black carbon and light-scattering aerosols from the boundary layer to the lower stratosphere. *Journal of Geophysical Research: Atmospheres, 111*(D16). https://doi.org/10.1029/2006JD007076

Taylor, J. W., Allan, J. D., Liu, D., Flynn, M., Weber, R., Zhang, X., Lefer, B. L., Grossberg, N., Flynn, J., & Coe, H. (2015). Assessment of the sensitivity of core / shell parameters derived using the single-particle soot photometer to density and refractive index. *Atmospheric Measurement Techniques*, *8*(4), 1701-1718. https://doi.org/10.5194/amt-8-1701-2015

Zanatta, M., Gysel, M., Bukowiecki, N., Müller, T., Weingartner, E., Areskoug, H., Fiebig, M., Yttri, K. E., Mihalopoulos, N., Kouvarakis, G., Beddows, D., Harrison, R. M., Cavalli, F., Putaud, J. P., Spindler, G., Wiedensohler, A., Alastuey, A., Pandolfi, M., Sellegri, K., … Laj, P. (2016). A European aerosol phenomenology-5 : Climatology of black carbon optical properties at 9 regional background sites across Europe. *Atmospheric Environment*, *145,* 346-364. https://doi.org/10.1016/j.atmosenv.2016.09.035

Zhang, X., Mao, M., Yin, Y., & Wang, B. (2018). Numerical Investigation on Absorption Enhancement of Black Carbon Aerosols Partially Coated With Nonabsorbing Organics. *Journal of Geophysical Research: Atmospheres*, *123*(2), 1297-1308. https://doi.org/10.1002/2017JD027833

---

## Author Response (AR3)

**Answer to the referee**
**Higher absorption enhancement of black carbon in summer shown by two year measurements at the high-altitude mountain site of Pic du Midi Observatory in the French Pyrenees**
**(Preprint egusphere-2023-570)**

Please find below the reviewer's comments in black and our responses in blue. The line numbers in the responses refer to the new version of the paper.

The authors have addressed all of my technical comments. The potential biases introduced by using different software for the data treatment of the SP2 data are now adequately explained. The numbering of figures and sections, as well as the nomenclature and the use of units, has been made consistent.

The study on the impact of BL dynamics and wet removal on BC properties is the core of the work, features a unique dataset and, most importantly, represents the outstanding novelty of the present research. However, I find that the general characterization of the aerosol and its seasonality (Sections 3.1, 3.2, and 3.3) dilutes and diminishes the visibility of Section 3.4.

I strongly recommend that the authors summarize and reduce the results and discussion of the general characterization to give more emphasis to Section 3.4. I maintain the opinion that the authors should exercise greater caution in discussing the implications for climate models, given the absence of hygroscopicity measurements or mixing state measurements.

Lastly, I suggest a final check of grammar and language.

Section 3.1 describing the meteorological parameters has been moved in the Supplements together with Fig. S6, according to the suggestion of the reviewer.

Documenting the seasonal variability of aerosol and BC properties relevant to climate studies through long-term measurements suited to users is essential. It is crucial for the modelling community to have access to both intensive and extensive parameters describing aerosol properties in order to evaluate and improve the aerosol representation in models and to better quantify their climate effects. Therefore we believe that parts 3.2 to 3.3 are as important as the discussions in Section 3,4. relating to the identification of atmospheric processes.

A sentence has been added in the conclusion to moderate interpretation of photochemistry: " However, the latter effect could not be rigorously demonstrated in this study."

A final proofreading by all co-authors was carried out. However, the reviewer must keep in mind that we are not English-native speakers.